# Efficient Sampling of Stochastic Differential Equations with Positive Semi-Definite Models

**Anant Raj**
Coordinated Science Laboraotry
University of Illinois Urbana-Champaign.
Inria, Ecole Normale Supérieure
PSL Research University, Paris, France.
`anant.raj@inria.fr`

**Umut Şimşekli**
Inria, CNRS, Ecole Normale Supérieure
PSL Research University, Paris, France.
`umut.simsekli@inria.fr`

**Alessandro Rudi**
Inria, Ecole Normale Supérieure
PSL Research University, Paris, France.
`alessandro.rudi@inria.fr`

## Abstract

This paper deals with the problem of efficient sampling from a stochastic differential equation, given the drift function and the diffusion matrix. The proposed approach leverages a recent model for probabilities [RC21] (the positive semi-definite – PSD model) from which it is possible to obtain independent and identically distributed (i.i.d.) samples at precision $\varepsilon$ with a cost that is $m^2 d \log(1/\varepsilon)$ where $m$ is the dimension of the model, $d$ the dimension of the space. The proposed approach consists of: first, computing the PSD model that satisfies the Fokker-Planck equation (or its fractional variant) associated with the SDE, up to error $\varepsilon$, and then sampling from the resulting PSD model. Assuming some regularity of the Fokker-Planck solution (i.e. $\beta$-times differentiability plus some geometric condition on its zeros) We obtain an algorithm that: (a) in the preparatory phase obtains a PSD model with L2 distance $\varepsilon$ from the solution of the equation, with a model of dimension $m = \varepsilon^{-(d+1)/(\beta-2s)}(\log(1/\varepsilon))^{d+1}$ where $1/2 \leq s \leq 1$ is the fractional power to the Laplacian, and total computational complexity of $O(m^{3.5} \log(1/\varepsilon))$ and then (b) for Fokker-Planck equation, it is able to produce i.i.d. samples with error $\varepsilon$ in Wasserstein-1 distance, with a cost that is $O(d\varepsilon^{-2(d+1)/\beta-2} \log(1/\varepsilon)^{2d+3})$ per sample. This means that, if the probability associated with the SDE is somewhat regular, i.e. $\beta \geq 4d + 2$, then the algorithm requires $O(\varepsilon^{-0.88} \log(1/\varepsilon)^{4.5d})$ in the preparatory phase, and $O(\varepsilon^{-1/2} \log(1/\varepsilon)^{2d+2})$ for each sample. Our results suggest that as the true solution gets smoother, we can circumvent the curse of dimensionality without requiring any sort of convexity.

## 1 Introduction

High dimensional stochastic differential equations (SDE) and their associated partial differential equations (PDEs) arise often in various scientific and engineering applications such as control systems, aviation, fluid dynamics, etc [LT00, HF08, Trö10]. Sampling from an SDE is an active area of research. Generally, a direct approach like Langevin sampling [DMM19, DK19, CDJB20] is commonly employed to sample from SDEs. However, an alternative approach involves approximating the solution of the associated PDE in a form that enables easy sampling. In this paper, we focus on the latter approach.

37th Conference on Neural Information Processing Systems (NeurIPS 2023).

The exact solutions of most of the high-dimensional PDEs are not computable in closed form. Hence, for this reason, finding approximate numerical solutions to high-dimensional SDEs and related PDEs has been an important area of research in numerical methods [Ame14, FGN13, Tad12]. With the advent of new machine learning methods, the question naturally arises as to whether we can leverage the expressive power of these models to effectively capture the solution of high-dimensional partial differential equations (PDEs). Recent research efforts have explored this direction by employing various machine learning models for different classes of PDEs [BHJK20, LLMD18, KKL+21]. For instance, works such as [CHOS21, Ste21, BQL+22] have harnessed the capabilities of reproducing kernel Hilbert spaces (RKHS) to model PDE solutions. With the recent developments in neural network models and architecture, [LLMD18, KLY21, KPRS22, CMW+21] consider neural networks to model and approximate the solution of high dimensional PDEs. Despite impressive algorithmic innovation, the theoretical understanding of these approaches is still limited, leaving a significant class of PDEs beyond the reach of existing machine learning-based modeling frameworks or with inadequate theoretical insights into the obtained solutions.

In this paper, our goal is to sample from the stochastic differential equations (SDE) that are driven by a Brownian motion or an $\alpha$-stable process. The time evolution of probability density functions (PDFs) of stochastic differential equations (SDE) that are driven by a Brownian motion or an $\alpha$-stable process is given by Fokker-Planck Equation (FPE) and the fractional Fokker-Planck Equation (fractional FPE) respectively (see e.g, [DMM19, Şim17, HMW21]. Therefore, our focus in this study lies in approximating the solution of the Fokker-Planck Equation (FPE), along with its fractional counterpart, as discussed in [UHK18], and subsequently sampling from the approximated solution. The modeling of solutions for FPEs (FPEs) and fractional FPEs poses significant challenges, encompassing the following key aspects: (i) Solutions of FPEs and fractional FPEs are probability densities that are always non-negative and vanish at infinity, (ii) Modeling a probability density is hard because of the normalization property, and (iii) Fractional FPEs involve a fractional Laplacian operator that is non-local in nature. In order for an approximate solution of FPEs or fractional FPEs to be useful for sampling from the corresponding stochastic differential equations (SDEs), it is necessary that sampling from the approximate solution of the corresponding partial differential equation (PDE) be straightforward. Unfortunately, a comprehensive algorithmic approach that addresses the above-mentioned issues and includes a formal approximation analysis is currently lacking in the literature.

This paper adopts a positive semi-definite (PSD) model-based approach to effectively tackle the problem at hand. The recently proposed PSD model [MFBR20, RMFB20, RC21] offers promising solutions to the challenges encountered in probability density modeling, making it an excellent choice for representing the solution of FPEs and their fractional counterparts. In a recent work by [RC21], it was demonstrated that employing a Gaussian kernel in the PSD model allows for the exact determination of the normalization constant and enables the characterization of a broad range of probability densities using such PSD models. Moreover, an algorithm for sampling from the PSD model is also proposed recently with statistical guarantees [MFBR22].

Driven by these insights, we explore the utilization of a PSD model for approximating the solution of FPEs and fractional FPEs. Consequently, we employ the sampling algorithm presented in[MFBR22] to generate samples from the approximate solution. Our primary objective in this study is to obtain rigorous guarantees on the approximation error of PSD models when applied to the approximation of solutions of FPEs and fractional FPEs. Under regularity conditions on the solution of these PDEs, we make the following contributions.

**Contributions:**    We make the following contributions in this paper:

- We show that the PSD-based representation of the solution of FPE has good approximation properties under the regularity assumptions on the true solution of the PDE. In particular, we show that we can achieve an approximation error of $O(\varepsilon)$ while using $m = O(\varepsilon^{-(d+1)/(\beta-2)}(\log \frac{1}{\varepsilon})^{(d+1)/2})$ number of samples for PSD representation where $d$ is the dimension of space variables and the true solution is $\beta$ times differentiable (**Theorem 1 and 2**). In addition, we are able to provide Wasserstein guarantees for the sampling error of a PSD model-driven solution when sampling from an SDE driven by a brownian motion (**Corollary 1**). An important contribution of our approach is that it relies on much weaker assumptions on the solution of associated PDE ($\beta$ times differentiable) compared to existing methods that employ dissipativity, log-Sobolev, and Poincaré like conditions to provide sampling error guarantees [DMM19, DK19, Dal17, DM17, CEL+21, BCE+22].

- We demonstrate that utilizing a positive semi-definite (PSD) based representation is a natural and effective method to approximate the fractional Laplacian operator when applied to a probability density. In particular, we provide results that illustrate the effectiveness of PSD-based representations for Gaussian and shift-invariant kernels, which can be represented using Bochner's theorem (**Theorem 3**).

- As a final contribution, we also establish an approximation guarantee for the solution of the fractional FPE under certain regularity assumptions on the true solution. This extends our earlier result for the FPE and demonstrates the effectiveness of the PSD-based approach in approximating the solution of both FPE and fractional FPE We achieve an approximation error of $O(\varepsilon)$ while using $m = O(\varepsilon^{-(d+1)/(\beta-2s)}(\log \frac{1}{\varepsilon})^{(d+1)/2})$ number of samples for PSD representation where $d$ is the dimension of space variables and the true solution is $\beta$ times differentiable (**Theorem 4 and 5**). This result directly implies a Wasserstein guarantee on the sampling error when sampling from an SDE driven by an $\alpha$-stable levy process using the PSD model. To the best of our knowledge, our algorithm stands as the first of its kind, providing a provable guarantee for sampling from an SDE driven by an $\alpha$-stable levy process as well as for approximating the solution of fractional FPEs under regularity assumption on the solution.

Our results clearly suggest that as the true solution gets smoother, we require less number of samples to approximate the true solution using PSD-based representation, hence circumventing the curse of dimensionality. All the proofs are provided in the appendix.

## 2 Related Work

[Eva22] provides a good introduction on PDEs. [GW79, Ise09, Ame14] have discussed numerical methods to solve a PDE at length.

**Machine learning-based solution to PDEs:**  Modeling the solution of a PDE using machine learning approaches and in particular kernel methods is a well-established technique leading to optimal sample complexity and very efficient algorithms (see for example [BKO+96, SW06, Sch07, Fas07, PR20, CHOS21, Ste21]). With the recent advancement of neural networks in machine learning, new approaches have emerged. A deep learning-based approach that can handle general high-dimensional parabolic PDEs was introduced in [HJE18]. Two popular approaches that were proposed in this context are (i) *Physics inspired neural networks (PINN)* [RPK19] and (ii) *Deep Ritz method* [Y+18]. However, no statistical optimality was provided in the original papers. Recently, [LCL+21] studied the statistical limits of deep learning and Fourier basis techniques for solving elliptic PDEs. The sample complexity to obtain a solution via a PINN and Deep Ritz model are shown to be $O\left(n^{-\frac{2\beta-2}{d+2\beta-4}} \log n\right)$ and $O\left(n^{-\frac{2\beta-2}{d+2\beta-2}} \log n\right)$ respectively where $\beta$ is the order of smoothness of the solution. Subsequently, in a series of papers [DRM22, LL22, LLW21, CLL21], the theoretical properties of solving a different class of PDEs are discussed under restrictive assumptions on the function class (barron space). For an overview, we refer to [BHJK20]. Solutions of PDEs that involve a fractional operator have also been modeled by function in RKHS [Arq18, AA20, ASA19, AAS20]. However, these works do not provide finite sample approximating error guarantees for the solution obtained. A review of kernel-based numerical methods for PDEs can be found in [SS+16, FF15].

**The (fractional) Fokker-Planck equation:**  Various efforts have been made to approximate the solution of FPE and its fractional counterpart by numerical methods [TSIG22, PMB13, Yan13, LAT04, Bar01]. [KRP12, DM14, SS22] utilized a radial basis kernel-based approach to numerically compute the solution of an FPE. RKHS-based methods have also been used to obtain an approximate solution [BH17, SAA20]. However, the above mentioned works are numerical in nature and do not discuss the approximation limits of the solution. On the more applied side, various works have been proposed by harnessing the expressive power of deep networks to model the solution of FPEs and fractional FPEs [XZL+20, ZDL22, WWLZ22]. However, a systematic theoretical study is missing from the literature.

**Langevin Monte Carlo:**  An important line of research considers the probabilistic counterpart of FPE that is directly simulating the SDEs associated with the FPEs by time-discretization. Under convexity assumptions [Dal17, DM17, DM19] proved guarantees for obtaining a *single* (approximate) sample from the SDE for a given time. These frameworks were later extended in [RRT17, XCZG18, EH21], so that the convexity assumptions were weakened to more general notions such as uniform dissipativity. [MCF15] introduced a more general SDE framework, which covers a larger class of PDEs. In the case of sampling SDEs with domain constraints [BEL15, BDMP17, HKRC18, SR20]

proposed Langevin-based methods under convexity assumption, which were later weakened in [Lam21, ZL22]. [Şim17, NSR19] considered SDEs which corresponded to the fractional FPE in dimension one and this approach was later extended in [HMW21, ZZ22]. An overview of these methods is provided in [NF21]. Contrary to this line of research, which typically requires convexity or dissipativity, our framework solely requires smoothness conditions.

**PSD Models:** PSD models are an effective way to model non-negative functions and enjoy the nice properties of linear models and more generally RKHS. It was introduced in [MFBR20] and their effectiveness for modeling probability distributions was shown in [RC21]. It has been also effectively used in optimal transport estimation [MVB+21, VMR+21], in finding global minima of a non-convex function [RMFB20], in optimal control [BCRB22]. In [MFBR22], an algorithm to draw samples from a PSD model was proposed.

## 3  Notations and Background

**Notation:** Space input domain is denoted as $\mathcal{X}$ and time input domain is denoted as $\mathcal{T}$. Combined space $\mathcal{X} \times \mathcal{T}$ is denoted as $\tilde{\mathcal{X}}$. $L^\infty(\tilde{\mathcal{X}})$, $L^1(\tilde{\mathcal{X}})$ and $L^2(\tilde{\mathcal{X}})$ denote respectively the space of essentially bounded, absolutely integrable and square-integrable functions with respect to Lebesgue measure over $\tilde{\mathcal{X}}$. $W_2^\beta(\tilde{\mathcal{X}})$ denotes the Sobolev space of functions whose weak derivatives up to order $\beta$ are square-integrable on $\tilde{\mathcal{X}}$. We denote by $\mathbb{R}_{++}^d$ the space vectors in $\mathbb{R}^d$ with positive entries, $\mathbb{R}^{n \times d}$ the space of $n \times d$ matrices, $\mathbb{S}_+^n = \mathbb{S}_+(\mathbb{R}^n)$ the space of positive semi-definite $n \times n$ matrices. Given a vector $\eta \in \mathbb{R}^d$, we denote $\operatorname{diag}(\eta) \in \mathbb{R}^{d \times d}$ the diagonal matrix associated to $\eta$.

### 3.1  (Fractional) Fokker-Planck Equation

The FPE describes the time evaluation of particle density if the particles are moving in a vector field and are also influenced by random noise like Brownian motion. Let us consider the following SDE driven by the standard Brownian motion (Wiener process) $W_t$:

$$dX_t = \mu(X_t, t)dt + \sigma dW_t \tag{1}$$

where $X_t \in \mathbb{R}^d$ is a random variable, $\mu(X_t, t) \in \mathbb{R}^d$ is drift, $\sigma \in \mathbb{R}^{d \times k}$ and $W_t$ is a $k$-dimensional Wienner process. $\mu_i(X_t, t)$ denotes the $i$th element of vector $\mu(X_t, t)$. The diffusion tensor[1] is defined by $D = \frac{1}{2}\sigma\sigma^\top$.

The probability density function $p(x, t)$ of the random variable $X_t$ solving the SDE (1) is given by the FPE, given as follows [RR96, Pav14]:

$$\frac{\partial p(x, t)}{\partial t} = -\sum_{i=1}^d \frac{\partial}{\partial x_i}(\mu_i(x, t)p(x, t)) + \sum_{i=1}^d \sum_{j=1}^d D_{ij}\frac{\partial^2}{\partial x_i \partial x_j}p(x, t), \text{ such that } p(0, x) = p_0(x). \tag{2}$$

It is known that when $\mu(X_t, t)$ is the gradient of a potential or objective function $f : \mathcal{X} \to \mathbb{R}$, i.e., $\mu = \nabla f$, where $f$ satisfies certain smoothness and growth conditions, the stationary distribution of the stochastic process in equation (1) exists and is described by the so-called Gibbs distribution [Pav14].

Despite of having nice theoretical properties, Brownian-driven SDEs are not suitable in certain models in engineering, physics and biology [STL96, Dua15]. A popular non-Gaussian SDE used in such applications is given as follows:

$$dX_t = \mu(X_t, t)dt + dL_t^\alpha, \tag{3}$$

where $L_t^\alpha$ denotes the (rotationally invariant) $\alpha$-stable Lévy process in $\mathbb{R}^d$ and is defined as follows for $\alpha \in (0, 2]$ [Dua15]:

1. $L_0^\alpha = 0$ almost surely;
2. For any $t_0 < t_1 < \cdots < t_N$, the increments $L_{t_n}^\alpha - L_{t_{n-1}}^\alpha$ are independent;
3. The difference $L_t^\alpha - L_s^\alpha$ and $L_{t-s}^\alpha$ have the same distribution, with the characteristic function $\mathbb{E}\left[e^{j\langle u, L_{t-s}^\alpha\rangle}\right] = \exp(-(t-s)^\alpha\|u\|_2^\alpha)$ for $t > s$ and $j = \sqrt{-1}$;
4. $L_t^\alpha$ is continuous in probability.

In the SDE described in equation (3), we choose $\sigma = 1$ for the simplicity of the expressions. When $\alpha = 2$, the process recovers the Brownian motion, i.e., $L_t^\alpha = \sqrt{2}W_t$. The fundamental difference

---

[1] To simplify our analysis, we will consider diffusion tensor $D$ independent from $X_t$ and $t$ in this work.

between $L_t^\alpha$ and $W_t$ when $\alpha < 2$ is that the increments of this process are heavy-tailed, with the following property: $\mathbb{E}[\|L_t^\alpha\|^p] = +\infty$ for all $p \geq \alpha$ and $t \geq 0$. This heavy-tailed structure makes the solution paths of (3) discontinuous: $(X_t)_{t \geq 0}$ can exhibit at most countable number of discontinuities. SDEs based on $\alpha$-stable processes have also received some attention in machine learning as well [SSG19, HMW21, NSGR19, SSDE20, ZFM$^+$20]. The resulting governing equation for stable-driven SDEs is similar to the traditional Fokker–Planck equation except that the order $\alpha$ of the highest derivative is fractional when $\alpha < 2$. The following generalization of the FPE is known in the literature as fractional FPE because of the presence of fractional Laplacian operator [UHK18]:

$$\frac{\partial p(x,t)}{\partial t} = -\sum_{i=1}^{d} \frac{\partial}{\partial x_i}(\mu_i(x,t)p(x,t)) - (-\Delta)^{\alpha/2}p(x,t), \text{ such that } p(0,x) = p_0(x), \qquad (4)$$

where $(-\Delta)^s$ is the non-local Laplacian operator which is also known as the fractional Laplacian operator and is induced by the stable process. For $0 < s < 1$, the fractional Laplacian of order $s$, $(-\Delta)^s$ can be defined on functions $f : \mathbb{R}^d \to \mathbb{R}$ as a singular integral as follows: $(-\Delta)^s f(x) = c_{d,s} \int_{\mathbb{R}^d} \frac{f(x)-f(y)}{\|x-y\|_2^{d+2s}} \, dy$, where $c_{d,s} = \frac{4^s \Gamma(d/2+s)}{\pi^{d/2}|\Gamma(-s)|}$. A more useful representation of the fractional Laplacian operator is given as a Fourier multiplier, $\mathcal{F}[(-\Delta)^s f](u) = \|u\|^{2s} \mathcal{F}[f](u)$, where $\mathcal{F}[f](u) = \int_{\mathbb{R}^d} f(x)e^{jx^\top u} \, dx$. We will utilize the Fourier multiplier-based representation of the fractional Laplacian operator to prove our approximation result for the fractional FPE.

### 3.2 PSD Models for Probability Representation

Consider a feature map representation $\phi : \mathcal{X} \to \mathcal{H}$ from input space $\mathcal{X}$ to a Hilbert space $\mathcal{H}$, and a linear operator $M \in \mathbb{S}_+(\mathcal{H})$, then PSD models are represented as in [MFBR20],

$$f(x; M, \phi) = \phi(x)^\top M \phi(x). \qquad (5)$$

It is clear that $f(x; M, \phi)$ is a non-negative function and hence PSD models offer a general way to parameterize non-negative functions . We consider $\mathcal{H}$ to be the RKHS corresponding to the kernel $k$ such that $\phi(x) = k(x, \cdot)$. In particular, we will consider the case where: $\phi = \phi_\eta : \mathbb{R}^d \to \mathcal{H}_\eta$ is the feature map associated with the Gaussian kernel $k_\eta(x,y) = \phi_\eta(x)^\top \phi_\eta(y) = e^{-\eta\|x-y\|^2}$ with $\eta > 0$. Let us assume that $M$ lies in the span of feature maps corresponding to $n$ data points, $X = \{x_1, \cdots, x_n\}$ then $M = \sum_{i,j} A_{ij}\phi(x_i)\phi(x_j)^\top$ for some $A \in \mathbb{S}_+^n$. Hence, a Gaussian PSD model can be defined as, $f(x; A, X, \eta) = \sum_{ij} A_{ij}k_\eta(x_i,x)k_\eta(x_j,x)$. Given two base point matrices $X \in \mathbb{R}^{n \times d}$ and $X' \in \mathbb{R}^{m \times d}$, then $K_{X,X',\eta}$ denotes the kernel matrix with entries $(K_{X,X',\eta})_{ij} = k_\eta(x_i, x'_j)$ where $x_i$ and $x'_j$ are the $i$-th and $j$-th rows of $X$ and $X'$ respectively. Gaussian PSD model has a lot of interesting properties and the details can be found in [RC21]. But, we would like to mention below an important aspect of PSD models which enable them to be used as an effective representation for probability density:

**Sum Rule (Marginalization and Integration)**: The integral of a PSD model can be computed as,

$$\int f(x; A, X, \eta) \, dx = c_{2\eta}\text{Tr}(AK_{X,X,\frac{\eta}{2}}), \text{ where } c_\eta = \int k(0,x) \, dx.$$

Similarly, only one variable of a PSD model can also be integrated to result in the sum rule.

**Proposition 1** (Marginalization in one variable [RC21]). *Let $X \in \mathbb{R}^{n \times d}$, $Y \in \mathbb{R}^{\kappa \times \kappa'}$, $A \in \mathbb{S}_+(\mathbb{R}^n)$ and $\eta, \eta' \in \mathbb{R}+$, then the following integral is a PSD model,*

$$\int f(x,y; A, [X,Y], (\eta, \eta')) \, dx = f(y; B, Y, \eta') \text{ with } B = c_{2\eta}A \circ K_{X,X,\frac{\eta}{2}}.$$

In a recent work, [MFBR22] gave an efficient algorithm to sample from a PSD model, with the following result

**Proposition 2** (Efficient sampling from PSD models [MFBR22]). *Let $\varepsilon > 0$. Given a PSD model $f(x; A, \eta)$ with $A \in \mathbb{R}^{m \times m}$ for $x \in \mathbb{R}^d$. There exists an algorithm (presented in [MFBR22]) that samples i.i.d. points from a probability $\tilde{f}_\varepsilon$ such that the 1-Wasserstein distance between $f(\cdot; A, \eta)$ and $\tilde{f}_\varepsilon$ satisfies $\mathbb{W}_1(f, \tilde{f}_\varepsilon) \leq \varepsilon$. Moreover, the cost of the algorithm is $O(m^2 d \log(d/\varepsilon))$.*

## 4    Approximation of the (Fractional) Fokker-Planck Equation

In this section, we investigate the approximating properties of PSD models for solving FPEs and fractional FPEs. Our focus is on the approximation guarantee achieved by fitting a single PSD model

in the joint space-time domain. Let's denote the solution of the PDE as $p^\star(x,t)$, where we consider the FPE in section 4.1 and the fractional FPE in section 4.2. We introduce mild assumptions on the domain and the solution probability density. Throughout the paper, we assume the spatial domain $\mathcal{X}$ to be $(-R, R)^d$ and the time domain $\mathcal{T}$ to be $(0, R)$. We have the following assumption from [RC21] on the solution $p^\star$ of the PDE,

**Assumption 1.** *Let $\beta > 2$, $q \in \mathbb{N}$. There exists $f_1, f_2, \cdots, f_q \in W_2^\beta(\tilde{\mathcal{X}}) \cap L^\infty(\tilde{\mathcal{X}})$, such that the density $p^\star : \tilde{\mathcal{X}} \to \mathbb{R}$ satisfies, $p^\star(x,t) = \sum_{i=j}^q f_j^2(x,t)$.*

The assumption above is quite general and satisfied by a wide family of probabilities, as discussed in the Proposition 5 of [RC21]. We reiterate the result here below.

**Proposition 3** (Proposition 5, [RC21])**.** *The assumption above is satisfied by*

    (i) *any $p^\star(x,t)$ that is $\beta$-times differentiable and strictly positive on $[-R, R]^d \times [0, R]$,*

    (ii) *any exponential model $p(x,t) = e^{-v(x,t)}$ such that $v \in W_2^\beta(\tilde{\mathcal{X}}) \cap L^\infty(\tilde{\mathcal{X}})$,*

    (iii) *any mixture model of Gaussians or, more generally, of exponential models from (ii),*

    (iv) *any $p$ that is $\beta + 2$-times differentiable on $[-R, R]^d \times [0, R]$ with a finite set of zeros in $\tilde{\mathcal{X}}$ and with positive definite Hessian in each zero.*

*Moreover when $p^\star$ is $\beta$-times differentiable over $[-R, R]^d \times [0, R]$, then it belongs to $W_2^\beta(\tilde{\mathcal{X}}) \cap L^\infty(\tilde{\mathcal{X}})$.*

We will need to impose some extra mild assumptions on $f_j$ for $j \in \{1, \cdots, q\}$ to approximate the solution of a fractional Fokker-Planck equation because of involvement of the fractional laplacian operator (non-local operator). We will cite these extra assumptions when we describe the approximation results in section 4.2. Apart from that, we also have a boundedness assumption for the coefficients of the PDE in the domain we consider.

**Assumption 2.** *We have for all $(x,t) \in \tilde{\mathcal{X}}$, $\mu_i(x,t) \le R_\mu$, $\frac{\partial \mu_i(x,t)}{\partial x_i} \le R_{\mu_p}$, and $D_{ij} \le R_d$ for all $(i,j) \in \{1, \cdots, d\}$.*

It is also obvious to see that the PSD-based representation of probability density vanishes at infinity.

## 4.1 Approximating Solution of Fokker-Planck Equation

Let us consider the Fokker-Planck equation given in equation (2) and $p^\star(x,t)$ is the solution of Fokker-Planck equation in equation (2) which satisfies the assumption 1. Hence, $p^\star(x,t)$ satisfies the following,

$$\frac{\partial p^\star(x,t)}{\partial t} = -\sum_{i=1}^d \frac{\partial}{\partial x_i}(\mu_i(x,t)p^\star(x,t)) + \sum_{i=1}^d \sum_{j=1}^d D_{ij} \frac{\partial^2}{\partial x_i \partial x_j} p^\star(x,t).$$

For now, we are ignoring the initial value as we only are interested in approximation error and we will have a uniform approximation of the solution density $p^\star$ and its derivatives up to the order 2 in the domain.

Given $m$ base points pair $(\tilde{x}_i, \tilde{t}_i) \in \tilde{\mathcal{X}}$ for $i \in \{1, 2, \cdots, m\}$ of space and time, let us consider the following approximation model for the solution of FPE,

$$\tilde{p}(x,t) = \sum_{i,j} A_{ij} k_{X,T}((x,t), (\tilde{x}_i, \tilde{t}_i)) \cdot k_{X,T}((x,t), (\tilde{x}_j, \tilde{t}_j)) \tag{6}$$

where $k_{X,T}$ is the joint kernel across space and time and $A$ is a positive semi-definite matrix of dimension $\mathbb{R}^{m \times m}$. $k_{X,T}$ can be defined as the product of two Gaussian kernels $k_X$ and $k_T$ which are defined on space and time domain respectively i.e $k_{X,T}((x,t), (x',t')) = k_X(x,x') \cdot k_T(t,t')$. We represent $\phi_X(x) = k_X(x,\cdot) \in \mathcal{H}_X$, $\phi_T(t) = k_T(t,\cdot) \in \mathcal{H}_T$ and $\phi_{X,T}(x,t) = \phi_X(x) \otimes \phi_T(t)$. Here, we assume that $k_X$ and $k_T$ are both Gaussian kernels and hence the RKHS $\mathcal{H}_X$ and $\mathcal{H}_T$ correspond to the RKHS of the Gaussian kernel. For simplicity, we keep the kernel bandwidth parameter $\eta$ the same for $k_X$ and $k_T$. It is clear that the joint kernel $k_{X,T}$ is also Gaussian with the kernel bandwidth parameter equal to $\eta$. Given $M = \sum_{i,j} A_{ij} \phi_{X,T}(x_i, t_i) \phi_{X,T}(x_j, t_j)^\top$, $p(x,t)$ can also be represented with function $f((x,t), A, X, \eta)$ as,

$$\tilde{p}(x,t) = \underbrace{\phi_{X,T}(x,t)^\top M \phi_{X,T}(x,t)}_{:=f((x,t);M,\phi_{X,T})} = \underbrace{\psi(x,t)^\top A \psi(x,t)}_{:=f((x,t),A,\tilde{X},\eta)}, \tag{7}$$

where $\psi(x,t) \in \mathbb{R}^n$, $[\psi(x,t)]_i = k_X(x,\tilde{x}_i) \cdot k_T(t,\tilde{t}_i)$ and $\tilde{X} \in \mathbb{R}^{m \times d+1}$ is the concatenated data matrix of space and time for $m$ pair of samples $(x_i, t_i)$ for $i \in \{1, \cdots, m\}$.

We can now easily optimize an associate loss function with the FPE to obtain the unknown PSD matrix $A$. Let us assume that there exists a positive semi-definite matrix $A$ for which the PSD model $\tilde{p}(x,t)$ approximates the true solution $p^\star(x,t)$ well. In that case, our goal is to obtain an upper bound on the following error objective:

$$\left\| \frac{\partial (p^\star - \tilde{p})(x,t)}{\partial t} + \sum_{i=1}^{d} \frac{\partial}{\partial x_i}(\mu_i(x,t)(p^\star - \tilde{p})(x,t)) - \sum_{i=1}^{d}\sum_{j=1}^{d} D_{ij} \frac{\partial^2}{\partial x_i \partial x_j}(p^\star - \tilde{p})(x,t) \right\|_{L^2(\tilde{x})}. \quad (8)$$

Our approximation guarantee works in two steps. In the first step, we find an infinite dimensional positive operator $M_\varepsilon : \mathcal{H}_X \otimes \mathcal{H}_T \to \mathcal{H}_X \otimes \mathcal{H}_T$ such that if we denote $\hat{p}(x,t) = \phi_{X,T}(x,t)^\top M_\varepsilon \phi_{X,T}(x,t)$ then for some $M_\varepsilon$, $\hat{p}$ is $\varepsilon$-approximation of the true density $p^\star$. Clearly, for a set of functions $\tilde{f}_1, \cdots, \tilde{f}_q \in \mathcal{H}_X \otimes \mathcal{H}_T$, $M_\varepsilon = \sum_{j=1}^{q} \tilde{f}_j \tilde{f}_j^\top$ is a positive operator from $\mathcal{H}_X \otimes \mathcal{H}_T \to \mathcal{H}_X \otimes \mathcal{H}_T$. Now, we carefully construct $\tilde{f}_j$ for $j \in \{1, \cdots, q\}$ in RKHS $\mathcal{H}_X \otimes \mathcal{H}_T$ such that approximation guarantee in theorem 1 is satisfied. Details of the construction is given in Appendix. We represent $(x,t)$ as a $d+1$ dimensional vector $\tilde{x}$ and a mollifier function $g$ (details are in the appendix) such that $g_v(x) = v^{-(d+1)}g(\tilde{x}/v)$. Then, we construct our $\tilde{f}_j = f_j \star g_v$ for $j \in \{1, 2, \cdots, q\}$. We prove the result in Theorem 1 with this construction. To prove our approximation error guarantee for the solution of FPE using PSD model, we assume the derivatives of $f_1, \cdots, f_q$ up to order 2 are bounded which is already covered in Assumption 1.

**Theorem 1.** *Let* $\beta > 2, q \in \mathbb{N}$. *Let* $f_1, \ldots, f_q \in W_2^\beta(\mathbb{R}^d) \cap L^\infty(\mathbb{R}^d)$ *and the function* $p^\star = \sum_{i=1}^{q} f_i^2$. *Let* $\varepsilon \in (0,1]$ *and let* $\eta \in \mathbb{R}_{++}^d$. *Let* $\phi_{X,T}$ *be the feature map of the Gaussian kernel with bandwidth* $\eta$ *and let* $\mathcal{H}_X \otimes \mathcal{H}_T$ *be the associated RKHS. Then there exists* $\mathsf{M}_\varepsilon \in \mathbb{S}_+(\mathcal{H}_X \otimes \mathcal{H}_T)$ *with* $\mathrm{rank}(\mathsf{M}_\varepsilon) \leq q$, *such that for the representation* $\hat{p}(x,t) = \phi_{X,T}^\top \mathsf{M}_\varepsilon \phi_{X,T}$, *following holds under assumption 2 on the coefficients of the FPE,*

$$\left\| \frac{\partial \hat{p}(x,t)}{\partial t} + \sum_{i=1}^{d} \frac{\partial}{\partial x_i}(\mu_i(x,t)\hat{p}(x,t)) - \sum_{i=1}^{d}\sum_{j=1}^{d} D_{ij} \frac{\partial^2}{\partial x_i \partial x_j}\hat{p}(x,t) \right\|_{L^2(\tilde{x})} = O(\varepsilon), \quad (9)$$

$$\text{and } \mathrm{tr}(\mathsf{M}_\varepsilon) \leq \hat{C}|\eta|^{1/2}\left(1 + \varepsilon^{\frac{2\beta}{\beta-2}}\exp\left(\frac{\tilde{C}'}{\eta_0}\varepsilon^{-\frac{2}{\beta-2}}\right)\right),$$

*where* $|\eta| = \det(\mathrm{diag}(\eta))$, *and* $\tilde{C}'$ *and* $\hat{C}$ *depend only on* $\beta, d, \|f_i\|_{W_2^\beta(\mathbb{R}^d)}, \|f_i\|_{L^\infty(\mathbb{R}^d)}, \left\|\frac{\partial f_i}{\partial x_j}\right\|_{L^2(\mathbb{R}^d)},$ *and* $\left\|\frac{\partial^2 f_i}{\partial x_j x_k}\right\|_{L^2(\mathbb{R}^d)}$.

Even though, from theorem 1, we know that there exists a positive operator $\mathsf{M}_\varepsilon : \mathcal{H}_X \otimes \mathcal{H}_T \to \mathcal{H}_X \otimes \mathcal{H}_T$ such that $\hat{p}(x,t) = \phi_{X,T}(x,t)^\top \mathsf{M}_\varepsilon \phi_{X,T}(x,t)$ approximate the solution $p^\star(x,t)$ of FPE from Theorem 1, it is not clear how to compute $\mathsf{M}_\varepsilon$ as we do not have access of $f_j$ for $j \in \{1, 2, \cdots, q\}$ beforehand. Hence, to get the final approximation error bound we need to further approximate $\hat{p}(x,t)$ with a PSD model of finite dimension $\tilde{p}(x,t)$ which can be computed if we have the access to a finite number of base points $(x_i, t_i)$ for $i \in \{1, \cdots, m\}$ in the domain $\tilde{\mathcal{X}}$. Let us now construct a matrix $A_m$ as follows. Consider $A_m = K_{(X,T)(X,T)}^{-1} \tilde{Z}M\tilde{Z}^\star K_{(X,T)(X,T)}^{-1}$, where $K_{(X,T)(X,T)}$ is a kernel matrix in $\mathbb{R}^{m \times m}$ such that $[K_{(X,T)(X,T)}]_{ij} = \phi_{X,T}(x_i,t_i)^\top \phi_{X,T}(x_j,t_j)$. Let us define the operator $\tilde{Z} : \mathcal{H}_X \otimes \mathcal{H}_T \to \mathbb{R}^m$ such that $\tilde{Z}u = (\psi(x_1,t_1)^\top u, \cdots, \psi(x_m,t_m)^\top u)$, and $\tilde{Z}^\star \alpha = \sum_{i=1}^{m} \psi(x_i,t_i)\alpha_i$. We further define a projection operator $\tilde{P} = \tilde{Z}^\star K_{(X,T)(X,T)}^{-1} \tilde{Z}$. Hence, we now define the approximation density $\tilde{p}(x,t)$ as

$$\tilde{p}(x,t) = \underbrace{\psi(x,t)^\top A_m \psi(x,t)}_{:=f((x,t),A_m,\tilde{X},\eta)} = \underbrace{\phi_{X,T}(x,t)^\top \tilde{P}M\tilde{P}\phi_{X,T}(x,t)}_{:=f((x,t);\tilde{P}M\tilde{P},\phi_{X,T})}.$$

In the main result of this section below, we show that $\tilde{p}(x,t)$ is a good approximation for $\hat{p}(x,t)$ and hence for the true solution $p^\star(x,t)$.

**Theorem 2.** *Let* $p^\star$ *satisfy Assumptions 1 and 2. Let* $\varepsilon > 0$ *and* $\beta > 2$. *There exists a Gaussian PSD model of dimension* $m \in \mathbb{N}$, *i.e.,* $\tilde{p}(x,t) = f((x,t),A_m,X,\eta)$, *with* $A_m \in \mathbb{S}_+^m$ *and* $X \in \mathbb{R}^{m \times d+1}$

*and $\eta_m \in \mathbb{R}^d_{++}$, such that with $m = O(\varepsilon^{-(d+1)/(\beta-2)}(\log \frac{1}{\varepsilon})^{(d+1)/2})$, we have,*

$$\left\| \frac{\partial \tilde{p}(x,t)}{\partial t} + \sum_{i=1}^{d} \frac{\partial}{\partial x_i}(\mu_i(x,t)\tilde{p}(x,t)) - \sum_{i=1}^{d}\sum_{j=1}^{d} D_{ij} \frac{\partial^2}{\partial x_i \partial x_j}\tilde{p}(x,t) \right\|_{L^2(\tilde{\mathcal{X}})} = O(\varepsilon). \tag{10}$$

Theorem 2 indicates that there is a positive semi-definite matrix $A_m \in \mathbb{R}^{m \times m}$ such that $\tilde{p}(x,t) = \psi(x,t)^\top A_m \psi(x,t)$ is a good approximation of $p^\star(x,t)$. Next, we would see the usefulness of the above result in Theorem 2 in sampling from the solution of an FPE given the drift term $\mu$ satisfies the regularity condition as in [BRS16, Theorem 1.1] and $\mu \in W_2^\beta(\tilde{\mathcal{X}})$. In this case, the drift function $\mu$ can be approximated by a function $\hat{\mu}$ in a Gaussian RKHS, and the approximation guarantees are due to interpolation result in Lemma 4. The following result is the direct consequence of results in the Theorem 2 and Proposition 2.

**Efficient sampling method**   Let $\hat{\mu} \in \mathcal{H}_\eta$ be the approximation of $\mu$ such that $\|\mu - \hat{\mu}\|_{L^2(\tilde{\mathcal{X}})} = O(\varepsilon)$, by Gaussian kernel approximation. Let us apply the construction of Thm.2 to $\hat{\mu}$.

**Step 1:** Find the best model $\hat{A}_m$ by minimizing the l.h.s. of equation (12) over the set of $m$-dimensional Gaussian PSD models. Note that in that case, since everything is a linear combination of Gaussians (or products of Gaussians), we can compute the $L^2$ distance in closed form and minimize it over the positive semidefinite cone of $m$-dimensional matrices, obtaining $\hat{A}_\gamma$ exactly.

**Step 2:** Denote the resulting PSD model by $\hat{p}_\gamma(x,t) = f(x,t; \hat{A}_\gamma, \eta)$. At any time $t$ of interest we sample from the probability $\hat{p}_\gamma(\cdot, t)$ via the sampling algorithm in [MFBR22].

**Corollary 1** (Error of sampling). *Assume that $\mu \in W_2^\beta(\tilde{\mathcal{X}})$ satisfies the regularity condition as in [BRS16, Theorem 1.1]. For any time $t$, the algorithm described above provides i.i.d. samples from a probability $\tilde{p}_t$ that satisfies in expectation $\mathbb{E}_t \, \mathbb{W}_1(p_t^\star, \tilde{p}_t)^2 = O(\varepsilon^2)$, where $p_t^\star := p^\star(\cdot, t)$ is the probability at time $t$ of a stochastic process with drift $\mu$ and diffusion $D$. Moreover, the cost of the algorithm is $O(m^2 d \log(d/\varepsilon))$ where $m = O(\varepsilon^{-d/\beta-2}(\log(1/\varepsilon))^d)$.*

This result is proven in the appendix after theorem 2. It turns out that the parameter of the PSD model can be learned by a simple semi-definite program [RMFB20] which has a computational complexity of $O(m^{3.5} \log(1/\varepsilon))$ where $m$ is the dimension of the model.

**Remark.** *It is evident from the aforementioned result that our method excels in sampling from the Stochastic Differential Equation (SDE) while ensuring the particle density adheres to Assumption 1 within a bounded domain. In contrast, traditional sampling methods such as Langevin struggle to sample from this extensive class due to the absence of dissipativity, log-sobolev, and Poincaré-like conditions.*

## 4.2   Approximating Solution of Fractional Fokker-Planck Equation

In this section, our aim is to approximate the solution $p^\star(x,t)$ of a fractional Fokker-Planck equation, similar to the previous section. The equation replaces the Laplacian operator with a fractional Laplacian operator while keeping other terms unchanged. The fractional Laplacian operator, being non-local, is generally challenging to compute analytically. Therefore, approximating the fractional operator on a function has become a separate research direction. Below, we demonstrate that utilizing a PSD model-based representation is a natural choice for approximating the fractional operator acting on probability density. We present results for two cases: (i) a Gaussian kernel with the bandwidth parameter $\eta$, and (ii) a shift-invariant kernel represented using Bochner's theorem [Rud17].

**Theorem 3.** *Consider probability density represented by a PSD model as $p(x) = f(x; A, X, \eta) = \sum_{i,j=1}^{m} A_{ij} k(x_i, x) k(x_j, x)$ for some PSD matrix $A$, where $X \in \mathbb{R}^{m \times d}$ is the data matrix whose $i$-th row represents sample $x_i \in \mathbb{R}^d$ for $i \in \{1, \cdots, m\}$, then*
*1. if the kernel $k$ is Gaussian with bandwidth parameter $\eta$, we have*

$$(-\Delta)^s p(x) = C \sum_{i,j=1}^{m} A_{ij} e^{-\eta(\|x_i\|^2 + \|x_j\|^2)} \mathbb{E}_{\xi \sim \mathcal{N}(\mu, \Sigma)}[\|\xi\|^{2s} e^{-j\xi^\top x}]$$

*where $\mu = 2j\eta(x_i + x_j)$, $\Sigma = 4\eta I$ and $C$ can be computed in the closed form.*
*2. if the kernel $k$ have the bochner's theorem representation, i.e. $k(x - y) = \int_{\mathbb{R}^d} q(\omega) e^{j\omega^\top(x-y)} \, d\omega$ where $q$ is a probability density, then*

$$(-\Delta)^s p(x) = \sum_{i,j=1}^{m} A_{ij} \, \mathbb{E}[\|\omega_\ell + \omega_k\|^{2s} e^{j\omega_\ell^\top(x_i+x)} e^{j\omega_k^\top(x_j+x)}],$$

*where expectations are over $\omega_\ell, \omega_k \sim q(\cdot)$.*

We utilize the Fourier multiplier-based definition of the fractional Laplacian operator in Theorem 3. In both cases discussed in Theorem 3, empirical estimation of $(-\Delta)^s p(x)$, obtained from finite samples (empirical average), is sufficient for practical purposes. Shifting our focus from the representation of a non-local operator acting on a probability density, we now turn to the approximation result for the solution of a fractional Fokker-Planck equation in this section. We assume that the optimal solution $p^\star(x, t)$ can be expressed as a sum of squares of $q$ functions, as stated in assumption 1. To establish the approximation results for the fractional Fokker-Planck equation, we need to impose an additional mild assumption on $f_j$ for $j \in 1, \cdots, q$, in addition to assumption 1, which we state next.

**Assumption 3.** *Let $\mathcal{F}[f](\cdot)$ denotes the Fourier transform of a function $f$, then $\mathcal{F}[f_i](\cdot) \in L^1(\tilde{\mathcal{X}}) \cap L^\infty(\tilde{\mathcal{X}})$, and $\mathcal{F}[(-\Delta)^s f_i](\cdot) \in L^1(\tilde{\mathcal{X}})$ for $i \in \{1, \cdots q\}$.*

Similar to the previous section, it requires two steps to obtain the approximation error bound. By similar construction as in section 4.1, we define an infinite dimensional positive operator $M_\varepsilon$ : $\mathcal{H}_X \otimes \mathcal{H}_T \rightarrow \mathcal{H}_X \otimes \mathcal{H}_T$ and based on this positive operator, we have a probability density $\hat{p}$ as $\hat{p}(x, t) = \phi_{X,T}(x, t)^\top M_\varepsilon \phi_{X,T}(x, t)$ which is $O(\varepsilon)$ approximation of $p^\star(x, t)$. The result is stated below in Theorem 4 and proven in Appendix C.3.

**Theorem 4.** *Let $\beta > 2, q \in \mathbb{N}$. Let $f_1, \ldots, f_q$ satisfy assumptions 1 and 3 and the function $p^\star = \sum_{i=1}^q f_i^2$. Let $\varepsilon \in (0, 1]$ and let $\eta \in \mathbb{R}_{++}^d$. Let $\phi_{X,T}$ be the feature map of the Gaussian kernel with bandwidth $\eta$ and let $\mathcal{H}_X \otimes \mathcal{H}_T$ be the associated RKHS. Then there exists $M_\varepsilon \in \mathbb{S}_+(\mathcal{H}_X \otimes \mathcal{H}_T)$ with $\mathrm{rank}(M_\varepsilon) \leq q$, such that for the representation $\hat{p}(x, t) = \phi_{X,T}^\top M_\varepsilon \phi_{X,T}$, following holds under assumption 2 on the coefficients of the fractional FPE,*

$$\left\| \frac{\partial \hat{p}(x, t)}{\partial t} + \sum_{i=1}^d \frac{\partial}{\partial x_i}(\mu_i(x, t)\hat{p}(x, t)) + (-\Delta)^s \hat{p}(x, t) \right\|_{L^2(\tilde{\mathcal{X}})} = O(\varepsilon), \tag{11}$$

$$and \ \ \mathrm{tr}(M_\varepsilon) \leq \hat{C}|\eta|^{1/2} \left( 1 + \varepsilon^{\frac{2\beta}{\beta-2s}} \exp\left( \frac{\tilde{C}'}{\eta_0} \varepsilon^{-\frac{2}{\beta-2s}} \right) \right),$$

*where $|\eta| = \det(\mathrm{diag}(\eta))$, and $\tilde{C}$ and $\hat{C}$ depend only on $\beta, d$ and properties of $f_1, \cdots, f_q$.*

Major difficulty in proving the result in Theorem 4 arises from the challenge of effectively managing the fractional Laplacian term. To tackle this issue, we utilize the Fourier-based representation to gain control over the error associated with the fractional Laplacian. The details can be found in Lemma7, presented in Appendix C.1. As a next step, we would follow a similar procedure as in section 4.1 to obtain the final approximation bound. In the second part of the proof, we utilize Gagliardo–Nirenberg inequality [MP18] for fractional Laplacian to obtain the final approximation guarantee. In the next result, we show that for a particular choice of a PSD matrix $A_m$, $\tilde{p}(x, t) = \psi(x, t)^\top A_m \psi(x, t) = \phi_{X,T}(x, t)^\top \tilde{P} M \tilde{P} \phi_{X,T}(x, t)$ is a good approximation for the true solution $p^\star(x, t)$ of the fractional Fokker-Planck equation.

**Theorem 5.** *Let $p^\star$ satisfy Assumptions 1, 2, and 3. Let $\varepsilon > 0$ and $\beta > 2$. There exists a Gaussian PSD model of dimension $m \in \mathbb{N}$, i.e., $\tilde{p}(x, t) = f((x, t), A_m, X, \phi_{X,T})$, with $A_m \in \mathbb{S}_+^m$ and $X \in \mathbb{R}^{m \times d+1}$ and $\eta_m \in \mathbb{R}_{++}^d$, such that with $m = O(\varepsilon^{-(d+1)/(\beta-2s)}(\log \frac{1}{\varepsilon})^{(d+1)/2})$, we have*

$$\left\| \frac{\partial \tilde{p}(x, t)}{\partial t} + \sum_{i=1}^d \frac{\partial}{\partial x_i}(\mu_i(x, t)\tilde{p}(x, t)) + (-\Delta)^s \tilde{p}(x, t) \right\|_{L^2(\tilde{\mathcal{X}})} \leq O(\varepsilon). \tag{12}$$

Similar results as in Corollary 1 can be obtained for the case fractional Fokker-Planck equations as well. We can directly utilize results from [RZGŞ23] to bound the Wasserstein metric between the solution of two fractional FPE whose drift terms are close in $L^\infty$ metric. However, to find a PSD matrix $A_m$ from finite samples does require two separate sampling procedures which would contribute to the estimation error, (i) sampling to estimate the mean in the estimation of the fractional Laplacian and (ii) sampling of a finite number of data points from the data generating distribution.

**Remark.** *As far as we know, Theorem 5 represents the first proof of approximation error bounds for the solution of the Fractional Fokker-Planck Equation when the approximated solution is a density function. This significant result allows for the utilization of the algorithm presented in [MFBR22] to sample from the approximate solution.*

# 5  Conclusion

In this paper, we study the approximation properties of PSD models in modeling the solutions of FPEs and fractional FPEs. For the FPE, we show that a PSD model of size $m = \tilde{O}(\varepsilon^{-(d+1)/(\beta-2)})$ can approximate the true solution density up to order $\varepsilon$ in $L^2$ metric where $\beta$ is the order of smoothness of the true solution. Furthermore, we extend our result to the fractional FPEs to show that the required model size to achieve $\varepsilon$-approximation error to the solution density is $m = \tilde{O}(\varepsilon^{-(d+1)/(\beta-2s)})$ where $s$ is the fractional order of the fractional Laplacian. In the process, we also show that PSD model-based representations for probability densities allow an easy way to approximate the fractional Laplacian operator (non-local operator) acting on probability density. As a future research direction, we would like to investigate and obtain a finite sample bound on the estimation error under regularity conditions on the drift term $\mu$.

## Acknowledgement

Anant Raj is supported by the a Marie Sklodowska-Curie Fellowship (project NN-OVEROPT 101030817). Umut Şimşekli's research is supported by the French government under management of Agence Nationale de la Recherche as part of the "Investissements d'avenir" program, reference ANR-19-P3IA-0001 (PRAIRIE 3IA Institute) and the European Research Council Starting Grant DYNASTY – 101039676. Alessandro Rudi acknowleges support of the French government under management of Agence Nationale de la Recherche as part of the "Investissements d'avenir" program, reference ANR-19-P3IA-0001 (PRAIRIE 3IA Institute)and support of the European Research Council (grant REAL 947908).

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

# Appendix

## A   Existence of an appropriate PSD matrix $M_\varepsilon$ (Fokker-Planck Equation)

### A.1   Useful Results

Consider the mollifier function $g$ defined as in [RC21].

$$g(x) = \frac{2^{-d/2}}{V_d} \|x\|^{-d} J_{d/2}(2\pi\|x\|) J_{d/2}(4\pi\|x\|), \tag{13}$$

where $J_{d/2}$ is the Bessel function of the first kind of order $d/2$ and $V_d = \int_{\|x\|\leq 1} dx = \frac{\pi^{d/2}}{\Gamma(d/2+1)}$. $g_v(x) = v^{-d} g(x/v)$.

**Lemma 1** ([RC21], Lemma D.3)**.** *The function $g$ defined above satisfies $g \in L^1(\mathbb{R}^d) \cap L^2(\mathbb{R}^d)$ and $\int g(x) dx = 1$. Moreover, for any $\omega \in \mathbb{R}^d$, we have*

$$\mathbf{1}_{\{\|\omega\|<1\}}(\omega) \leq \mathcal{F}[g](\omega) \leq \mathbf{1}_{\{\|\omega\|\leq 3\}}(\omega).$$

The following proposition provides a useful characterization of the space $W_2^\beta(\mathbb{R}^d)$

**Proposition 4** (Characterization of the Sobolev space $W_2^k(\mathbb{R}^d)$, [Wen04])**.** *Let $k \in \mathbb{N}$. The norm of the Sobolev space $\|\cdot\|_{W_2^k(\mathbb{R}^d)}$ is equivalent to the following norm*

$$\|f\|'^2_{W_2^k(\mathbb{R}^d)} = \int_{\mathbb{R}^d} |\mathcal{F}[f](\omega)|^2 (1+\|\omega\|^2)^k \, d\omega, \quad \forall f \in L^2(\mathbb{R}^d) \tag{14}$$

*and satisfies*

$$\frac{1}{(2\pi)^{2k}} \|f\|^2_{W_2^k(\mathbb{R}^d)} \leq \|f\|'_{W_2^k(\mathbb{R}^d)} \leq 2^{2k} \|f\|^2_{W_2^k(\mathbb{R}^d)}, \quad \forall f \in L^2(\mathbb{R}^d). \tag{15}$$

*Moreover, when $k > d/2$, then $W_2^k(\mathbb{R}^d)$ is a reproducing kernel Hilbert space.*

**Lemma 2.** *Consider the definition of $g_v(x) = v^{-d} g(x/v)$ where $g$ is defined in equation (13). Let $\beta > 2, q \in \mathbb{N}$. Let $f_1, \ldots, f_q \in W_2^\beta(\mathbb{R}^d) \cap L^\infty(\mathbb{R}^d)$ and the function $p^\star = \sum_{i=1}^q f_i^2$. Let us assume that the Assumption 2 is also satisfied. Let $\varepsilon \in (0,1]$ and let $\eta \in \mathbb{R}^d_{++}$. Let $\phi_{X,T}$ be the feature map of the Gaussian kernel with bandwidth $\eta$ and let $\mathcal{H}_X \otimes \mathcal{H}_T$ be the associated RKHS. Then there exists $M_\varepsilon \in \mathbb{S}_+(\mathcal{H}_X \otimes \mathcal{H}_T)$ with $\mathrm{rank}(M_\varepsilon) \leq q$, such that for the representation $\hat{p}(x,t) = \phi_{X,T}^\top M_\varepsilon \phi_{X,T}$, following holds for $v > 0$ and $k \in \{1, \cdots d\}$,*

$$\left\| \frac{\partial \hat{p}(x,t)}{\partial t} - \frac{\partial p^\star(x,t)}{\partial t} \right\|_{L^1(\mathbb{R}^d)} \leq 8\pi(2v)^{(\beta-1)} \sum_{i=1}^q \|f_i\|_{W_2^\beta(\mathbb{R}^d)} \|f_i\|_{L_2(\mathbb{R}^d)}$$

$$+ 2(2v)^\beta \sum_{i=1}^q \|f_i\|_{W_2^\beta(\mathbb{R}^d)} \left\| \frac{\partial f_i}{\partial t} \right\|_{L_2(\mathbb{R}^d)} \|g\|_{L_1(\mathbb{R}^d)}, \tag{16}$$

$$\left\| \frac{\partial p^\star(x,t)}{\partial t} - \frac{\partial \hat{p}(x,t)}{\partial t} \right\|_{L^2(\mathbb{R}^d)} \leq 8\pi(2v)^{(\beta-1)} \sum_{i=1}^q \|f_i\|_{W_2^\beta(\mathbb{R}^d)} \|f_i\|_{L^\infty(\mathbb{R}^d)}$$

$$+ 2(2v)^\beta \sum_{i=1}^q \|f_i\|_{W_2^\beta(\mathbb{R}^d)} \left\| \frac{\partial f_i}{\partial t} \right\|_{L^\infty(\mathbb{R}^d)} \|g\|_{L_1(\mathbb{R}^d)} \tag{17}$$

$$\left\| \frac{\partial p^\star(x,t)}{\partial x_k} - \frac{\partial \hat{p}(x,t)}{\partial x_k} \right\|_{L^1(\mathbb{R}^d)} \leq 8\pi(2v)^{(\beta-1)} \sum_{i=1}^q \|f_i\|_{W_2^\beta(\mathbb{R}^d)} \|f_i\|_{L_2(\mathbb{R}^d)}$$

$$+ 2(2v)^\beta \sum_{i=1}^q \|f_i\|_{W_2^\beta(\mathbb{R}^d)} \left\| \frac{\partial f_i}{\partial x_k} \right\|_{L_2(\mathbb{R}^d)} \|g\|_{L_1(\mathbb{R}^d)} \tag{18}$$

$$\left\| \frac{\partial p^\star(x,t)}{\partial x_k} - \frac{\partial \hat{p}(x,t)}{\partial x_k} \right\|_{L^2(\mathbb{R}^d)} \leq 8\pi(2v)^{(\beta-1)} \sum_{i=1}^q \|f_i\|_{W_2^\beta(\mathbb{R}^d)} \|f_i\|_{L^\infty(\mathbb{R}^d)} \tag{19}$$

$$+ 2(2v)^\beta \sum_{i=1}^{q} \|f_i\|_{W_2^\beta(\mathbb{R}^d)} \left\| \frac{\partial f_i}{\partial x_k} \right\|_{L^\infty(\mathbb{R}^d)} \|g\|_{L_1(\mathbb{R}^d)}. \tag{20}$$

*Proof.* We have $g$ defined as in [RC21].

$$g_v(x) = \frac{2^{-d/2}}{V_d} \|x\|^{-d} J_{d/2}(2\pi \|x\|) J_{d/2}(4\pi \|x\|), \tag{21}$$

where $J_{d/2}$ is the Bessel function of the first kind of order $d/2$ and $V_d = \int_{\|x\| \leq 1} dx = \frac{\pi^{d/2}}{\Gamma(d/2+1)}$.
$g_v(x) = v^{-d} g(x/v)$. We use result of lemma 1 to prove the statement. For a function $f \in W_2^\beta(\mathbb{R}^d)$ ,

$$\left\| \frac{\partial}{\partial x_i} f - \frac{\partial}{\partial x_i}(f \star g_v) \right\|_{L_2(\mathbb{R}^d)}^2 = \left\| \mathcal{F}\left[ \frac{\partial}{\partial x_i} f \right] - \mathcal{F}\left[ \frac{\partial}{\partial x_i}(f \star g_v) \right] \right\|_{L_2(\mathbb{R}^d)}^2$$

$$= \|2\pi\iota\omega_i \mathcal{F}[f] - 2\pi\iota\omega_i \mathcal{F}[f \star g_v]\|_{L_2(\mathbb{R}^d)} = \|2\pi\iota\omega_i \mathcal{F}[f](1 - \mathcal{F}[g_v])\|_{L_2(\mathbb{R}^d)}^2$$

$$= 4\pi^2 \int_{\mathbb{R}^d} \omega_i^2 |\mathcal{F}[f](\omega)|^2 |1 - \mathcal{F}[g](v\omega)|^2 \, d\omega \leq 4\pi^2 \int_{t\|\omega\| \geq 1} \omega_i^2 |\mathcal{F}[f](\omega)|^2 \, d\omega$$

$$= 4\pi^2 \int_{v\|\omega\| \geq 1} \omega_i^2 (1 + \|\omega\|^2)^{-\beta} (1 + \|\omega\|^2)^{\beta} |\mathcal{F}[f](\omega)|^2 \, d\omega$$

$$\leq \sup_{v\|\omega\| \geq 1} \frac{4\pi^2 \omega_i^2}{(1 + \|\omega\|^2)^\beta} \int (1 + \|\omega\|^2)^\beta |\mathcal{F}[f](\omega)|^2 \, d\omega$$

$$\leq \sup_{v\|\omega\| \geq 1} \frac{4\pi^2 \omega_i^2}{(1 + \|\omega\|^2)^\beta} 2^{2\beta} \|f\|_{W_2^\beta(\mathbb{R}^d)}^2 \leq \sup_{v\|\omega\| \geq 1} \frac{4\pi^2}{(1 + \|\omega\|^2)^{\beta-1}} 2^{2\beta} \|f\|_{W_2^\beta(\mathbb{R}^d)}^2$$

$$\leq \frac{16\pi^2 (2v)^{2(\beta-1)}}{(1 + v^2)^{\beta-1}} \|f\|_{W_2^\beta(\mathbb{R}^d)}^2$$

Hence,

$$\left\| \frac{\partial}{\partial x_i} f - \frac{\partial}{\partial x_i}(f \star g_v) \right\|_{L_2(\mathbb{R}^d)}^2 \leq \frac{16\pi^2 (2v)^{2(\beta-1)}}{(1 + v^2)^{\beta-1}} \|f\|_{W_2^\beta(\mathbb{R}^d)}^2 \tag{22}$$

The same result will also hold for $\frac{\partial}{\partial t}$ without loss of generality. Hence,

$$\left\| \frac{\partial}{\partial t} f - \frac{\partial}{\partial t}(f \star g_v) \right\|_{L_2(\mathbb{R}^d)}^2 \leq \frac{16\pi^2 (2v)^{2(\beta-1)}}{(1 + v^2)^{\beta-1}} \|f\|_{W_2^\beta(\mathbb{R}^d)}^2 \tag{23}$$

We now bound $\left\| \frac{\partial p^\star(x,t)}{\partial t} - \frac{\partial \hat{p}(x,t)}{\partial t} \right\|_{L^r(\mathbb{R}^d)}$ and $\left\| \frac{\partial p^\star(x,t)}{\partial x_i} - \frac{\partial \hat{p}(x,t)}{\partial x_i} \right\|_{L^r(\mathbb{R}^d)}$.

As in [RC21], let us denote, $f_i \star g_v$ as $f_{i,v}$. We consider

$$M_\varepsilon = \sum_{i=1}^{q} f_{i,v} f_{i,v}^\top.$$

Hence,

$$\hat{p}(x,t) = f(x,t) = \phi_{X,T}(x,t)^\top M_\varepsilon \phi_{X,T}(x,t) = \sum_{i=1}^{q} f_{i,v}^2(x,t).$$

This gives,

$$\frac{\partial}{\partial x_i} p^\star - \frac{\partial}{\partial x_i} \hat{p} = 2\sum_{i=1}^{q} \left[ f_i \frac{\partial f_i}{\partial x_i} - f_{i,v} \frac{\partial f_{i,v}}{\partial x_i} \right] = 2\sum_{i=1}^{q} \left[ f_i \frac{\partial f_i}{\partial x_i} - f_i \frac{\partial f_{i,v}}{\partial x_i} + f_i \frac{\partial f_{i,v}}{\partial x_i} - f_{i,v} \frac{\partial f_{i,v}}{\partial x_i} \right]$$

$$= 2\sum_{i=1}^{q} \left[ f_i \left( \frac{\partial f_i}{\partial x_i} - \frac{\partial f_{i,v}}{\partial x_i} \right) + (f_i - f_{i,v}) \frac{\partial f_{i,v}}{\partial x_i} \right]$$

Hence, we have

$$\left\|\frac{\partial}{\partial x_i}p^\star - \frac{\partial}{\partial x_i}\hat{p}\right\|_{L_1(\mathbb{R}^d)} \leq 2\sum_{i=1}^q \left\|\frac{\partial f_i}{\partial x_i} - \frac{\partial f_{i,v}}{\partial x_i}\right\|_{L_2(\mathbb{R}^d)} \|f_i\|_{L_2(\mathbb{R}^d)} + 2\sum_{i=1}^q \|f_i - f_{i,v}\|_{L_2(\mathbb{R}^d)} \left\|\frac{\partial f_{i,v}}{\partial x_i}\right\|_{L_2(\mathbb{R}^d)}.$$

(24)

Similarly,

$$\left\|\frac{\partial}{\partial t}p^\star - \frac{\partial}{\partial t}\hat{p}\right\|_{L_1(\mathbb{R}^d)} \leq 2\sum_{i=1}^q \left\|\frac{\partial f_i}{\partial t} - \frac{\partial f_{i,v}}{\partial t}\right\|_{L_2(\mathbb{R}^d)} \|f_i\|_{L_2(\mathbb{R}^d)} + 2\sum_{i=1}^q \|f_i - f_{i,v}\|_{L_2(\mathbb{R}^d)} \left\|\frac{\partial f_{i,v}}{\partial t}\right\|_{L_2(\mathbb{R}^d)}.$$

(25)

Hence,

$$\left\|\frac{\partial p^\star(x,t)}{\partial t} - \frac{\partial \hat{p}(x,t)}{\partial t}\right\|_{L^1(\mathbb{R}^d)} \leq 2\sum_{i=1}^q \left\|\frac{\partial f_i}{\partial t} - \frac{\partial f_{i,v}}{\partial t}\right\|_{L^2(\mathbb{R}^d)} \|f_i\|_{L^2(\mathbb{R}^d)} + 2\sum_{i=1}^q \|f_i - f_{i,v}\|_{L^2(\mathbb{R}^d)} \left\|\frac{\partial f_{i,v}}{\partial t}\right\|_{L^2(\mathbb{R}^d)}$$

$$\leq \frac{8\pi(2v)^{(\beta-1)}}{(1+v^2)^{\frac{\beta-1}{2}}} \sum_{i=1}^q \|f_i\|_{W_2^\beta(\mathbb{R}^d)} \|f_i\|_{L_2(\mathbb{R}^d)} + 2(2v)^\beta \sum_{i=1}^q \|f_i\|_{W_2^\beta(\mathbb{R}^d)} \left\|\frac{\partial f_{i,v}}{\partial t}\right\|_{L_2(\mathbb{R}^d)}$$

$$\leq 8\pi(2v)^{(\beta-1)} \sum_{i=1}^q \|f_i\|_{W_2^\beta(\mathbb{R}^d)} \|f_i\|_{L_2(\mathbb{R}^d)} + 2(2v)^\beta \sum_{i=1}^q \|f_i\|_{W_2^\beta(\mathbb{R}^d)} \left\|\frac{\partial f_i}{\partial t} \star g_v\right\|_{L_2(\mathbb{R}^d)}$$

$$\leq 8\pi(2v)^{(\beta-1)} \sum_{i=1}^q \|f_i\|_{W_2^\beta(\mathbb{R}^d)} \|f_i\|_{L_2(\mathbb{R}^d)} + 2(2v)^\beta \sum_{i=1}^q \|f_i\|_{W_2^\beta(\mathbb{R}^d)} \left\|\frac{\partial f_i}{\partial t}\right\|_{L_2(\mathbb{R}^d)} \|g_v\|_{L_1(\mathbb{R}^d)}$$

$$= 8\pi(2v)^{(\beta-1)} \sum_{i=1}^q \|f_i\|_{W_2^\beta(\mathbb{R}^d)} \|f_i\|_{L_2(\mathbb{R}^d)} + 2(2v)^\beta \sum_{i=1}^q \|f_i\|_{W_2^\beta(\mathbb{R}^d)} \left\|\frac{\partial f_i}{\partial t}\right\|_{L_2(\mathbb{R}^d)} \|g\|_{L_1(\mathbb{R}^d)}.$$

If we choose

$$v = \min\left(\left(\frac{\varepsilon}{2^{\beta-1}16\pi C_1}\right)^{\frac{1}{\beta-1}}, \left(\frac{\varepsilon}{2^{\beta+1}C_2}\right)^{\frac{1}{\beta}}\right)$$

(26)

where $C_1 = \sum_{i=1}^q \|f_i\|_{W_2^\beta(\mathbb{R}^d)} \|f_i\|_{L_2(\mathbb{R}^d)}$ and $C_2 = \sum_{i=1}^q \|f_i\|_{W_2^\beta(\mathbb{R}^d)} \left\|\frac{\partial f_i}{\partial t}\right\|_{L_2(\mathbb{R}^d)} \|g\|_{L_1(\mathbb{R}^d)}$.
Hence,

$$\left\|\frac{\partial p^\star(x,t)}{\partial t} - \frac{\partial \hat{p}(x,t)}{\partial t}\right\|_{L^1(\mathbb{R}^d)} \leq \varepsilon.$$

Similarly,

$$\left\|\frac{\partial p^\star(x,t)}{\partial t} - \frac{\partial \hat{p}(x,t)}{\partial t}\right\|_{L^2(\mathbb{R}^d)} = 2\sum_{i=1}^q \left\|\frac{\partial f_i}{\partial t} - \frac{\partial f_{i,v}}{\partial t}\right\|_{L^2(\mathbb{R}^d)} \|f_i\|_{L^\infty(\mathbb{R}^d)} + 2\sum_{i=1}^q \|f_i - f_{i,v}\|_{L^2(\mathbb{R}^d)} \left\|\frac{\partial f_{i,v}}{\partial t}\right\|_{L^\infty(\mathbb{R}^d)}.$$

Now,

$$\left\|\frac{\partial f_{i,v}}{\partial t}\right\|_{L^\infty(\mathbb{R}^d)} = \left\|\frac{\partial f_i}{\partial t} \star g_v\right\|_{L^\infty(\mathbb{R}^d)} \leq \left\|\frac{\partial f_i}{\partial t}\right\|_{L^\infty(\mathbb{R}^d)} \|g_v\|_{L^1(\mathbb{R}^d)} = \left\|\frac{\partial f_i}{\partial t}\right\|_{L^\infty(\mathbb{R}^d)} \|g\|_{L^1(\mathbb{R}^d)}.$$

Hence,

$$\left\|\frac{\partial p^\star(x,t)}{\partial t} - \frac{\partial \hat{p}(x,t)}{\partial t}\right\|_{L^2(\mathbb{R}^d)} \leq 8\pi(2v)^{(\beta-1)} \sum_{i=1}^q \|f_i\|_{W_2^\beta(\mathbb{R}^d)} \|f_i\|_{L^\infty(\mathbb{R}^d)}$$

$$+ 2(2v)^\beta \sum_{i=1}^q \|f_i\|_{W_2^\beta(\mathbb{R}^d)} \left\|\frac{\partial f_i}{\partial t}\right\|_{L^\infty(\mathbb{R}^d)} \|g\|_{L^1(\mathbb{R}^d)}.$$

If we choose

$$v = \min\left(\left(\frac{\varepsilon}{2^{\beta-1}16\pi C_1}\right)^{\frac{1}{\beta-1}}, \left(\frac{\varepsilon}{2^{\beta+1}C_2}\right)^{\frac{1}{\beta}}\right)$$

where $C_1 = \sum_{i=1}^q \|f_i\|_{W_2^\beta(\mathbb{R}^d)} \|f_i\|_{L^\infty(\mathbb{R}^d)}$ and $C_2 = \sum_{i=1}^q \|f_i\|_{W_2^\beta(\mathbb{R}^d)} \left\|\frac{\partial f_i}{\partial t}\right\|_{L^\infty(\mathbb{R}^d)} \|g\|_{L_1(\mathbb{R}^d)}$. Hence,

$$\left\| \frac{\partial p^\star(x,t)}{\partial t} - \frac{\partial \hat{p}(x,t)}{\partial t} \right\|_{L^2(\mathbb{R}^d)} \le \varepsilon.$$

Similarly,

$$\left\| \frac{\partial p^\star(x,t)}{\partial x_k} - \frac{\partial \hat{p}(x,t)}{\partial x_k} \right\|_{L^1(\mathbb{R}^d)} \le 8\pi(2v)^{(\beta-1)} \sum_{i=1}^q \|f_i\|_{W_2^\beta(\mathbb{R}^d)} \|f_i\|_{L_2(\mathbb{R}^d)} \tag{27}$$
$$+ 2(2v)^\beta \sum_{i=1}^q \|f_i\|_{W_2^\beta(\mathbb{R}^d)} \left\|\frac{\partial f_i}{\partial x_k}\right\|_{L_2(\mathbb{R}^d)} \|g\|_{L_1(\mathbb{R}^d)}$$

and

$$\left\| \frac{\partial p^\star(x,t)}{\partial x_k} - \frac{\partial \hat{p}(x,t)}{\partial x_k} \right\|_{L^2(\mathbb{R}^d)} \le 8\pi(2v)^{(\beta-1)} \sum_{i=1}^q \|f_i\|_{W_2^\beta(\mathbb{R}^d)} \|f_i\|_{L^\infty(\mathbb{R}^d)}$$
$$+ 2(2v)^\beta \sum_{i=1}^q \|f_i\|_{W_2^\beta(\mathbb{R}^d)} \left\|\frac{\partial f_i}{\partial x_k}\right\|_{L^\infty(\mathbb{R}^d)} \|g\|_{L_1(\mathbb{R}^d)}.$$

$\square$

**Lemma 3.** *Consider the definition of $g_v(x) = v^{-d}g(x/v)$ where $g$ is defined in equation (13). Let $\beta > 2, q \in \mathbb{N}$. Let $f_1, \ldots, f_q \in W_2^\beta(\mathbb{R}^d) \cap L^\infty(\mathbb{R}^d)$ and the function $p^\star = \sum_{i=1}^q f_i^2$. Let us assume that the Assumption 2 is also satisfied. Let $\varepsilon \in (0,1]$ and let $\eta \in \mathbb{R}_{++}^d$. Let $\phi_{X,T}$ be the feature map of the Gaussian kernel with bandwidth $\eta$ and let $\mathcal{H}_X \otimes \mathcal{H}_T$ be the associated RKHS. Then there exists $\mathsf{M}_\varepsilon \in \mathbb{S}_+(\mathcal{H}_X \otimes \mathcal{H}_T)$ with $\mathrm{rank}(\mathsf{M}_\varepsilon) \le q$, such that for the representation $\hat{p}(x,t) = f(x,t) = \phi_{X,T}^\top \mathsf{M}_\varepsilon \phi_{X,T}$, following holds for all $i,j \in \{1, \cdots d\}.,$*

$$\left\| \frac{\partial^2}{\partial x_i \partial x_j} p^\star(x,t) - \frac{\partial^2}{\partial x_i \partial x_j} \hat{p}(x,t) \right\|_{L^1(\mathbb{R}^d)} \le \frac{16\pi(2v)^{(\beta-1)}}{(1+v^2)^{\frac{\beta-1}{2}}} (1 + \|g\|_{L^1(\mathbb{R}^d)}) \sum_{m=1}^q \|f\|_{W_2^\beta(\mathbb{R}^d)} \left\|\frac{\partial f_m}{\partial x_j}\right\|_{L^2(\mathbb{R}^d)}$$
$$+ \frac{32\pi^2(2v)^{(\beta-2)}}{(1+v^2)^{\frac{\beta-2}{2}}} \sum_{m=1}^q \|f\|_{W_2^\beta(\mathbb{R}^d)} \|f_m\|_{L^2(\mathbb{R}^d)} + 2(2v)^\beta \sum_{m=1}^q \|f_m\|_{W_2^\beta(\mathbb{R}^d)} \left\|\frac{\partial^2 f_m}{\partial x_i \partial x_j}\right\|_{L^2(\mathbb{R}^d)} \|g\|_{L^1(\mathbb{R}^d)},$$

*and*

$$\left\| \frac{\partial^2}{\partial x_i \partial x_j} p^\star(x,t) - \frac{\partial^2}{\partial x_i \partial x_j} \hat{p}(x,t) \right\|_{L^2(\mathbb{R}^d)} \le \frac{16\pi(2v)^{(\beta-1)}}{(1+v^2)^{\frac{\beta-1}{2}}} (1 + \|g\|_{L^1(\mathbb{R}^d)}) \sum_{m=1}^q \|f\|_{W_2^\beta(\mathbb{R}^d)} \left\|\frac{\partial f_m}{\partial x_j}\right\|_{L^\infty(\mathbb{R}^d)}$$
$$+ \frac{32\pi^2(2v)^{(\beta-2)}}{(1+v^2)^{\frac{\beta-2}{2}}} \sum_{m=1}^q \|f\|_{W_2^\beta(\mathbb{R}^d)} \|f_m\|_{L^\infty(\mathbb{R}^d)} + 2(2v)^\beta \sum_{m=1}^q \|f_m\|_{W_2^\beta(\mathbb{R}^d)} \left\|\frac{\partial^2 f_m}{\partial x_i \partial x_j}\right\|_{L^\infty(\mathbb{R}^d)} \|g\|_{L^1(\mathbb{R}^d)}.$$

*Proof.* We have $g$ defined as in [RC21].

$$g_v(x) = \frac{2^{-d/2}}{V_d} \|x\|^{-d} J_{d/2}(2\pi\|x\|) J_{d/2}(4\pi\|x\|), \tag{28}$$

where $J_{d/2}$ is the Bessel function of the first kind of order $d/2$ and $V_d = \int_{\|x\|\le 1} dx = \frac{\pi^{d/2}}{\Gamma(d/2+1)}$. $g_v(x) = v^{-d}g(x/v)$. We use result of lemma 1 to prove the statement. We have, for a function $f \in W_2^\beta(\mathbb{R}^d)$

$$\left\| \frac{\partial^2}{\partial x_i \partial x_j} f - \frac{\partial^2}{\partial x_i x_j} (f \star g_v) \right\|_{L_2(\mathbb{R}^d)}^2 = \left\| \mathcal{F}\left[ \frac{\partial^2}{\partial x_i \partial x_j} f \right] - \mathcal{F}\left[ \frac{\partial^2}{\partial x_i \partial x_j} (f \star g_v) \right] \right\|_{L_2(\mathbb{R}^d)}^2$$
$$= \left\| 4\pi^2 \omega_i \omega_j \mathcal{F}[f] - 4\pi^2 \omega_i \omega_j \mathcal{F}[f \star g_v] \right\|_{L_2(\mathbb{R}^d)}$$
$$= \left\| 4\pi^2 \omega_i \omega_j \mathcal{F}[f](1 - \mathcal{F}[g_v]) \right\|_{L_2(\mathbb{R}^d)}^2$$

$$= 16\pi^4 \int_{\mathbb{R}^d} \omega_i^2 \omega_j^2 |\mathcal{F}[f](\omega)|^2 |1 - \mathcal{F}[g](v\omega)|^2 \, d\omega$$

$$\leq 16\pi^4 \pi^2 \int_{t\|\omega\|\geq 1} \omega_i^2 \omega_j^2 |\mathcal{F}[f](\omega)|^2 \, d\omega$$

$$= 16\pi^4 \int_{v\|\omega\|\geq 1} \omega_i^2 \omega_j^2 (1 + \|\omega\|^2)^{-\beta} (1 + \|\omega\|^2)^{\beta} |\mathcal{F}[f](\omega)|^2 \, d\omega$$

$$\leq 16\pi^4 \sup_{v\|\omega\|\geq 1} \frac{\omega_i^2 \omega_j^2}{(1 + \|\omega\|^2)^{\beta}} \int (1 + \|\omega\|^2)^{\beta} |\mathcal{F}[f](\omega)|^2 \, d\omega$$

$$\leq 16\pi^4 \sup_{v\|\omega\|\geq 1} \frac{\omega_i^2 \omega_j^2}{(1 + \|\omega\|^2)^{\beta}} 2^{2\beta} \|f\|_{W_2^{\beta}(\mathbb{R}^d)}^2$$

$$\leq 16\pi^4 \sup_{v\|\omega\|\geq 1} \frac{1}{(1 + \|\omega\|^2)^{\beta-2}} 2^{2\beta} \|f\|_{W_2^{\beta}(\mathbb{R}^d)}^2$$

$$\leq \frac{256\pi^4 (2v)^{2(\beta-2)}}{(1 + v^2)^{\beta-2}} \|f\|_{W_2^{\beta}(\mathbb{R}^d)}^2.$$

As in [RC21], let us denote, $f_i \star g_v$ as $f_{i,v}$. We consider

$$M_\varepsilon = \sum_{i=1}^q f_{i,v} f_{i,v}^\top.$$

Hence,

$$\hat{p}(x,t) = f(x,t) = \phi_{X,T}(x,t)^\top M_\varepsilon \phi_{X,T}(x,t) = \sum_{i=1}^q f_{i,v}^2(x,t).$$

Now we have,

$$\left\| \frac{\partial^2}{\partial x_i \partial x_j} p^\star(x,t) - \frac{\partial^2}{\partial x_i \partial x_j} f(x,t) \right\|_{L^1(\mathbb{R}^d)} = \left\| 2 \sum_{m=1}^q \left[ \frac{\partial f_m}{\partial x_i} \frac{\partial f_m}{\partial x_j} - \frac{\partial f_{m,v}}{\partial x_i} \frac{\partial f_{m,v}}{\partial x_j} \right. \right.$$

$$\left. \left. + f_m \frac{\partial^2 f_m}{\partial x_i \partial x_j} - f_{m,v} \frac{\partial^2 f_{m,v}}{\partial x_i \partial x_j} \right] \right\|_{L^1(\mathbb{R}^d)}$$

$$\leq 2 \sum_{m=1}^q \left[ \left\| \frac{\partial f_m}{\partial x_i} - \frac{\partial f_{m,v}}{\partial x_i} \right\|_{L^2(\mathbb{R}^d)} \left\| \frac{\partial f_m}{\partial x_j} \right\|_{L^2(\mathbb{R}^d)} + \left\| \frac{\partial f_m}{\partial x_j} - \frac{\partial f_{m,v}}{\partial x_j} \right\|_{L^2(\mathbb{R}^d)} \left\| \frac{\partial f_{m,v}}{\partial x_j} \right\|_{L^2(\mathbb{R}^d)} \right]$$

$$+ 2 \sum_{m=1}^q \left[ \left\| \frac{\partial^2 f_m}{\partial x_i \partial x_j} - \frac{\partial^2 f_{m,v}}{\partial x_i \partial x_j} \right\|_{L^2(\mathbb{R}^d)} \|f_m\|_{L^2(\mathbb{R}^d)} + \|f_m - f_{m,v}\|_{L^2(\mathbb{R}^d)} \left\| \frac{\partial^2 f_{m,v}}{\partial x_i \partial x_j} \right\|_{L^2(\mathbb{R}^d)} \right]$$

Similarly,

$$\left\| \sum_{i=1}^d \sum_{j=1}^d \left[ \frac{\partial^2}{\partial x_i \partial x_j} (D_{ij} p^\star(x,t)) - \frac{\partial^2}{\partial x_i \partial x_j} (D_{ij} f(x,t)) \right] \right\|_{L_1(\mathbb{R}^d)}$$

$$\leq \underbrace{\sup_{i,j} D_{ij}}_{:=R_d} \sum_{i=1}^d \sum_{j=1}^d \left\| \frac{\partial^2}{\partial x_i \partial x_j} p^\star(x,t) - \frac{\partial^2}{\partial x_i \partial x_j} f(x,t) \right\|_{L_1(\mathbb{R}^d)}. \tag{29}$$

Now,

$$\left\| \frac{\partial^2}{\partial x_i \partial x_j} p^\star(x,t) - \frac{\partial^2}{\partial x_i \partial x_j} f(x,t) \right\|_{L^1(\mathbb{R}^d)}$$

$$\leq 2 \sum_{m=1}^q \left[ \left\| \frac{\partial f_m}{\partial x_i} - \frac{\partial f_{m,v}}{\partial x_i} \right\|_{L^2(\mathbb{R}^d)} \left\| \frac{\partial f_m}{\partial x_j} \right\|_{L^2(\mathbb{R}^d)} + \left\| \frac{\partial f_m}{\partial x_j} - \frac{\partial f_{m,v}}{\partial x_j} \right\|_{L^2(\mathbb{R}^d)} \left\| \frac{\partial f_{m,v}}{\partial x_j} \right\|_{L^2(\mathbb{R}^d)} \right]$$

$$+2\sum_{m=1}^{q}\left[\left\|\frac{\partial^2 f_m}{\partial x_i \partial x_j}-\frac{\partial^2 f_{m,v}}{\partial x_i \partial x_j}\right\|_{L^2(\mathbb{R}^d)}\|f_m\|_{L^2(\mathbb{R}^d)}+\|f_m-f_{m,v}\|_{L^2(\mathbb{R}^d)}\left\|\frac{\partial^2 f_{m,v}}{\partial x_i \partial x_j}\right\|_{L^2(\mathbb{R}^d)}\right]$$

Let's consider all the terms separately.

$$2\sum_{m=1}^{q}\left\|\frac{\partial f_m}{\partial x_i}-\frac{\partial f_{m,v}}{\partial x_i}\right\|_{L^2(\mathbb{R}^d)}\left\|\frac{\partial f_m}{\partial x_j}\right\|_{L^2(\mathbb{R}^d)} \leq \frac{16\pi(2v)^{(\beta-1)}}{(1+v^2)^{\frac{\beta-1}{2}}}\sum_{m=1}^{q}\|f\|_{W_2^\beta(\mathbb{R}^d)}\left\|\frac{\partial f_m}{\partial x_j}\right\|_{L^2(\mathbb{R}^d)}$$

$$2\sum_{m=1}^{q}\left\|\frac{\partial f_m}{\partial x_j}-\frac{\partial f_{m,v}}{\partial x_j}\right\|_{L^2(\mathbb{R}^d)}\left\|\frac{\partial f_{m,v}}{\partial x_j}\right\|_{L^2(\mathbb{R}^d)} \leq \frac{16\pi(2v)^{(\beta-1)}}{(1+v^2)^{\frac{\beta-1}{2}}}\sum_{m=1}^{q}\|f\|_{W_2^\beta(\mathbb{R}^d)}\left\|\frac{\partial f_m}{\partial x_j}\right\|_{L^2(\mathbb{R}^d)}\|g\|_{L^1(\mathbb{R}^d)}$$

$$2\sum_{m=1}^{q}\left\|\frac{\partial^2 f_m}{\partial x_i \partial x_j}-\frac{\partial^2 f_{m,v}}{\partial x_i \partial x_j}\right\|_{L^2(\mathbb{R}^d)}\|f_m\|_{L^2(\mathbb{R}^d)} \leq \frac{32\pi^2(2v)^{(\beta-2)}}{(1+v^2)^{\frac{\beta-2}{2}}}\sum_{m=1}^{q}\|f\|_{W_2^\beta(\mathbb{R}^d)}\|f_m\|_{L^2(\mathbb{R}^d)}$$

$$2\sum_{m=1}^{q}\|f_m-f_{m,v}\|_{L^2(\mathbb{R}^d)}\left\|\frac{\partial^2 f_{m,v}}{\partial x_i \partial x_j}\right\|_{L^2(\mathbb{R}^d)} \leq 2(2v)^\beta \sum_{m=1}^{q}\|f_m\|_{W_2^\beta(\mathbb{R}^d)}\left\|\frac{\partial^2 f_{m,v}}{\partial x_i \partial x_j}\right\|_{L^2(\mathbb{R}^d)}$$

$$= 2(2v)^\beta \sum_{m=1}^{q}\|f_m\|_{W_2^\beta(\mathbb{R}^d)}\left\|\frac{\partial^2 f_m \star g_v}{\partial x_i \partial x_j}\right\|_{L^2(\mathbb{R}^d)}$$

$$= 2(2v)^\beta \sum_{m=1}^{q}\|f_m\|_{W_2^\beta(\mathbb{R}^d)}\left\|\frac{\partial^2 f_m}{\partial x_i \partial x_j}\right\|_{L^2(\mathbb{R}^d)}\|g\|_{L^1(\mathbb{R}^d)}.$$

Hence,

$$\left\|\frac{\partial^2}{\partial x_i \partial x_j}p^\star(x,t)-\frac{\partial^2}{\partial x_i \partial x_j}f(x,t)\right\|_{L^1(\mathbb{R}^d)} \leq \frac{16\pi(2v)^{(\beta-1)}}{(1+v^2)^{\frac{\beta-1}{2}}}(1+\|g\|_{L^1(\mathbb{R}^d)})\sum_{m=1}^{q}\|f\|_{W_2^\beta(\mathbb{R}^d)}\left\|\frac{\partial f_m}{\partial x_j}\right\|_{L^2(\mathbb{R}^d)}$$

$$+\frac{32\pi^2(2v)^{(\beta-2)}}{(1+v^2)^{\frac{\beta-2}{2}}}\sum_{m=1}^{q}\|f\|_{W_2^\beta(\mathbb{R}^d)}\|f_m\|_{L^2(\mathbb{R}^d)}+2(2v)^\beta \sum_{m=1}^{q}\|f_m\|_{W_2^\beta(\mathbb{R}^d)}\left\|\frac{\partial^2 f_m}{\partial x_i \partial x_j}\right\|_{L^2(\mathbb{R}^d)}\|g\|_{L^1(\mathbb{R}^d)}.$$

Hence, if we choose,

$$v = \min\left(\left(\frac{\varepsilon}{48\pi 2^{\beta-1}C_1}\right)^{\frac{1}{\beta-1}}, \left(\frac{\varepsilon}{96\pi^2 2^{\beta-2}C_2}\right)^{\frac{1}{\beta-2}}, \left(\frac{1}{2^{\beta+2}C_3}\right)^{\frac{1}{\beta}}\right)$$

where $C_1 = (1+\|g\|_{L^1(\mathbb{R}^d)})\sum_{m=1}^{q}\|f\|_{W_2^\beta(\mathbb{R}^d)}\left\|\frac{\partial f_m}{\partial x_j}\right\|_{L^2(\mathbb{R}^d)}$, $C_2 = \sum_{m=1}^{q}\|f\|_{W_2^\beta(\mathbb{R}^d)}\|f_m\|_{L^2(\mathbb{R}^d)}$ and $C_3 = \sum_{m=1}^{q}\|f_m\|_{W_2^\beta(\mathbb{R}^d)}\left\|\frac{\partial^2 f_m}{\partial x_i \partial x_j}\right\|_{L^2(\mathbb{R}^d)}\|g\|_{L^1(\mathbb{R}^d)}$, then,

$$\left\|\frac{\partial^2}{\partial x_i \partial x_j}p^\star(x,t)-\frac{\partial^2}{\partial x_i \partial x_j}f(x,t)\right\|_{L^1(\mathbb{R}^d)} \leq \varepsilon.$$

This gives,

$$\left\|\sum_{i,j=1}^{d}D_{ij}\frac{\partial^2}{\partial x_i \partial x_j}p^\star(x,t)-D_{ij}\frac{\partial^2}{\partial x_i \partial x_j}\hat{p}(x,t)\right\|_{L^1(\mathbb{R}^d)} \leq d^2 R_d \varepsilon.$$

In similar way, we can also obtain bound for $L^2$ metric. Now we have,

$$\left\|\frac{\partial^2}{\partial x_i \partial x_j}p^\star(x,t)-\frac{\partial^2}{\partial x_i \partial x_j}f(x,t)\right\|_{L^2(\mathbb{R}^d)} = \left\|2\sum_{m=1}^{q}\left[\frac{\partial f_m}{\partial x_i}\frac{\partial f_m}{\partial x_j}-\frac{\partial f_{m,v}}{\partial x_i}\frac{\partial f_{m,v}}{\partial x_j}\right.\right.$$

$$+ f_m \frac{\partial^2 f_m}{\partial x_i \partial x_j} - f_{m,v} \frac{\partial^2 f_{m,v}}{\partial x_i \partial x_j} \Bigg] \Bigg\|_{L^2(\mathbb{R}^d)}$$

$$\leq 2 \sum_{m=1}^{q} \left[ \left\| \frac{\partial f_m}{\partial x_i} - \frac{\partial f_{m,v}}{\partial x_i} \right\|_{L^2(\mathbb{R}^d)} \left\| \frac{\partial f_m}{\partial x_j} \right\|_{L^\infty(\mathbb{R}^d)} + \left\| \frac{\partial f_m}{\partial x_j} - \frac{\partial f_{m,v}}{\partial x_j} \right\|_{L^2(\mathbb{R}^d)} \left\| \frac{\partial f_{m,v}}{\partial x_j} \right\|_{L^\infty(\mathbb{R}^d)} \right]$$

$$+ 2 \sum_{m=1}^{q} \left[ \left\| \frac{\partial^2 f_m}{\partial x_i \partial x_j} - \frac{\partial^2 f_{m,v}}{\partial x_i \partial x_j} \right\|_{L^2(\mathbb{R}^d)} \| f_m \|_{L^\infty(\mathbb{R}^d)} + \| f_m - f_{m,v} \|_{L^2(\mathbb{R}^d)} \left\| \frac{\partial^2 f_{m,v}}{\partial x_i \partial x_j} \right\|_{L^\infty(\mathbb{R}^d)} \right]$$

Let us now compute the following,

$$\left\| \frac{\partial f_{m,v}}{\partial x_j} \right\|_{L^\infty(\mathbb{R}^d)} \leq \left\| \frac{\partial f_m}{\partial x_j} \right\|_{L^\infty(\mathbb{R}^d)} \| g \|_{L^1(\mathbb{R}^d)}$$

and

$$\left\| \frac{\partial^2 f_{m,v}}{\partial x_i \partial x_j} \right\|_{L^\infty(\mathbb{R}^d)} \leq \left\| \frac{\partial^2 f_m}{\partial x_i \partial x_j} \right\|_{L^\infty(\mathbb{R}^d)} \| g \|_{L^1(\mathbb{R}^d)}$$

Similarly,

$$\left\| \sum_{i=1}^{d} \sum_{j=1}^{d} \left[ \frac{\partial^2}{\partial x_i \partial x_j} (D_{ij} p^\star(x,t)) - \frac{\partial^2}{\partial x_i \partial x_j} (D_{ij} f(x,t)) \right] \right\|_{L^2(\mathbb{R}^d)}$$

$$\leq \underbrace{\sup_{i,j} D_{ij}}_{:=R_d} \sum_{i=1}^{d} \sum_{j=1}^{d} \left\| \frac{\partial^2}{\partial x_i \partial x_j} p^\star(x,t) - \frac{\partial^2}{\partial x_i \partial x_j} f(x,t) \right\|_{L^2(\mathbb{R}^d)}. \tag{30}$$

Combining everything together, we have

$$\left\| \frac{\partial^2}{\partial x_i \partial x_j} p^\star(x,t) - \frac{\partial^2}{\partial x_i \partial x_j} f(x,t) \right\|_{L^2(\mathbb{R}^d)} \leq \frac{16\pi (2v)^{(\beta-1)}}{(1+v^2)^{\frac{\beta-1}{2}}} (1 + \| g \|_{L^1(\mathbb{R}^d)}) \sum_{m=1}^{q} \| f \|_{W_2^\beta(\mathbb{R}^d)} \left\| \frac{\partial f_m}{\partial x_j} \right\|_{L^\infty(\mathbb{R}^d)}$$

$$+ \frac{32\pi^2 (2v)^{(\beta-2)}}{(1+v^2)^{\frac{\beta-2}{2}}} \sum_{m=1}^{q} \| f \|_{W_2^\beta(\mathbb{R}^d)} \| f_m \|_{L^\infty(\mathbb{R}^d)} + 2(2v)^\beta \sum_{m=1}^{q} \| f_m \|_{W_2^\beta(\mathbb{R}^d)} \left\| \frac{\partial^2 f_m}{\partial x_i \partial x_j} \right\|_{L^\infty(\mathbb{R}^d)} \| g \|_{L^1(\mathbb{R}^d)}.$$

Hence, we get the final result by replacing $f(x,t)$ with $\hat{p}(x,t)$. $\qquad \square$

## A.2 Approximation Result by a PSD Model (Proof of Theorem 1)

*Proof.* Now, we just need to collect results from Lemmas 2 and 3 to get the final bound. Let us assume that there exist an optimal density function $p^\star(x,t)$ which is the solution of fokker-planck equation. Hence, We would like to show that there exists a psd operator $M_\varepsilon$ such that if $\hat{p}(x,t) = f(x,t) = \phi_{X,T}(x,t)^\top M_\varepsilon \phi_{X,T}(x,t)$ such that we can obtain bound for following three quantities for $r \in \{1,2\}$:

1. $\left\| \frac{\partial p^\star(x,t)}{\partial t} - \frac{\partial \hat{p}(x,t)}{\partial t} \right\|_{L^r(\mathbb{R}^{d+1})} = O(\varepsilon)$.

2.
$$\left\| \sum_{i=1}^{d} \frac{\partial}{\partial x_i} (\mu_i(x,t) p^\star(x,t) - \mu_i(x,t) \hat{p}(x,t)) \right\|_{L^r(\mathbb{R}^{d+1})} = \left\| \sum_{i=1}^{d} \left[ \mu_i(x,t) \left[ \frac{\partial p^\star(x,t)}{\partial x_i} - \frac{\partial \hat{p}(x,t)}{\partial x_i} \right] \right. \right.$$
$$\left. \left. + \frac{\partial \mu_i(x,t)}{\partial x_i} (p^\star(x,t) - \hat{p}(x,t)) \right] \right\|_{L^1(\mathbb{R}^{d+1})} = O(\varepsilon),$$

3. $\left\| \sum_{i=1}^{d} \sum_{j=1}^{d} \left[ \frac{\partial^2}{\partial x_i \partial x_j} (D_{ij} p^\star(x,t)) - \frac{\partial^2}{\partial x_i \partial x_j} (D_{ij} \hat{p}(x,t)) \right] \right\|_{L^1(\mathbb{R}^{d+1})} = O(\varepsilon)$.

If the above three condition holds, then, we can show that for $r \in \{1, 2\}$,

$$\left\| \frac{\partial \hat{p}(x,t)}{\partial t} + \sum_{i=1}^{d} \frac{\partial}{\partial x_i}(\mu_i(x,t)\hat{p}(x,t)) - \sum_{i=1}^{d}\sum_{j=1}^{d} \frac{\partial^2}{\partial x_i \partial x_j}(D_{ij}\hat{p}(x,t)) \right\|_{L^r(\mathbb{R}^{d+1})} = O(\varepsilon).$$

We have from Lemma 2 and equation (26), if we choose

$$v = \min\left( \left( \frac{\varepsilon}{2^{\beta-1}16\pi C_1} \right)^{\frac{1}{\beta-1}}, \left( \frac{\varepsilon}{2^{\beta+1}C_2} \right)^{\frac{1}{\beta}} \right)$$

where $C_1 = \sum_{i=1}^{q} \|f_i\|_{W_2^{\beta}(\mathbb{R}^{d+1})} \|f_i\|_{L^{\infty}(\mathbb{R}^{d+1})}$ and $C_2 = \sum_{i=1}^{q} \|f_i\|_{W_2^{\beta}(\mathbb{R}^{d+1})} \left\| \frac{\partial f_i}{\partial t} \right\|_{L^{\infty}(\mathbb{R}^{d+1})} \|g\|_{L_1(\mathbb{R}^{d+1})}$. Hence,

$$\left\| \frac{\partial p^{\star}(x,t)}{\partial t} - \frac{\partial f(x,t)}{\partial t} \right\|_{L^2(\mathbb{R}^{d+1})} \leq \varepsilon.$$

From [RC21, Theorem D.4 (Eq. D.25)], we have,

$$\|p^{\star}(x,t) - \hat{p}(x,t)\|_{L^2(\mathbb{R}^{d+1})} \leq (2v)^{\beta}(1 + \|g\|_{L^1(\mathbb{R}^{d+1})}) \sum_{i=1}^{q} \|f_i\|_{W_2^{\beta}(\mathbb{R}^{d+1})} \|f_i\|_{L^{\infty}(\mathbb{R}^{d+1})}.$$

This proves the first part. Going further,

$$\left\| \sum_{i=1}^{d} \frac{\partial}{\partial x_i}(\mu_i(x,t)p^{\star}(x,t) - \mu_i(x,t)\hat{p}(x,t)) \right\|_{L^2(\mathbb{R}^{d+1})} = \left\| \sum_{i=1}^{d} \left[ \mu_i(x,t) \left[ \frac{\partial p^{\star}(x,t)}{\partial x_i} - \frac{\partial \hat{p}(x,t)}{\partial x_i} \right] \right. \right.$$
$$\left. \left. + \frac{\partial \mu_i(x,t)}{\partial x_i}(p^{\star}(x,t) - \hat{p}(x,t)) \right] \right\|_{L^2(\mathbb{R}^{d+1})}$$

$$\leq \sum_{i=1}^{d} \left\| \mu_i(x,t) \left[ \frac{\partial p^{\star}(x,t)}{\partial x_i} - \frac{\partial \hat{p}(x,t)}{\partial x_i} \right] \right\|_{L^2(\mathbb{R}^{d+1})} + \sum_{i=1}^{d} \left\| \frac{\partial \mu_i(x,t)}{\partial x_i}(p^{\star}(x,t) - \hat{p}(x,t)) \right\|_{L^2(\mathbb{R}^{d+1})}$$

$$\leq \underbrace{\sup_{x,t \in (\mathfrak{X},\mathfrak{T})} \sup_{i} \mu_i(x,t)}_{\leq R_{\mu}} \sum_{i=1}^{d} \left\| \frac{\partial p^{\star}(x,t)}{\partial x_i} - \frac{\partial \hat{p}(x,t)}{\partial x_i} \right\|_{L^2(\mathbb{R}^{d+1})}$$

$$+ d \underbrace{\sup_{x,t \in (\mathfrak{X},\mathfrak{T})} \sup_{i} \frac{\partial \mu_i(x,t)}{\partial x_i}}_{\leq R_{\mu_p}} \|p^{\star}(x,t) - \hat{p}(x,t)\|_{L^2(\mathbb{R}^{d+1})}$$

$$\leq R_{\mu} \sum_{i=1}^{d} \left\| \frac{\partial p^{\star}(x,t)}{\partial x_i} - \frac{\partial \hat{p}(x,t)}{\partial x_i} \right\|_{L^2(\mathbb{R}^{d+1})} + dR_{\mu_p} \|p^{\star}(x,t) - \hat{p}(x,t)\|_{L^2(\mathbb{R}^{d+1})} \tag{31}$$

Now,

$$\left\| \sum_{i=1}^{d} \frac{\partial}{\partial x_i}(\mu_i(x,t)p^{\star}(x,t) - \mu_i(x,t)\hat{p}(x,t)) \right\|_{L^2(\mathbb{R}^{d+1})}$$

$$\leq R_{\mu} \sum_{i=1}^{d} \left\| \frac{\partial p^{\star}(x,t)}{\partial x_i} - \frac{\partial \hat{p}(x,t)}{\partial x_i} \right\|_{L^2(\mathbb{R}^{d+1})} + dR_{\mu_p} \|p^{\star}(x,t) - \hat{p}(x,t)\|_{L^2(\mathbb{R}^{d+1})}$$

From [RC21, Theorem D.4 (Eq. D.25)], we have,

$$\|p^{\star}(x,t) - \hat{p}(x,t)\|_{L^1(\mathbb{R}^{d+1})} \leq (2v)^{\beta}(1 + \|g\|_{L^1(\mathbb{R}^{d+1})}) \sum_{i=1}^{q} \|f_i\|_{W_2^{\beta}(\mathbb{R}^{d+1})} \|f_i\|_{L^{\infty}(\mathbb{R}^{d+1})}.$$

And previously, we showed in lemma 2

$$\left\| \frac{\partial p^{\star}(x,t)}{\partial x_k} - \frac{\partial \hat{p}(x,t)}{\partial x_k} \right\|_{L^2(\mathbb{R}^{d+1})} \leq 8\pi(2v)^{(\beta-1)} \sum_{i=1}^{q} \|f_i\|_{W_2^{\beta}(\mathbb{R}^{d+1})} \|f_i\|_{L^{\infty}(\mathbb{R}^{d+1})} \tag{32}$$

$$+ 2(2v)^{\beta} \sum_{i=1}^{q} \|f_i\|_{W_2^{\beta}(\mathbb{R}^{d+1})} \left\| \frac{\partial f_i}{\partial x_k} \right\|_{L^{\infty}(\mathbb{R}^{d+1})} \|g\|_{L_1(\mathbb{R}^{d+1})}$$

$$(33)$$

If we choose,

$$v = \min\left( \left( \frac{\varepsilon}{2^{\beta-1}16\pi C_1} \right)^{\frac{1}{\beta-1}}, \left( \frac{\varepsilon}{2^{\beta+1}C_2} \right)^{\frac{1}{\beta}}, \left( \frac{\varepsilon}{2^{\beta+1}C_3} \right)^{\frac{1}{\beta}} \right)$$

where $C_1 = \sum_{i=1}^{q} \|f_i\|_{W_2^{\beta}(\mathbb{R}^{d+1})} \|f_i\|_{L^{\infty}(\mathbb{R}^{d+1})}$, $C_2 = \sum_{i=1}^{q} \|f_i\|_{W_2^{\beta}(\mathbb{R}^{d+1})} \left\| \frac{\partial f_i}{\partial t} \right\|_{L^{\infty}(\mathbb{R}^{d+1})} \|g\|_{L_1(\mathbb{R}^{d+1})}$ and $C_3 = (1 + \|g\|_{L^1(\mathbb{R}^{d+1})}) \sum_{i=1}^{q} \|f_i\|_{W_2^{\beta}(\mathbb{R}^{d+1})} \|f_i\|_{L_2(\mathbb{R}^{d+1})}$, then, we have

$$\left\| \sum_{i=1}^{d} \frac{\partial}{\partial x_i} (\mu_i(x,t)p^{\star}(x,t) - \mu_i(x,t)\hat{p}(x,t)) \right\|_{L^2(\mathbb{R}^{d+1})} \leq d(R_\mu + R_{\mu_p})\varepsilon.$$

We showed in lemma 3 that

$$\left\| \frac{\partial^2}{\partial x_i \partial x_j} p^{\star}(x,t) - \frac{\partial^2}{\partial x_i \partial x_j} \hat{p}(x,t) \right\|_{L^2(\mathbb{R}^{d+1})}$$

$$\leq \frac{16\pi(2v)^{(\beta-1)}}{(1+v^2)^{\frac{\beta-1}{2}}} (1 + \|g\|_{L^1(\mathbb{R}^{d+1})}) \sum_{m=1}^{q} \|f\|_{W_2^{\beta}(\mathbb{R}^{d+1})} \left\| \frac{\partial f_m}{\partial x_j} \right\|_{L^{\infty}(\mathbb{R}^{d+1})}$$

$$+ \frac{32\pi^2(2v)^{(\beta-2)}}{(1+v^2)^{\frac{\beta-2}{2}}} \sum_{m=1}^{q} \|f\|_{W_2^{\beta}(\mathbb{R}^{d+1})} \|f_m\|_{L^{\infty}(\mathbb{R}^{d+1})} + 2(2v)^{\beta} \sum_{m=1}^{q} \|f_m\|_{W_2^{\beta}(\mathbb{R}^{d+1})} \left\| \frac{\partial^2 f_m}{\partial x_i \partial x_j} \right\|_{L^{\infty}(\mathbb{R}^{d+1})} \|g\|_{L^1(\mathbb{R}^{d+1})}.$$

Hence, if we choose,

$$v = \min\left( \left( \frac{\varepsilon}{48\pi 2^{\beta-1}C_1} \right)^{\frac{1}{\beta-1}}, \left( \frac{\varepsilon}{96\pi^2 2^{\beta-2}C_2} \right)^{\frac{1}{\beta-2}}, \left( \frac{1}{2^{\beta+2}C_3} \right)^{\frac{1}{\beta}} \right)$$

where $C_1 = (1 + \|g\|_{L^1(\mathbb{R}^{d+1})}) \sum_{m=1}^{q} \|f\|_{W_2^{\beta}(\mathbb{R}^{d+1})} \left\| \frac{\partial f_m}{\partial x_j} \right\|_{L^{\infty}(\mathbb{R}^{d+1})}$, $C_2 = \sum_{m=1}^{q} \|f\|_{W_2^{\beta}(\mathbb{R}^{d+1})} \|f_m\|_{L^{\infty}(\mathbb{R}^{d+1})}$ and $C_3 = \sum_{m=1}^{q} \|f_m\|_{W_2^{\beta}(\mathbb{R}^{d+1})} \left\| \frac{\partial^2 f_m}{\partial x_i \partial x_j} \right\|_{L^{\infty}(\mathbb{R}^{d+1})} \|g\|_{L^1(\mathbb{R}^{d+1})}$, then,

$$\left\| \frac{\partial^2}{\partial x_i \partial x_j} p^{\star}(x,t) - \frac{\partial^2}{\partial x_i \partial x_j} \hat{p}(x,t) \right\|_{L^2(\mathbb{R}^{d+1})} \leq \varepsilon.$$

This gives,

$$\left\| \frac{\partial \hat{p}(x,t)}{\partial t} + \sum_{i=1}^{d} \frac{\partial}{\partial x_i} (\mu_i(x,t)\hat{p}(x,t)) - \sum_{i=1}^{d} \sum_{j=1}^{d} \frac{\partial^2}{\partial x_i \partial x_j} (D_{ij}\hat{p}(x,t)) \right\|_{L^2(\mathbb{R}^{d+1})} \leq \varepsilon(1 + d(R_\mu + R_{\mu_p}) + d^2 R_d).$$

We can bound the trace of the matrix as follows. We have,

$$M_\varepsilon = \sum_{i=1}^{q} f_{i,v} f_{i,v}^{\top}.$$

From [RC21, Equation D.21], we have

$$Tr(M_\varepsilon) = \sum_{i=1}^{q} \|f_{i,v}\|_{\mathcal{H}} = c_\eta 2^{2\beta} (1 + (\frac{v}{3})^{2\beta} e^{\frac{89}{\eta_0 v^2}}) \sum_{i=1}^{q} \|f_i\|_{W_2^{\beta}(\mathbb{R}^{d+1})}^2$$

Since, we choose,

$$v = \left( \frac{\varepsilon}{96\pi^2 2^{\beta-2}C} \right)^{\frac{1}{\beta-2}},$$

where $C = C_1 + C_2 + C_3$, $C_1 = (1 + \|g\|_{L^1(\mathbb{R}^{d+1})}) \sum_{m=1}^{q} \|f\|_{W_2^\beta(\mathbb{R}^{d+1})} \left\| \frac{\partial f_m}{\partial x_j} \right\|_{L^\infty(\mathbb{R}^{d+1})}$, $C_2 = \sum_{m=1}^{q} \|f\|_{W_2^\beta(\mathbb{R}^{d+1})} \|f_m\|_{L^\infty(\mathbb{R}^{d+1})}$ and $C_3 = \sum_{m=1}^{q} \|f_m\|_{W_2^\beta(\mathbb{R}^{d+1})} \left\| \frac{\partial^2 f_m}{\partial x_i \partial x_j} \right\|_{L^\infty(\mathbb{R}^{d+1})} \|g\|_{L^1(\mathbb{R}^{d+1})}$.

$$
\begin{aligned}
Tr(M_\varepsilon) &= c_\eta 2^{2\beta} (1 + (\frac{v}{3})^{2\beta} e^{\frac{89}{\eta_0 v^2}}) \sum_{i=1}^{q} \|f_i\|_{W_2^\beta(\mathbb{R}^{d+1})}^2 \\
&= c_\eta 2^{2\beta} \left( 1 + 3^{-2\beta} \left( \frac{\varepsilon}{96\pi^2 2^{\beta-2} C} \right)^{\frac{2\beta}{\beta-2}} e^{\frac{89}{\eta_0 v^2}} \right) \sum_{i=1}^{q} \|f_i\|_{W_2^\beta(\mathbb{R}^{d+1})}^2 \\
&= c_\eta 2^{2\beta} \sum_{i=1}^{q} \|f_i\|_{W_2^\beta(\mathbb{R}^{d+1})}^2 \left( 1 + 3^{-2\beta} \left( \frac{\varepsilon}{\tilde{C}} \right)^{\frac{2\beta}{\beta-2}} e^{\frac{89}{\eta_0 \left( \frac{\varepsilon}{\tilde{C}} \right)^{\frac{2}{\beta-2}}}} \right) \\
&= \hat{C} |\eta|^{1/2} \left( 1 + \varepsilon^{\frac{2\beta}{\beta-2}} e^{\frac{\tilde{C}'}{\eta_0 \varepsilon^{\frac{2}{\beta-2}}}} \right) \leq \hat{C} |\eta|^{1/2} \left( 1 + \varepsilon^{\frac{2\beta}{\beta-2}} \exp \left( \frac{\tilde{C}'}{\eta_0} \varepsilon^{-\frac{2}{\beta-2}} \right) \right)
\end{aligned}
$$

where $\tilde{C} = 96\pi^2 2^{\beta-2} C$, $\tilde{C}' = 89\tilde{C}^{\frac{2}{\beta-2}}$, and $\hat{C} = \pi^{-d/2} 2^{2\beta} \sum_{i=1}^{q} \|f_i\|_{W_2^\beta(\mathbb{R}^{d+1})}^2 \max \left( 1, \frac{3^{-2\beta}}{\tilde{C}^{\frac{2\beta}{\beta-2}}} \right)$. $\qquad \square$

## A.3 Compression Result

Let $\mathcal{X} \subset \mathbb{R}^d$ be an open set with Lipschitz boundary, contained in the hypercube $[-R, R]^d$ with $R > 0$. Given $\tilde{x}_1, \ldots, \tilde{x}_m \in \mathcal{X}$ be $m$ points in $[-R, R]^d$. Define the base point matrix $\tilde{X} \in \mathbb{R}^{m \times d}$ to be the matrix whose $j$-th row is the point $\tilde{x}_j$. The following result holds. We introduce the so called *fill distance* [Wen04]

$$
h = \max_{x \in [-R, R]^d} \min_{z \in \tilde{X}} \|x - z\|. \tag{34}
$$

Now, we state some useful results below.

## A.4 Useful Results

**Lemma 4** ([RC21], Lemma C.1 ). *Let $T = (-R, R)^d$ and $\eta \in \mathbb{R}_{++}^d$. Let $u \in \mathcal{H}_\eta$ satisfying $u(\tilde{x}_1) = \cdots = u(\tilde{x}_m) = 0$. There exists three constants $c, C, C'$ depending only on $d$ (and in particular, independent from $R, \eta, u, \tilde{x}_1, \ldots, \tilde{x}_m$), such that, when $h \leq \sigma/C'$, then,*

$$
\|u\|_{L^\infty(T)} \leq Cq_\eta e^{-\frac{c\sigma}{h} \log \frac{c\sigma}{h}} \|u\|_{\mathcal{H}_\eta},
$$

*with $q_\eta = \det(\frac{1}{\eta_+} \operatorname{diag}(\eta))^{-1/4}$ and $\sigma = \min(R, \frac{1}{\sqrt{\eta_+}})$ and $\eta_+ = \max_{i=1,\ldots,d} \eta_i$.*

**Lemma 5** (Norm of function's derivative with scattered zeros). *Let $T = (-R, R)^d$ and $\eta \in \mathbb{R}_{++}^d$. Let $u \in \mathcal{H}_\eta$ satisfying $u(\tilde{x}_1) = \cdots = u(\tilde{x}_m) = 0$. There exists three constants $c, C, C'$ depending only on $d$ (and in particular, independent from $R, \eta, u, \tilde{x}_1, \ldots, \tilde{x}_m$), such that, when $h \leq \sigma/C'$, then,*

$$
\|D^j u\|_{L^\infty(T)} \leq Ge^{-c\sigma/h} \log(c\sigma/h) \|u\|_{\mathcal{H}_\eta},
$$

*with $G$ be some positive constant and $\sigma = \min(R, \frac{1}{\sqrt{\eta_+}})$ and $\eta_+ = \max_{i=1,\ldots,d} \eta_i$.*

*Proof.* Here, we utilize Gagliardo–Nirenberg inequality [Nir11]. We have, for $0 < k < \infty$, $0 \leq j < \infty$ and $\vartheta(j, m) = \frac{j}{k} + \frac{d}{2k} < 1$, then

$$
\|D^j u\|_{L^\infty(T)} \leq G\|u\|_{L^2(T)}^{1-\vartheta(j,k)} \|D^k u\|_{L^2(T)}^{\vartheta(j,k)}, \tag{35}
$$

where $G = \frac{1}{2^{d/2} \pi^{d/4-1/2} \sqrt{\Gamma(d/2)(1-\vartheta)^{1-\vartheta} \vartheta^\vartheta k \sin \pi\vartheta}}$.

Considering the case for $j = 1$ and $j = 2$, we have,

$$
\|D^1 u\|_{L^\infty(T)} \leq G_1 \|u\|_{L^2(T)}^{1-\frac{d+2}{2k}} \|D^k u\|_{L^2(T)}^{\frac{d+2}{2k}},
$$

$$\|D^2 u\|_{L^\infty(T)} \le G_2 \|u\|_{L^2(T)}^{1-\frac{d+4}{2k}} \|D^k u\|_{L^2(T)}^{\frac{d+4}{2k}}$$

where $G_1 = \dfrac{1}{2^{d/2}\pi^{d/4-1/2}\sqrt{\Gamma(d/2)(1-\frac{d+2}{2k})^{1-\frac{d+2}{2k}}(\frac{d+2}{2k})^{\frac{d+2}{2k}}k\sin\pi(\frac{d+2}{2k})}}$ and

$G_2 = \dfrac{1}{2^{d/2}\pi^{d/4-1/2}\sqrt{\Gamma(d/2)(1-\frac{d+4}{2k})^{1-\frac{d+4}{2k}}(\frac{d+4}{2m})^{\frac{d+4}{2k}}k\sin\pi(\frac{d+4}{2k})}}$. We here have assumed that $k > \frac{d}{2} + 2$.

From the defeinition, we have

$$\|D^1 u\|_{L^\infty(T)} \le G_1 \|u\|_{L^2(T)}^{1-\frac{d+2}{2k}} \|D^k u\|_{L^2(T)}^{\frac{d+2}{2k}},$$

$$\|D^2 u\|_{L^\infty(T)} \le G_2 \|u\|_{L^2(T)}^{1-\frac{d+4}{2k}} \|D^k u\|_{L^2(T)}^{\frac{d+4}{2k}}.$$

From the definition,

$$\|D^1 u\|_{L^\infty(T)} \le G_1 \|u\|_{L^2(T)}^{1-\frac{d+2}{2k}} \|u\|_{W_2^k(T)}^{\frac{d+2}{2k}},$$

$$\|D^2 u\|_{L^\infty(T)} \le G_2 \|u\|_{L^2(T)}^{1-\frac{d+4}{2k}} \|u\|_{W_2^k(T)}^{\frac{d+4}{2k}}.$$

Now by Theorem 4.3 of [RZ10], we have for some constants $B_d$ and $B_d'$ depending only on d,

$$\|u\|_{L^2(T)} \le \frac{B_d^k k^k}{k!} h^k \|u\|_{W_2^k(T)},$$

when $kh \le \frac{R}{B_d'}$. Hence, we have,

$$\|D^1 u\|_{L^\infty(T)} \le G_1 \frac{B_d^k k^k}{k!} h^k \|u\|_{W_2^k(T)},$$

$$\|D^2 u\|_{L^\infty(T)} \le G_2 \frac{B_d^k k^k}{k!} h^k \|u\|_{W_2^k(T)}.$$

From equation C.9 in [RC21, Proof of Lemma C.1], we have

$$\|u\|_{W_2^k(T)} \le \|u\|_{\mathcal{H}_\eta} c_\eta^{1/2} (4\eta_+)^{k/2} \sqrt{k!}.$$

Hence,

$$\|D^1 u\|_{L^\infty(T)} \le G_1 \frac{B_d^k k^k}{k!} h^k \|u\|_{\mathcal{H}_\eta} c_\eta^{1/2} (4\eta_+)^{k/2} \sqrt{k!} = G_1 \frac{(2\sqrt{\eta_+} B_d h k)^k}{\sqrt{k!}} \|u\|_{\mathcal{H}_\eta},$$

$$\|D^2 u\|_{L^\infty(T)} \le G_2 \frac{B_d^k k^k}{k!} h^k \|u\|_{\mathcal{H}_\eta} c_\eta^{1/2} (4\eta_+)^{k/2} \sqrt{k!} = G_2 \frac{(2\sqrt{\eta_+} B_d h k)^k}{\sqrt{k!}} \|u\|_{\mathcal{H}_\eta}.$$

Now, we make similar choice of the parameter $h$ and $k$ as in [RC21]. If we choose, $C_3 = \frac{1}{\max(2B_d, B_d')}$, $h \le \frac{C_3}{d+4}\min(R, 1/\sqrt{\eta_+})$, $k = \lfloor s \rfloor$ and $s = \frac{C_3}{h}\min(R, 1/\sqrt{\eta_+})$, then we have $hk \le \frac{R}{B_d'}$ and $2\sqrt{\eta_+} B_d h k \le 1$. Moreover, $\frac{1}{\sqrt{k!}} \le e^{-\frac{k}{2}\log\frac{k}{2}}$. If we also assume that $G = \max(G_1, G_2)$, then for $j = \{1, 2\}$

$$\|D^j u\|_{L^\infty(T)} \le G \frac{1}{\sqrt{k!}} \|u\|_{\mathcal{H}_\eta} \le G e^{-s/4} \log(s/4) \|u\|_{\mathcal{H}_\eta}.$$

The final result is obtained by writing $s/4 = c\sigma/h$ with $\sigma = \min(R, 1/\sqrt{\eta_+})$ and $c = C_3/4$. $\qquad\square$

**Lemma 6** (Lemma 3, page 28 [PRDVR20]). *Let $\mathcal{X} \subset \mathbb{R}^d$ with non-zero volume. Let $\mathcal{H}$ be a reproducing kernel Hilbert space on $\mathcal{X}$, associated to a continuous uniformly bounded feature map $\phi : \mathcal{X} \to \mathcal{H}$. Let $A : \mathcal{H} \to \mathcal{H}$ be a bounded linear operator. Then,*

$$\sup_{x \in \mathcal{X}} \|A\phi(x)\|_{\mathcal{H}} \le \sup_{\|f\|_{\mathcal{H}} \le 1} \|A^* f\|_{C(\mathcal{X})}.$$

*In particular, if $\mathcal{X} \subset \mathbb{R}^d$ is a non-empty open set, then $\sup_{x \in \mathcal{X}} \|A\phi(x)\|_{\mathcal{H}} \le \sup_{\|f\|_{\mathcal{H}} \le 1} \|A^* f\|_{L^\infty(\mathcal{X})}.$*

**Theorem 6** ( Theorem C.3 [RC21]). *Let $R > 0, \eta \in \mathbb{R}_{++}^d, m \in \mathbb{N}$. Let $\mathfrak{X} \subseteq T = (-R, R)^d$ be a non-empty open set and let $\tilde{x}_1, \ldots, \tilde{x}_m$ be a set of distinct points. Let $h > 0$ be the fill distance associated to the points w.r.t $T$ (defined in eq. (34)). Let $\tilde{P} : \mathcal{H}_\eta \to \mathcal{H}_\eta$ be the associated projection operator (see definition in eq. (38)). There exists three constants $c, C, C'$, such that, when $h \le \sigma/C'$,*

$$\sup_{x \in \mathfrak{X}} \|(I - \tilde{P})\phi_\eta(x)\|_{\mathcal{H}_\eta} \le C q_\eta \, e^{-\frac{c\sigma}{h} \log \frac{c\sigma}{h}}.$$

*Here $q_\eta = \det(\frac{1}{\eta_+} \operatorname{diag}(\eta))^{-1/4}$ and $\sigma = \min(R, \frac{1}{\sqrt{\eta_+}})$, $\eta_+ = \max_i \eta_i$. The constants $c, C', C''$ depend only on $d$ and, in particular, are independent from $R, \eta, \tilde{x}_1, \ldots, \tilde{x}_m$.*

Similar to the results in theorem 6, we will obtain results for derivatives.

**Theorem 7.** *Let $R > 0, \eta \in \mathbb{R}_{++}^d, m \in \mathbb{N}$. Let $\mathfrak{X} \subseteq T = (-R, R)^d$ be a non-empty open set and let $\tilde{x}_1, \ldots, \tilde{x}_m$ be a set of distinct points. Let $h > 0$ be the fill distance associated to the points w.r.t $T$ (defined in eq. (34)). Let $\tilde{P} : \mathcal{H}_\eta \to \mathcal{H}_\eta$ be the associated projection operator (see definition in eq. (38)). There exists three constants $c, C, C'$, such that, when $h \le \sigma/C'$,*

$$\sup_{x \in \mathfrak{X}} \|(I - \tilde{P})D^j\phi_\eta(x)\|_{\mathcal{H}_\eta} \le C_1 q_\eta \, e^{-\frac{c\sigma}{h} \log \frac{c\sigma}{h}},$$

*for some positive constant $C_1$. Here $q_\eta = \det(\frac{1}{\eta_+} \operatorname{diag}(\eta))^{-1/4}$ and $\sigma = \min(R, \frac{1}{\sqrt{\eta_+}})$, $\eta_+ = \max_i \eta_i$. The constants $c, C', C''$ depend only on $d$ and, in particular, are independent from $R, \eta, \tilde{x}_1, \ldots, \tilde{x}_m$.*

*Proof.* If the function $f$ lies in the RKHS, then using the property of the projection operator $\tilde{P}$, we have

$$D^j(\tilde{P}f)(x_i) = D^j\langle \tilde{P}f, \phi(x_i)\rangle = D^j\langle f, \tilde{P}\phi(x_i)\rangle = D^j\langle f, \phi(x_i)\rangle = \langle f, D^j\phi(x_i) = (D^j f)(x_i).$$

Hence, $D^j f - \tilde{P}D^j f(\tilde{x}_i) = 0$ for all $i \in \{1, \cdots, m\}$. By lemma 4, we know that there exist three constants $c, C, C'$ depending only on $d$ such that when $h \le \sigma/C'$ we have that the following holds $\|u\|_{L^\infty(T)} \le C q_\eta \, e^{-\frac{c\sigma}{h} \log \frac{c\sigma}{h}}$, for any $u \in \mathcal{H}_\eta$ such that $u(\tilde{x}_1) = \cdots = u(\tilde{x}_m) = 0$. Since, for any $u \in \mathcal{H}_\eta$, we have that $D^j f - \tilde{P}D^j f$ belongs to $\mathcal{H}_\eta$ and satisfies such property, we can apply lemma 4 with $u = (I - \tilde{P})D^j f$, obtaining, under the same assumption on $h$,

$$\|(I - \tilde{P})D^j f\|_{L^\infty(T)} \le C q_\eta \, e^{-\frac{c\sigma}{h} \log \frac{c\sigma}{h}} \|D^j f\|_{\mathcal{H}_\eta}, \quad \forall f \in \mathcal{H}_\eta,$$

where we used the fact that $\|(I - \tilde{P})D^j f\|_{\mathcal{H}_\eta} \le \|I - \tilde{P}\|\|D^j f\|_{\mathcal{H}_\eta}$ and $\|I - \tilde{P}\| \le 1$, since $P$ is a projection operator and so also $I - P$ satisfies this property. Derivative reproducing properties for kernel can be looked at in [Zho08]. The final result is obtained by applying lemma 6 with $A = I - \tilde{P}$, from which we have

$$\sup_{x \in \mathfrak{X}} \|(I - \tilde{P})D^j\phi(x)\|_{\mathcal{H}} \le \sup_{\|D^j f\| \le 1} \|(I - \tilde{P})D^j f\|_{L^\infty(T)} \le \sup_{\|D^j f\| \le 1} C q_\eta \, e^{-\frac{c\sigma}{h} \log \frac{c\sigma}{h}} \|D^j f\|_{\mathcal{H}_\eta}$$
$$= C_1 q_\eta \, e^{-\frac{c\sigma}{h} \log \frac{c\sigma}{h}},$$

for some positive $C_1$. The same result can also be obtained by the use of Lemma 5 with a condition that $f$ is $d/2 + 1$ differentiable function which is a slightly weaker condition on the order of smoothness. $\square$

## A.5 Compression of a PSD model for Fokker-Planck

We have $(\tilde{x}_1, \tilde{t}_1), \tilde{x}_2, \cdots (\tilde{x}_m, \tilde{t}_m) \in \mathfrak{X} \times \mathcal{T}$ which is an open bounded subset with Lipschitz boundary of the cube $[-R, R]^d \times [0, T]$. $\tilde{X}$ be the base point matrix whose j-rows are the points $\tilde{x}_j$. Let us now consider the model $p = f(\cdot, M, \psi)$ and $\tilde{p} = f(\cdot; A_m, \tilde{X}, \psi)$ where $A_m = K_{(X,T)(X,T)}^{-1} \tilde{Z} M \tilde{Z}^\star K_{(X,T)(X,T)}^{-1}$.

$$\tilde{Z}u = (\psi(\tilde{x}_1, \tilde{t}_1)^\top u, \cdots, \psi(\tilde{x}_m, \tilde{t}_m)^\top u). \tag{36}$$

and

$$\tilde{Z}^{\star}\alpha = \sum_{i=1}^{m} \psi(\tilde{x}_i, \tilde{t}_i)\alpha_i. \tag{37}$$

Hence, for any $A \in \mathbb{R}^{m \times m}$

$$\tilde{Z}^{\star}A\tilde{Z} = \sum_{i,j=1}^{m} A_{i,j}\psi(\tilde{x}_i, \tilde{t}_i)\psi(\tilde{x}_i, \tilde{t}_i)^{\top}.$$

Let us also define associated projection operator,

$$\tilde{P} = \tilde{Z}^{\star}K_{(X,T)(X,T)}^{-1}\tilde{Z}. \tag{38}$$

Some properties associated with it,

$$\tilde{P}^2 = \tilde{P}, \ \tilde{P}\tilde{Z}^{\star} = \tilde{Z}^{\star}, \ \tilde{Z}\tilde{P} = \tilde{Z}.$$

**Theorem 8.** *Let $\eta \in \mathbb{R}_{++}^d$ and let $\mathsf{M} \in \mathbb{S}_+(\mathcal{H}_\eta)$. Let $\tilde{\mathcal{X}}$ be an open bounded subset with the Lipschitz boundary of the cube $[-R, R]^d \times [0, R]$, $R > 0$. Let $\tilde{x}_1, \ldots, \tilde{x}_m \in \tilde{\mathcal{X}}$ and $\tilde{X}$ be the base point matrix whose $j$-rows are the points $\tilde{x}_j$ with $j = 1, \ldots, m$. Consider the model $p = f(\cdot\,; \mathsf{M}, \psi)$ and the the compressed model $\tilde{p} = f(\cdot\,; A_m, \tilde{X}, \eta)$ with*

$$A_m \;=\; K_{\tilde{X}, \tilde{X}, \eta}^{-1} \, \tilde{Z}\mathsf{M}\tilde{Z}^* \, K_{\tilde{X}, \tilde{X}, \eta}^{-1},$$

*where $\tilde{Z} : \mathcal{H}_\eta \to \mathbb{R}^m$ is defined in eq. (36) in terms of $\tilde{X}_m$. Let $h$ be the fill distance (defined in eq. (34)) associated to the points $\tilde{x}_1, \ldots, \tilde{x}_m$. The there exist three constants $c, C, C'$ depending only on $d$ such that, when $h \leq \sigma/C'$, with $\sigma = \min(R, 1/\sqrt{\eta_+}), \eta_+ = \max_{i=1,\ldots,d} \eta_i, q_\eta = \det(\frac{1}{\eta_+}\operatorname{diag}(\eta))^{-1/4}$, then we have*

$$\left| \frac{\partial(p(x,t) - \tilde{p}(x,t))}{\partial t} + \sum_{i=i}^{d} \frac{\partial(\mu_i(x,t)(p(x,t) - \tilde{p}(x,t)))}{\partial x_i} - \sum_{i=1}^{d}\sum_{j=1}^{d} D_{ij}\frac{\partial^2}{\partial x_i \partial x_j}(p(x,t) - \tilde{p}(x,t)) \right|$$

$$\leq (2 + dR_\mu + 2dR_\mu + 4d^2 R_d)\|M\|C^2 q_\eta^2 e^{\frac{-2c\sigma}{h} \log \frac{c\sigma}{h}} + (2 + dR_{\mu_p} + 2dR_\mu + 2d^2 R_d)\sqrt{\|M\|p(x,t)}\,Cq_\eta e^{-\frac{c\sigma}{h} \log \frac{c\sigma}{h}}$$

$$+ 2Cq_\eta e^{-\frac{c\sigma}{h} \log \frac{c\sigma}{h}}\sqrt{\|M\|\tilde{f}_t(x; M, \psi)} + (2R_\mu + 4dR_d)Cq_\eta e^{-\frac{c\sigma}{h} \log \frac{c\sigma}{h}}\sum_{i=1}^{d}\sqrt{\|M\|\tilde{f}_{x_i}(x; M, \psi)}$$

$$+ 2R_d Cq_\eta e^{-\frac{c\sigma}{h} \log \frac{c\sigma}{h}}\sum_{i=1}^{d}\sum_{j=1}^{d}\sqrt{\|M\|\tilde{f}_{x_j x_i}(x; M, \psi)},$$

*where $\tilde{f}_{x_i}(x; M, \psi) = \frac{\partial \psi(x,t)}{\partial x_i}^{\top} M \frac{\partial \psi(x,t)}{\partial x_i}$, $\tilde{u}_{x_i}(x) = \left\| (I - \tilde{P})\frac{\partial \psi(x,t)}{\partial x_i} \right\|_{\mathcal{H}_\eta}$, $\tilde{f}_{x_j x_i}(x; M, \psi) = \frac{\partial^2 \psi(x,t)}{\partial x_j x_i}^{\top} M \frac{\partial^2 \psi(x,t)}{\partial x_j x_i}$ and $\tilde{u}_{x_j x_i}(x) = \left\| (I - \tilde{P})\frac{\partial^2 \psi(x,t)}{\partial x_j x_i} \right\|_{\mathcal{H}_\eta}$.*

*Proof.* We have,

$$p(x,t) - \tilde{p}(x,t) = \psi(x,t)^{\top}(\tilde{P}M\tilde{P} - M)\psi(x,t). \tag{39}$$

Let now consider every term separately.

$$\frac{\partial p(x,t)}{\partial t} - \frac{\partial \tilde{p}(x,t)}{\partial t} = 2\frac{\partial \psi(x,t)}{\partial t}^{\top} M\psi(x,t) - 2\frac{\partial \psi(x,t)}{\partial t}^{\top}\tilde{P}M\tilde{P}\psi(x,t)$$

$$= 2\frac{\partial \psi(x,t)}{\partial t}^{\top}(M - \tilde{P}M\tilde{P})\psi(x,t). \tag{40}$$

Similarly,

$$\frac{\partial p(x,t)}{\partial x_i} - \frac{\partial \tilde{p}(x,t)}{\partial x_i} = 2\frac{\partial \psi(x,t)}{\partial x_i}^{\top} M\psi(x,t) - 2\frac{\partial \psi(x,t)}{\partial x_i}^{\top}\tilde{P}M\tilde{P}\psi(x,t)$$

$$= 2\frac{\partial \psi(x,t)}{\partial x_i}^\top (M - \tilde{P}M\tilde{P})\psi(x,t). \tag{41}$$

Finally,

$$\frac{\partial^2 p(x,t)}{\partial x_i x_j} - \frac{\partial^2 \tilde{p}(x,t)}{\partial x_i x_j} = 2\psi_i'(x,t)^\top M\psi_j'(x,t) + 2\psi_{ji}''(x,t)^\top M\psi(x,t)$$
$$- 2\psi_i'(x,t)^\top \tilde{P}M\tilde{P}\psi_j'(x,t) - 2\psi_{ji}''(x,t)^\top \tilde{P}M\tilde{P}\psi(x,t)$$
$$= 2\psi_{ji}''(x,t)^\top (M - \tilde{P}M\tilde{P})\psi(x,t) + 2\psi_i'(x,t)^\top (M - \tilde{P}M\tilde{P})\psi_j'(x,t). \tag{42}$$

We have,

$$\tilde{P}M\tilde{P} - M = (I - \tilde{P})M(I - \tilde{P}) - M(I - \tilde{P}) - (I - \tilde{P})M.$$

Since $|a^\top ABAa| \leq \|Aa\|_{\mathcal{H}}^2 \|B\|$ and $|a^\top ABa| \leq \|Aa\|_{\mathcal{H}}\|B^{1/2}\|\|B^{1/2}a\|_{\mathcal{H}}$, for any $a$ in a Hilbert space $\mathcal{H}$ and for $A, B$ bounded symmetric linear operators with $B \in \mathbb{S}_+(\mathcal{H})$, by bounding the terms of the equation above, we have for any $x \in \mathbb{R}^d$,

$$|\psi(x,t)^\top (\tilde{P}M\tilde{P} - M)\psi(x,t)| \leq 2\|(I - \tilde{P})\psi(x,t)\|_{\mathcal{H}_\eta}\|M\|^{1/2}\|M^{1/2}\psi(x,t)\|_{\mathcal{H}_\eta}$$
$$+ \|(I - \tilde{P})\psi(x,t)\|_{\mathcal{H}_\eta}^2 \|M\|$$
$$= 2c_M^{1/2}f(x; M, \psi)^{1/2}\tilde{u}_1(x) + c_M\tilde{u}_1(x)^2, \tag{43}$$

where $c_M = \|M\|$ and we denoted by $\tilde{u}_1(x)$ the quantity $\tilde{u}_1(x) = \|(I - \tilde{P})\psi(x,t)\|_{\mathcal{H}_\eta}$ and we noted that $\|M^{1/2}\psi(x,t)\|_{\mathcal{H}_\eta}^2 = \psi(x,t)^\top M\psi(x,t) = f(x; M, \psi(x,t))$. Now,

$$\left|\frac{\partial \psi(x,t)}{\partial x_i}^\top (M - \tilde{P}M\tilde{P})\psi(x,t)\right| \leq \|(I - \tilde{P})\psi(x,t)\|_{\mathcal{H}_\eta}\|M\|\left\|(I - \tilde{P})\frac{\partial \psi(x,t)}{\partial x_i}\right\|_{\mathcal{H}_\eta}$$
$$+ \left\|(I - \tilde{P})\frac{\partial \psi(x,t)}{\partial x_i}\right\|_{\mathcal{H}_\eta}\|M\|^{1/2}\|M^{1/2}\psi(x,t)\|_{\mathcal{H}_\eta} + \left\|(I - \tilde{P})\frac{\partial \psi(x,t)}{\partial x_i}\right\|_{\mathcal{H}_\eta}\|M\|^{1/2}\left\|M^{1/2}\frac{\partial \psi(x,t)}{\partial x_i}\right\|_{\mathcal{H}_\eta}$$
$$= c_M\tilde{u}_1(x)\tilde{u}_{x_i}(x) + c_M^{1/2}f(x; M, \psi)^{1/2}\tilde{u}_{x_i}(x) + c_M^{1/2}\tilde{f}_{x_i}(x; M, \psi)^{1/2}\tilde{u}_1(x), \tag{44}$$

where $\tilde{f}_{x_i}(x; M, \psi) = \frac{\partial \psi(x,t)}{\partial x_i}^\top M\frac{\partial \psi(x,t)}{\partial x_i}$ and $\tilde{u}_{x_i}(x) = \left\|(I - \tilde{P})\frac{\partial \psi(x,t)}{\partial x_i}\right\|_{\mathcal{H}_\eta}$.

Following similar steps, we get

$$\left|\frac{\partial \psi(x,t)}{\partial t}^\top (M - \tilde{P}M\tilde{P})\psi(x,t)\right| \leq c_M\tilde{u}_1(x)\tilde{u}_t(x) + c_M^{1/2}f(x; M, \psi)^{1/2}\tilde{u}_t(x)$$
$$+ c_M^{1/2}\tilde{f}_t(x; M, \psi)^{1/2}\tilde{u}_1(x), \tag{45}$$

where $\tilde{f}_t(x; M, \psi) = \frac{\partial \psi(x,t)}{\partial t}^\top M\frac{\partial \psi(x,t)}{\partial t}$ and $\tilde{u}_t(x) = \left\|(I - \tilde{P})\frac{\partial \psi(x,t)}{\partial t}\right\|_{\mathcal{H}_\eta}$.

$$\left|\psi_{ji}''(x,t)^\top (M - \tilde{P}M\tilde{P})\psi(x,t)\right| \leq c_M\tilde{u}_1(x)\tilde{u}_{x_j x_i}(x) + c_M^{1/2}f(x; M, \psi)^{1/2}\tilde{u}_{x_j x_i}(x)$$
$$+ c_M^{1/2}\tilde{f}_{x_j x_i}(x; M, \psi)^{1/2}\tilde{u}_1(x), \tag{46}$$

where $\tilde{f}_{x_j x_i}(x; M, \psi) = \frac{\partial^2 \psi(x,t)}{\partial x_j x_i}^\top M\frac{\partial^2 \psi(x,t)}{\partial x_j x_i}$ and $\tilde{u}_{x_j x_i}(x) = \left\|(I - \tilde{P})\frac{\partial^2 \psi(x,t)}{\partial x_j x_i}\right\|_{\mathcal{H}_\eta}$.

And,

$$\psi_i'(x,t)^\top (M - \tilde{P}M\tilde{P})\psi_j'(x,t) \leq c_M\tilde{u}_{x_j}(x)\tilde{u}_{x_i}(x) + c_M^{1/2}f_{x_i}(x; M, \psi)^{1/2}\tilde{u}_{x_j}(x)$$
$$+ c_M^{1/2}\tilde{f}_{x_j}(x; M, \psi)^{1/2}\tilde{u}_{x_i}(x). \tag{47}$$

Hence, combining everything together, we have:

$$\left|\frac{\partial p(x,t)}{\partial t} - \frac{\partial \tilde{p}(x,t)}{\partial t} + \sum_{i=i}^d \frac{\partial(\mu_i(x,t)p(x,t))}{\partial x_i} - \frac{\partial(\mu_i(x,t)\tilde{p}(x,t))}{\partial x_i} - \sum_{i=1}^d\sum_{j=1}^d D_{ij}\frac{\partial^2}{\partial x_i\partial x_j}p(x,t) - D_{ij}\frac{\partial^2}{\partial x_i\partial x_j}\tilde{p}(x,t)\right|$$

$$\leq \left| \frac{\partial p(x,t)}{\partial t} - \frac{\partial \tilde{p}(x,t)}{\partial t} \right| + \sum_{i=1}^{d} \left| \mu_i(x,t)\frac{\partial p(x,t)}{\partial x_i} - \mu_i(x,t)\frac{\partial \tilde{p}(x,t)}{\partial x_i} \right| + \sum_{i=1}^{d} \left| \frac{\partial \mu_i(x,t)}{\partial x_i}p(x,t) - \frac{\partial \mu_i(x,t)}{\partial x_i}\tilde{p}(x,t) \right|$$

$$+ \left| \sum_{i=1}^{d}\sum_{j=1}^{d} D_{ij}\frac{\partial^2}{\partial x_i \partial x_j}p(x,t) - D_{ij}\frac{\partial^2}{\partial x_i \partial x_j}\tilde{p}(x,t) \right|$$

$$\leq \left| \frac{\partial p(x,t)}{\partial t} - \frac{\partial \tilde{p}(x,t)}{\partial t} \right| + R_\mu \sum_{i=1}^{d} \left| \frac{\partial p(x,t)}{\partial x_i} - \frac{\partial \tilde{p}(x,t)}{\partial x_i} \right| + dR_{\mu_p}(p(x,t) - \tilde{p}(x,t))$$

$$+ R_d \sum_{i=1}^{d}\sum_{j=1}^{d} \left| \frac{\partial^2}{\partial x_i \partial x_j}p(x,t) - \frac{\partial^2}{\partial x_i \partial x_j}\tilde{p}(x,t) \right|$$

$$= 2\left| \frac{\partial \psi(x,t)}{\partial t}^\top (M - \tilde{P}M\tilde{P})\psi(x,t) \right| + 2R_\mu \sum_{i=1}^{d} \left| \frac{\partial \psi(x,t)}{\partial x_i}^\top (M - \tilde{P}M\tilde{P})\psi(x,t) \right| + dR_{\mu_p}|\psi(x,t)^\top (M - \tilde{P}M\tilde{P})\psi(x,t)|$$

$$+ 2R_d|\psi''_{ji}(x,t)^\top (M - \tilde{P}M\tilde{P})\psi(x,t) + \psi'_i(x,t)^\top (M - \tilde{P}M\tilde{P})\psi'_j(x,t)|$$

$$\leq 2\left| \frac{\partial \psi(x,t)}{\partial t}^\top (M - \tilde{P}M\tilde{P})\psi(x,t) \right| + 2R_\mu \sum_{i=1}^{d} \left| \frac{\partial \psi(x,t)}{\partial x_i}^\top (M - \tilde{P}M\tilde{P})\psi(x,t) \right| + dR_{\mu_p}|\psi(x,t)^\top (M - \tilde{P}M\tilde{P})\psi(x,t)|$$

$$+ 2R_d|\psi''_{ji}(x,t)^\top (M - \tilde{P}M\tilde{P})\psi(x,t)| + 2R_d|\psi'_i(x,t)^\top (M - \tilde{P}M\tilde{P})\psi'_j(x,t)|.$$

Now, combining everything together, we get

$$\left| \frac{\partial p(x,t)}{\partial t} - \frac{\partial \tilde{p}(x,t)}{\partial t} + \sum_{i=i}^{d} \frac{\partial(\mu_i(x,t)p(x,t))}{\partial x_i} - \frac{\partial(\mu_i(x,t)\tilde{p}(x,t))}{\partial x_i} - \sum_{i=1}^{d}\sum_{j=1}^{d} D_{ij}\frac{\partial^2}{\partial x_i \partial x_j}p(x,t) - D_{ij}\frac{\partial^2}{\partial x_i \partial x_j}\tilde{p}(x,t) \right|$$

$$\leq 2c_M\tilde{u}_1(x)\tilde{u}_t(x) + 2c_M^{1/2}f(x;M,\psi)^{1/2}\tilde{u}_t(x) + 2c_M^{1/2}\tilde{f}_t(x;M,\psi)^{1/2}\tilde{u}_1(x) + dR_{\mu_p}(2c_M^{1/2}f(x;M,\psi)^{1/2}\tilde{u}_1(x) + c_M\tilde{u}_1(x)^2)$$

$$+ 2R_\mu \sum_{i=1}^{d}[c_M\tilde{u}_1(x)\tilde{u}_{x_i}(x) + c_M^{1/2}f(x;M,\psi)^{1/2}\tilde{u}_{x_i}(x) + c_M^{1/2}\tilde{f}_{x_i}(x;M,\psi)^{1/2}\tilde{u}_1(x)]$$

$$+ 2R_d \sum_{i=1}^{d}\sum_{j=1}^{d}[c_M\tilde{u}_1(x)\tilde{u}_{x_j x_i}(x) + c_M^{1/2}f(x;M,\psi)^{1/2}\tilde{u}_{x_j x_i}(x) + c_M^{1/2}\tilde{f}_{x_j x_i}(x;M,\psi)^{1/2}\tilde{u}_1(x)]$$

$$+ 2R_d \sum_{i=1}^{d}\sum_{j=1}^{d}[c_M\tilde{u}_{x_j}(x)\tilde{u}_{x_i}(x) + c_M^{1/2}f_{x_i}(x;M,\psi)^{1/2}\tilde{u}_{x_j}(x) + c_M^{1/2}\tilde{f}_{x_j}(x;M,\psi)^{1/2}\tilde{u}_{x_i}(x)].$$

Now from theorems 6 and 7, we have that when the fill distance $h < \frac{\sigma}{C'}$ with $\sigma = \min(R, \frac{1}{\tau})$, then

$$\|\tilde{u}_1\|_{L^\infty(\mathcal{X})} \leq Cq_\eta e^{-\frac{c\sigma}{h}\log\frac{c\sigma}{h}}, \ \|\tilde{u}_t\|_{L^\infty(\mathcal{X})} \leq Cq_\eta e^{-\frac{c\sigma}{h}\log\frac{c\sigma}{h}}, \|\tilde{u}_{x_i}\|_{L^\infty(\mathcal{X})} \leq Cq_\eta e^{-\frac{c\sigma}{h}\log\frac{c\sigma}{h}}$$

(48)

$$\text{and } \|\tilde{u}_{x_i x_j}\|_{L^\infty(\mathcal{X})} \leq Cq_\eta e^{-\frac{c\sigma}{h}\log\frac{c\sigma}{h}}$$

for all $i \in \{1,\cdots,d\}, j \in \{1,\cdots,d\}$ and some constant $C, c, C'$ depending only on $d$. Hence, we have

$$\left| \frac{\partial(p(x,t) - \tilde{p}(x,t))}{\partial t} + \sum_{i=i}^{d} \frac{\partial(\mu_i(x,t)(p(x,t) - \tilde{p}(x,t)))}{\partial x_i} - \sum_{i=1}^{d}\sum_{j=1}^{d} D_{ij}\frac{\partial^2}{\partial x_i \partial x_j}(p(x,t) - \tilde{p}(x,t)) \right|$$

$$\leq (2 + dR_\mu + 2dR_\mu + 4d^2 R_d)\|M\|C^2 q_\eta^2 e^{\frac{-2c\sigma}{h}\log\frac{c\sigma}{h}} + (2 + dR_{\mu_p} + 2dR_\mu + 2d^2 R_d)\sqrt{\|M\|p(x,t)}\, Cq_\eta e^{-\frac{c\sigma}{h}\log\frac{c\sigma}{h}}$$

$$+ 2Cq_\eta e^{-\frac{c\sigma}{h}\log\frac{c\sigma}{h}}\sqrt{\|M\|\tilde{f}_t(x;M,\psi)} + (2R_\mu + 4dR_d)Cq_\eta e^{-\frac{c\sigma}{h}\log\frac{c\sigma}{h}} \sum_{i=1}^{d} \sqrt{\|M\|\tilde{f}_{x_i}(x;M,\psi)}$$

$$+ 2R_d Cq_\eta e^{-\frac{c\sigma}{h}\log\frac{c\sigma}{h}} \sum_{i=1}^{d}\sum_{j=1}^{d} \sqrt{\|M\|\tilde{f}_{x_j x_i}(x;M,\psi)}.$$

$\square$

## A.6  Final Approximation Bound (Proof of Theorem 2)

*Proof.* From the previous result in theorem 1, we know that there exists an $M_\varepsilon \in \mathbb{S}_+(\mathcal{H}_X \otimes \mathcal{H}_T)$ with $\text{rank}(M_\varepsilon) \leq q$, such that for the representation $\hat{p}(x,t) = \phi_{X,T}^\top M_\varepsilon \phi_{X,T}$, following holds under

assumption 2 on the coefficients of the fractional FPE,

$$\left\| \frac{\partial \hat{p}(x,t)}{\partial t} + \sum_{i=1}^{d} \frac{\partial}{\partial x_i}(\mu_i(x,t)\hat{p}(x,t)) - \sum_{i=1}^{d}\sum_{j=1}^{d} D_{ij}\frac{\partial^2}{\partial x_i \partial x_j}\hat{p}(x,t) \right\|_{L^2(\tilde{x})} = O(\varepsilon), \qquad (49)$$

for an appropriate choice of parameters. We also know that

$$\frac{\partial p^\star(x,t)}{\partial t} + \sum_{i=1}^{d} \frac{\partial}{\partial x_i}(\mu_i(x,t)p^\star(x,t)) - \sum_{i=1}^{d}\sum_{j=1}^{d} D_{ij}\frac{\partial^2}{\partial x_i \partial x_j}p^\star(x,t) = 0. \qquad (50)$$

We know that, for given $A_m$, $\tilde{p}(x,t) = \psi(x,t)^\top \tilde{P} \mathsf{M}_\varepsilon \tilde{P}\psi(x,t)^\top$ where $\tilde{P}$ is the projection operator defined earlier. From the results in theorem 8, we have

$$\left| \frac{\partial(\hat{p}(x,t) - \tilde{p}(x,t))}{\partial t} + \sum_{i=i}^{d} \frac{\partial(\mu_i(x,t)(\hat{p}(x,t) - \tilde{p}(x,t)))}{\partial x_i} - \sum_{i=1}^{d}\sum_{j=1}^{d} D_{ij}\frac{\partial^2}{\partial x_i \partial x_j}(\hat{p}(x,t) - \tilde{p}(x,t)) \right|$$

$$\leq (2 + dR_\mu + 2dR_\mu + 4d^2 R_d)\|\mathsf{M}_\varepsilon\|C^2 q_\eta^2 e^{\frac{-2c\sigma}{h}\log\frac{c\sigma}{h}} + (2 + dR_{\mu_p} + 2dR_\mu + 2d^2 R_d)\sqrt{\|\mathsf{M}_\varepsilon\|p(x,t)}\, Cq_\eta e^{-\frac{c\sigma}{h}\log\frac{c\sigma}{h}}$$

$$+ 2Cq_\eta e^{-\frac{c\sigma}{h}\log\frac{c\sigma}{h}}\sqrt{\|\mathsf{M}_\varepsilon\|\tilde{f}_t(x;M,\psi)} + (2R_\mu + 4dR_d)Cq_\eta e^{-\frac{c\sigma}{h}\log\frac{c\sigma}{h}}\sum_{i=1}^{d}\sqrt{\|\mathsf{M}_\varepsilon\|\tilde{f}_{x_i}(x;M,\psi)}$$

$$+ 2R_d Cq_\eta e^{-\frac{c\sigma}{h}\log\frac{c\sigma}{h}}\sum_{i=1}^{d}\sum_{j=1}^{d}\sqrt{\|\mathsf{M}_\varepsilon\|\tilde{f}_{x_j x_i}(x;M_\varepsilon,\psi)}.$$

It is clear that some big constant $Q$, $\tilde{f}_{x_j x_i}(x;\mathsf{M}_\varepsilon,\eta) \leq Q\|\mathsf{M}_\varepsilon\|$, for all $i,j \in \{1,\cdots,q\}$. Similarly, $\tilde{f}_{x_i}(x;\mathsf{M}_\varepsilon,\eta) \leq Q\|\mathsf{M}_\varepsilon\|$ for all $i \in \{1,\cdots,q\}$ and $\tilde{f}_t(x;\mathsf{M}_\varepsilon,\eta) \leq Q\|\mathsf{M}_\varepsilon\|$. Hence, we have,

$$\left| \frac{\partial(\hat{p}(x,t) - \tilde{p}(x,t))}{\partial t} + \sum_{i=i}^{d} \frac{\partial(\mu_i(x,t)(\hat{p}(x,t) - \tilde{p}(x,t)))}{\partial x_i} - \sum_{i=1}^{d}\sum_{j=1}^{d} D_{ij}\frac{\partial^2}{\partial x_i \partial x_j}(\hat{p}(x,t) - \tilde{p}(x,t)) \right|$$

$$\leq (2 + dR_\mu + 2dR_\mu + 4d^2 R_d)\|\mathsf{M}_\varepsilon\|C^2 q_\eta^2 e^{\frac{-2c\sigma}{h}\log\frac{c\sigma}{h}} + (2 + dR_{\mu_p} + 2dR_\mu + 2d^2 R_d)\sqrt{\|\mathsf{M}_\varepsilon\|p(x,t)}\, Cq_\eta e^{-\frac{c\sigma}{h}\log\frac{c\sigma}{h}}$$

$$+ 2QCq_\eta e^{-\frac{c\sigma}{h}\log\frac{c\sigma}{h}}\|\mathsf{M}_\varepsilon\| + (2QR_\mu d + 4Qd^2 R_d)Cq_\eta e^{-\frac{c\sigma}{h}\log\frac{c\sigma}{h}}\|\mathsf{M}_\varepsilon\| + 2QR_d d^2 Cq_\eta e^{-\frac{c\sigma}{h}\log\frac{c\sigma}{h}}\|\mathsf{M}_\varepsilon\|$$

$$\leq (2 + dR_\mu + 2dR_\mu + 4d^2 R_d)\|\mathsf{M}_\varepsilon\|C^2 q_\eta^2 e^{\frac{-2c\sigma}{h}\log\frac{c\sigma}{h}} + (2 + dR_{\mu_p} + 2dR_\mu + 2d^2 R_d)\sqrt{\|\mathsf{M}_\varepsilon\|p(x,t)}\, Cq_\eta e^{-\frac{c\sigma}{h}\log\frac{c\sigma}{h}}$$

$$+ Q(2C + 2R_\mu d + 6Qd^2 R_d)Cq_\eta e^{-\frac{c\sigma}{h}\log\frac{c\sigma}{h}}\|\mathsf{M}_\varepsilon\|.$$

Now, let us compute $\|\mathsf{M}_\varepsilon\|$. We have

$$\|\mathsf{M}_\varepsilon\| \leq Tr(M_\varepsilon) = \hat{C}\tau^{(d+1)/2}\left(1 + \varepsilon^{\frac{2\beta}{\beta-2}}e^{\frac{\tilde{C}'}{\tau}\varepsilon^{-\frac{2}{\beta-2}}}\right).$$

Let us choose, $\tau = \frac{\tilde{C}'\varepsilon^{-\frac{2}{\beta-2}}}{\frac{2\beta}{\beta-2}\log(\frac{1+R}{\varepsilon})}$. Hence,

$$\|M_\varepsilon\| \leq C_3(1+R)^{\frac{2\beta}{\beta-2}}\varepsilon^{-\frac{d+1}{\beta-2}}.$$

Now,

$$\left\| \frac{\partial(\hat{p}(x,t) - \tilde{p}(x,t))}{\partial t} + \sum_{i=i}^{d} \frac{\partial(\mu_i(x,t)(\hat{p}(x,t) - \tilde{p}(x,t)))}{\partial x_i} - \sum_{i=1}^{d}\sum_{j=1}^{d} D_{ij}\frac{\partial^2}{\partial x_i \partial x_j}(\hat{p}(x,t) - \tilde{p}(x,t)) \right\|_{L^2(\tilde{x})}$$

$$\leq (2 + 3dR_\mu + 4d^2 R_d)\|\mathsf{M}_\varepsilon\|C^2 q_\eta^2 e^{\frac{-2c\sigma}{h}\log\frac{c\sigma}{h}}R^{(d+1)/2} + Q(2C + 2R_\mu d + 6Qd^2 R_d)Cq_\eta e^{-\frac{c\sigma}{h}\log\frac{c\sigma}{h}}\|\mathsf{M}_\varepsilon\|R^{(d+1)/2}$$

$$+ (2 + dR_{\mu_p} + 2dR_\mu + 2d^2 R_d)\sqrt{\|\mathsf{M}_\varepsilon\|p(x,t)}\, Cq_\eta e^{-\frac{c\sigma}{h}\log\frac{c\sigma}{h}}$$

$$= \underbrace{(2 + 3dR_\mu + 4d^2 R_d)}_{:=P_1}\|\mathsf{M}_\varepsilon\|C^2 q_\eta^2 e^{\frac{-2c\sigma}{h}\log\frac{c\sigma}{h}}R^{(d+1)/2} + \underbrace{Q(2C + 2R_\mu d + 6Qd^2 R_d)}_{:=P_2}Cq_\eta e^{-\frac{c\sigma}{h}\log\frac{c\sigma}{h}}\|\mathsf{M}_\varepsilon\|R^{(d+1)/2}$$

$$+ \underbrace{(2 + 3dR_\mu + 2d^2 R_d)}_{:=P_3}\|\sqrt{\|\mathsf{M}_\varepsilon\|p(x,t)}\|_{L^2(\tilde{x})}\, Cq_\eta e^{-\frac{c\sigma}{h}\log\frac{c\sigma}{h}}$$

$$= P_1\|\mathsf{M}_\varepsilon\|C^2 q_\eta^2 e^{\frac{-2c\sigma}{h}\log\frac{c\sigma}{h}}R^{(d+1)/2} + P_2 Cq_\eta e^{-\frac{c\sigma}{h}\log\frac{c\sigma}{h}}\|\mathsf{M}_\varepsilon\|R^{(d+1)/2}$$

$$+ P_3\|\mathsf{M}_\varepsilon\|^{1/2}Cq_\eta e^{-\frac{c\sigma}{h}\log\frac{c\sigma}{h}}\|\sqrt{p(x,t)}\|_{L^2(\tilde{x})}. \qquad (51)$$

Then, note that $\|p^{1/2}\|_{L^2(\mathcal{X})} = \|p\|_{L^1(\mathcal{X})}^{1/2}$. Hence, using the argument in [RC21], we have

$$\|p^{1/2}\|_{L^2(\tilde{\mathcal{X}})} \le 2R.$$

Term $R$ comes from the fact that for each time instance $p(x,t)$ is a density and $t \in (0 < R)$. Hence,

$$\left\| \frac{\partial(\hat{p}(x,t) - \tilde{p}(x,t))}{\partial t} + \sum_{i=i}^{d} \frac{\partial(\mu_i(x,t)(\hat{p}(x,t) - \tilde{p}(x,t)))}{\partial x_i} - \sum_{i=1}^{d}\sum_{j=1}^{d} D_{ij} \frac{\partial^2}{\partial x_i \partial x_j}(\hat{p}(x,t) - \tilde{p}(x,t)) \right\|_{L^2(\tilde{\mathcal{X}})}$$

$$\le P_1 \|\mathsf{M}_\varepsilon\| C^2 q_\eta^2 e^{\frac{-2c\sigma}{h} \log \frac{c\sigma}{h}} R^{(d+1)/2} + P_2 C q_\eta e^{-\frac{c\sigma}{h}\log\frac{c\sigma}{h}} \|\mathsf{M}_\varepsilon\| R^{(d+1)/2}$$

$$+ 2P_3 R \|\mathsf{M}_\varepsilon\|^{1/2} C q_\eta e^{-\frac{c\sigma}{h}\log\frac{c\sigma}{h}}$$

$$\le P_1 C_3 (1+R)^{\frac{2\beta}{\beta-2}} \varepsilon^{-\frac{d+1}{\beta-2}} C^2 q_\eta^2 e^{\frac{-2c\sigma}{h}\log\frac{c\sigma}{h}} R^{(d+1)/2} + P_2 C q_\eta e^{-\frac{c\sigma}{h}\log\frac{c\sigma}{h}} C_3 (1+R)^{\frac{2\beta}{\beta-2}} \varepsilon^{-\frac{d+1}{\beta-2}} R^{(d+1)/2}$$

$$+ 2P_3 R C_3^{1/2}(1+R)^{\frac{\beta}{\beta-2}} \varepsilon^{-\frac{d+1}{2(\beta-2)}} C q_\eta e^{-\frac{c\sigma}{h}\log\frac{c\sigma}{h}}.$$

By choosing $h = c\sigma/s$ with $s = \max(C', (1 + \frac{d+1}{2(\beta-2)})\log\frac{1}{\varepsilon} + (1 + \frac{d}{2})\log(1+R) + \log(\hat{C}) + e)$, for some big enough $\hat{C}$. Since $s \ge e$, then $\log s \ge 1$, so

$$Ce^{-\frac{c\sigma}{h}\log\frac{c\sigma}{h}} = Ce^{-s\log s} \le Ce^{-s} \le C'(1+R)^{-d/2}\varepsilon^{1+\frac{d+1}{2(\beta-2)}}.$$

Hence, we have,

$$\left\| \frac{\partial(\hat{p}(x,t) - \tilde{p}(x,t))}{\partial t} + \sum_{i=i}^{d} \frac{\partial(\mu_i(x,t)(\hat{p}(x,t) - \tilde{p}(x,t)))}{\partial x_i} - \sum_{i=1}^{d}\sum_{j=1}^{d} D_{ij} \frac{\partial^2}{\partial x_i \partial x_j}(\hat{p}(x,t) - \tilde{p}(x,t)) \right\|_{L^2(\tilde{\mathcal{X}})}$$

$$= O(\varepsilon).$$

Combining evrything together and by the use of triangles inequality, we have

$$\left\| \frac{\partial \hat{p}(x,t)}{\partial t} + \sum_{i=1}^{d} \frac{\partial}{\partial x_i}(\mu_i(x,t)\hat{p}(x,t)) - \sum_{i=1}^{d}\sum_{j=1}^{d} D_{ij} \frac{\partial^2}{\partial x_i \partial x_j}\hat{p}(x,t) \right\|_{L^2(\tilde{\mathcal{X}})} = O(\varepsilon).$$

To conclude we recall the fact that $\tilde{x}_1, \ldots, \tilde{x}_m$ is a $h$-covering of $T$, guarantees that the number of centers $m$ in the covering satisfies

$$m \le (1 + \frac{2R\sqrt{d+1}}{h})^{d+1}.$$

Then, since $h \ge c\sigma/(C_4 \log \frac{C_5 \log(1+R)}{\varepsilon})$ with $C_4 = 1 + d/\min(2(\beta-2), 2)$ and $C_5 = (\hat{C})^{1/C_4}$, and since $\sigma = \min(R, 1/\sqrt{\tau})$, then $R/\sigma = \max(1, R\sqrt{\tau}) \le 1 + \sqrt{\tilde{C}'}\varepsilon^{-1/(\beta-2)}(\log\frac{1+R}{\varepsilon})^{-1/2}$, so after final calculation, we have

$$m^{\frac{1}{d+1}} \le 1 + 2R\sqrt{d+1}/h \le C_8 + C_9 \log\frac{1+R}{\varepsilon} + C_{10}\varepsilon^{-\frac{1}{\beta-2}}\left(\log\frac{1+R}{\varepsilon}\right)^{1/2},$$

for some constants $C_8$, $C_9$ and $C_{10}$ independent of $\varepsilon$. $\qquad\square$

### A.7 Proof of Corollary 1

*Proof of Corollary 1.* Let $p^\star$ be the probability corresponding to the solution of the Fokker-Planck equation with drift $\mu$ and $\hat{p}$ be the conditional probability corresponding to the solution of the Fokker-Planck equation with drift $\hat{\mu}$. And denote by $\hat{p}_\gamma(x,t)$ the $m$ dimensional PSD model obtained by applying the construction of Thm. 2 to the Fokker-Planck equation defined in terms of $\hat{\mu}$. Hence, from the direct application of the result in [BRS16, Theorem 1.1] since $\|\mu - \hat{\mu}\|_{L^2} = O(\varepsilon)$ we have,

$$\|p^\star - \hat{p}\|_{L^2(T\times X)} = O(\varepsilon). \tag{52}$$

From, theorem 2, we have,

$$\|\hat{p} - \hat{p}_\gamma\|_{L^2(T\times X)} = O(\varepsilon).$$

Finally due to proposition 2, for any PSD model $f$ there exist an algorithm that samples i.i.d. from the probability $p_f$ satisfying

$$\mathbb{W}_1(f, p_f) = O(\varepsilon).$$

Now note that for any $t$, the function $\hat{p}_\gamma(t, \cdot)$ is still a PSD model (see [RC21]). Then, for any $t \in (0, T)$, we have that there exists probabilities $\hat{p}_{t,\alpha}$ such that

$$\mathbb{W}_1(\hat{p}_\gamma(t, \cdot), \hat{p}_{t,\alpha}(\cdot)) = O(\varepsilon).$$

Then, since $\mathcal{W}_1$ is a distance and is bounded by the $L^2$ distance since we are on a bounded domain, there exists a $C$ depending on the diameter and the volume of the domain, such that for any $t$

$$\mathbb{W}_1(p^\star(t, \cdot), \hat{p}_{t,\alpha}(\cdot))^2 \leq C(\|p^\star(t, \cdot) - \hat{p}^\star(t, \cdot)\|_{L^2(X)}^2 + \|\hat{p}(t, \cdot) - \hat{p}_\gamma(t, \cdot)\|_{L^2(X)}^2 + \mathbb{W}_1(\hat{p}_\gamma, p_\alpha)^2)$$

Then, by taking the expectation over $t \in (0, T)$ we have

$$\mathbb{E}_t \, \mathbb{W}_1(p^\star(t, \cdot), \hat{p}_{t,\alpha}(\cdot))^2 \leq C(\|p^\star - \hat{p}^\star\|_{L^2(T \times X)}^2 + \|\hat{p}(t, \cdot) - \hat{p}_\gamma(t, \cdot)\|_{L^2(T \times X)}^2 + \mathbb{W}_1(\hat{p}_\gamma, p_\alpha)^2) = O(\varepsilon^2)$$

$\square$

# B    Estimation of Fractional Laplacian Operator on a Probability Density

Before we go into the details of the estimation, let us first recall bochner's theorem.

**Theorem 9** ([Rud17]). *A continuous kernel $k(x, y) = k(x - y)$ on $\mathbb{R}^d$ is positive definite if and only if $k(\delta)$ is the Fourier transform of a non-negative measure.*

*Proof of Theorem 3.*  In our psd model approximation, we have

$$p(x) = f(x; A, X, \eta) = \sum_{i,j=1}^m A_{ij} k(x_i, x) k(x_j, x).$$

We will first prove the part (i) of the result. Now, we know that, fractional laplacian operator is defined as,

$$\mathcal{F}[(-\Delta)^s f](\xi) = \|\xi\|^{2s} \mathcal{F}[f](\xi)$$

$$= \|\xi\|^{2s} \sum_{i,j=1}^m A_{ij} \mathcal{F}[k(x_i, \cdot) k(x_j, \cdot)](\xi)$$

$$= \|\xi\|^{2s} \sum_{i,j}^m A_{ij} \mathcal{F}[k(x_i, \cdot)](\xi) * \mathcal{F}[k(x_j, \cdot)](\xi).$$

Now that, $k$ is a gaussian kernel with bandwidth $\eta$. Hence,

$$k(y, x) = e^{-\eta \|y - x\|_2^2}$$

$$\mathcal{F}[k(y, \cdot)](\xi) = c e^{j \xi^\top y - \frac{1}{4\eta} \|\xi\|_2^2}$$

$$= c e^{-\frac{1}{4\eta}\left(\|\xi\|_2^2 - 4j\eta \xi^\top y - 4\eta^2 \|y\|^2 + 4\eta^2 \|y\|^2\right)}$$

$$= c e^{-\eta \|y\|^2} \cdot e^{-\frac{1}{4\eta} \|\xi - 2j\eta y\|^2},$$

for some $c > 0$ which can be computed in closed form. Now, we know that convolution of two multivariate gaussian functions are still gaussian. Hence,

$$\mathcal{F}[k(x_i, \cdot)](\xi) * \mathcal{F}[k(x_j, \cdot)](\xi) = c' e^{-\eta(\|x_i\|^2 + \|x_j\|^2)} e^{-\frac{1}{8\eta} \|\xi - 2j\eta(x_i + x_j)\|^2},$$

for some constant $c' > 0$ which again can be computed in the closed form. Hence, we have

$$\mathcal{F}[(-\Delta)^s f](\xi) = \|\xi\|^{2s} \sum_{i,j=1}^m A_{ij} \mathcal{F}[k(x_i, \cdot)](\xi) * \mathcal{F}[k(x_j, \cdot)](\xi)$$

$$= c' \sum_{i,j=1}^m A_{ij} \|\xi\|^{2s} e^{-\eta(\|x_i\|^2 + \|x_j\|^2)} e^{-\frac{1}{8\eta} \|\xi - 2j\eta(x_i + x_j)\|^2}. \tag{53}$$

Finally we have,

$$(-\Delta)^s f(x) = C' \sum_{i,j=1}^{m} A_{ij} \int \|\xi\|^{2s} e^{-\eta(\|x_i\|^2 + \|x_j\|^2)} e^{-\frac{1}{8\eta}\|\xi - 2j\eta(x_i+x_j)\|^2} e^{-j\xi^\top x} \, d\xi$$

$$= C' \sum_{i,j=1}^{m} A_{ij} e^{-\eta(\|x_i\|^2 + \|x_j\|^2)} \int \|\xi\|^{2s} e^{-j\xi^\top x} \cdot e^{-\frac{1}{8\eta}\|\xi - 2j\eta(x_i+x_j)\|^2} \, d\xi.$$

By inspecting the above equation carefully, we can see that after appropriate scaling,

$$(-\Delta)^s f(x) = C \sum_{i,j=1}^{m} A_{ij} e^{-\eta(\|x_i\|^2 + \|x_j\|^2)} \mathbb{E}_{\xi \sim \mathcal{N}(\mu, \Sigma)}[\|\xi\|^{2s} e^{-j\xi^\top x}] \tag{54}$$

where $\mu = 2j\eta(x_i + x_j)$ and $\Sigma = 4\eta I$. One can sample from $\mathcal{N}(\mu, \Sigma)$ to estimate the fractional laplacian operator for PSD model.

Now, we will prove part (ii) of the theorem statement. From the bochner's theorem, we have,

$$k(x - y) = \int_{\mathbb{R}^d} q(\omega) e^{j\omega^\top(x-y)} \, d\omega,$$

where $q$ is the probability density. In our psd model approximation, we have

$$p(x) = f(x; A, X, \eta) = \sum_{i,j=1}^{m} A_{ij} k(x_i, x) k(x_j, x)$$

$$= \sum_{i,j=1}^{m} A_{ij} \left( \int_{\mathbb{R}^d} e^{j\omega_\ell^\top(x_i - x)} q(\omega_\ell) \, d\omega_\ell \right) \left( \int_{\mathbb{R}^d} e^{j\omega_k^\top(x_j - x)} q(\omega_k) \, d\omega_k \right)$$

$$= \sum_{i,j=1}^{m} A_{ij} \int_{\mathbb{R}^d} \int_{\mathbb{R}^d} e^{j\omega_\ell^\top(x_i - x)} e^{j\omega_k^\top(x_j - x)} q(\omega_\ell) q(\omega_k) \, d\omega_\ell d\omega_k.$$

Now, we know that, fractional laplacian operator is defined as,

$$\mathcal{F}[(-\Delta)^s f](\xi) = \|\xi\|^{2s} \mathcal{F}[f](\xi)$$

$$= \|\xi\|^{2s} \left( \int \left( \sum_{i,j=1}^{m} A_{ij} \int_{\mathbb{R}^d} \int_{\mathbb{R}^d} e^{j\omega_\ell^\top(x_i - x)} e^{j\omega_k^\top(x_j - x)} q(\omega_\ell) q(\omega_k) \, d\omega_\ell d\omega_k \right) e^{-j\xi^\top x} \, dx \right)$$

$$= \|\xi\|^{2s} \sum_{i,j=1}^{m} A_{ij} \int_{\mathbb{R}^d} \int_{\mathbb{R}^d} \left( \int_{\mathbb{R}^d} e^{j\omega_\ell^\top(x_i - x)} e^{j\omega_k^\top(x_j - x)} e^{-j\xi^\top x} \, dx \right) q(\omega_\ell) q(\omega_k) \, d\omega_\ell d\omega_k$$

$$= \|\xi\|^{2s} \sum_{i,j=1}^{m} A_{ij} \int_{\mathbb{R}^d} \int_{\mathbb{R}^d} \left( \int e^{j(\omega_\ell^\top x_i + \omega_k x_j)} \cdot e^{-jx^\top(\omega_\ell + \omega_k + \xi)} \, dx \right) q(\omega_\ell) q(\omega_k) \, d\omega_\ell d\omega_k$$

$$= \|\xi\|^{2s} \sum_{i,j=1}^{m} A_{ij} \int_{\mathbb{R}^d} \int_{\mathbb{R}^d} e^{j(\omega_\ell^\top x_i + \omega_k x_j)} \left( \int_{\mathbb{R}^d} e^{-jx^\top(\omega_\ell + \omega_k + \xi)} \, dx \right) q(\omega_\ell) q(\omega_k) \, d\omega_\ell d\omega_k$$

$$= \frac{1}{p^2} \sum_{i,j=1}^{m} A_{ij} \int_{\mathbb{R}^d} \int_{\mathbb{R}^d} e^{j(\omega_\ell^\top x_i + \omega_k x_j)} \|\xi\|^{2s} \delta(\omega_\ell + \omega_k + \xi) \, q(\omega_\ell) q(\omega_k) \, d\omega_\ell d\omega_k.$$

Hence,

$$(-\Delta)^s f(x) = \sum_{i,j=1}^{m} A_{ij} \int_{\mathbb{R}^d} \int_{\mathbb{R}^d} e^{j(\omega_\ell^\top x_i + \omega_k x_j)} \left( \int_{\mathbb{R}^d} \|\xi\|^{2s} \delta(\omega_\ell + \tilde{\omega}_k + \xi) e^{-jx^\top \xi} \, d\xi \right) q(\omega_\ell) q(\omega_k) \, d\omega_\ell d\omega_k$$

$$= \sum_{i,j=1}^{m} A_{ij} \int_{\mathbb{R}^d} \int_{\mathbb{R}^d} e^{j(\omega_\ell^\top x_i + \omega_k x_j)} \|\omega_\ell + \tilde{\omega}_k\|^{2s} e^{jx^\top(\omega_\ell + \tilde{\omega}_k)} \, q(\omega_\ell) q(\omega_k) \, d\omega_\ell d\omega_k$$

$$= \sum_{i,j=1}^{m} A_{ij} \int_{\mathbb{R}^d} \int_{\mathbb{R}^d} e^{j\omega_\ell^\top(x_i + x)} e^{j\omega_k^\top(x_j + x)} \|\omega_\ell + \tilde{\omega}_k\|^{2s} \, q(\omega_\ell) q(\omega_k) \, d\omega_\ell d\omega_k$$

$$= \sum_{i,j=1}^{m} A_{ij} \, \mathbb{E}[\|\omega_\ell + \omega_k\|^{2s} e^{j\omega_\ell^\top (x_i + x)} e^{j\omega_k^\top (x_j + x)}].$$

$\square$

## C  Existence of an appropriate PSD matrix $M_\varepsilon$ (Fractional Fokker-Planck Equation)

Let us redefine the mollifier function $g$ defined as in [RC21].

$$g(x) = \frac{2^{-d/2}}{V_d} \|x\|^{-d} J_{d/2}(2\pi\|x\|) J_{d/2}(4\pi\|x\|), \tag{55}$$

where $J_{d/2}$ is the Bessel function of the first kind of order $d/2$ and $V_d = \int_{\|x\|\le 1} dx = \frac{\pi^{d/2}}{\Gamma(d/2+1)}$. $g_v(x) = v^{-d} g(x/v)$. $\mathcal{F}[f](\cdot)$ denotes the fourier transform of $f$.

### C.1  Useful Results ( Fractional Fokker Planck Equation)

**Lemma 7.** *Consider the definition of $g_v(x) = v^{-d} g(x/v)$ where $g$ is defined in equation (13). Let $\beta > 2, q \in \mathbb{N}$. Let $f_1, \ldots, f_q \in W_2^\beta(\mathbb{R}^d) \cap L^\infty(\mathbb{R}^d)$ and the function $p^\star = \sum_{i=1}^q f_i^2$. Let $\varepsilon \in (0,1]$ and let $\eta \in \mathbb{R}_{++}^d$. Let $\phi_{X,T}$ be the feature map of the Gaussian kernel with bandwidth $\eta$ and let $\mathcal{H}_X \otimes \mathcal{H}_T$ be the associated RKHS. Then there exists $M_\varepsilon \in \mathbb{S}_+(\mathcal{H}_X \otimes \mathcal{H}_T)$ with $\mathrm{rank}(M_\varepsilon) \le q$, such that for the representation $\hat{p}(x,t) = \phi_{X,T}^\top M_\varepsilon \phi_{X,T}$, following holds for $v > 0$ and $k \in \{1, \cdots d\}$,*

$$\|(-\Delta)^s(p^\star - \hat{p})\|_{L^2(\mathbb{R}^d)} \le c_1 (2v)^{(\beta-s)} \sum_{i=1}^q C_{T_1}^{(i)} \|f_i\|_{W_2^\beta(\mathbb{R}^{d+1})}^2 + c_2(2v)^\beta \sum_{i=1}^q C_{T_2}^{(i)} \|f_i\|_{W_2^\beta(\mathbb{R}^{d+1})}^2,$$

*for some positive constants $c_1$ and $c_2$, where $C_{T_1}^{(i)} = \|\mathcal{F}[f_i]\|_{L^1(\mathbb{R}^{d+1})} + \|\mathcal{F}[f_i]\|_{L^\infty(\mathbb{R}^{d+1})} \|g\|_{L^1(\mathbb{R}^{d+1})}$ and $C_{T_2}^{(i)} = \big\|\|\omega\|^{2s} \mathcal{F}(f_i)(\omega)\big\|_{L^1(\mathbb{R}^{d+1})} + \big\|\|\omega\|^{2s} \mathcal{F}[f_i](\omega)\big\|_{L^1(\mathbb{R}^{d+1})} \|g\|_{L^1(\mathbb{R}^{d+1})}$ for $i \in \{1, \cdots, q\}$.*

*Proof.* As in [RC21], let us denote, $f_i \star g_v$ as $f_{i,v}$. We consider

$$M_\varepsilon = \sum_{i=1}^q f_{i,v} f_{i,v}^\top.$$

Hence,

$$\hat{p}(x,t) = f(x,t) = \phi_{X,T}(x,t)^\top M_\varepsilon \phi_{X,T}(x,t) = \sum_{i=1}^q f_{i,v}^2(x,t).$$

$$\|(-\Delta)^s(p^\star - \hat{p})\|_{L^2(\mathbb{R}^d)} = \|\mathcal{F}\left[(-\Delta)^s(p^\star - \hat{p})\right]\|_{L^2(\mathbb{R}^{d+1})}$$

$$= \|\|\omega\|^{2s} \mathcal{F}[p^\star - \hat{p}](\omega)\|_{L^2(\mathbb{R}^{d+1})} = \left\|\|\omega\|^{2s} \mathcal{F}\left[\sum_{i=1}^q f_i^2 - f_{i,v}^2\right](\omega)\right\|_{L^2(\mathbb{R}^{d+1})}$$

$$= \left\|\|\omega\|^{2s} \mathcal{F}\left[\sum_{i=1}^q (f_i - f_{i,v})(f_i + f_{i,v})\right](\omega)\right\|_{L^2(\mathbb{R}^{d+1})} = \left\|\|\omega\|^{2s} \sum_{i=1}^q \mathcal{F}\left[(f_i - f_{i,v})(f_i + f_{i,v})\right](\omega)\right\|_{L^2(\mathbb{R}^{d+1})}$$

$$\le \sum_{i=1}^q \left\|\|\omega\|^{2s} \mathcal{F}\left[(f_i - f_{i,v})(f_i + f_{i,v})\right](\omega)\right\|_{L^2(\mathbb{R}^{d+1})} = \sum_{i=1}^q \left\|\|\omega\|^{2s} \mathcal{F}\left[(f_i - f_{i,v})\right](\omega) \star \mathcal{F}\left[f_i + f_{i,v}\right](\omega)\right\|_{L^2(\mathbb{R}^{d+1})}.$$

Let us consider the function,

$$\|\omega\|^{2s} \underbrace{\mathcal{F}\left[(f_i - f_{i,v})\right](\omega)}_{:=m(\omega)} \star \underbrace{\mathcal{F}\left[f_i + f_{i,v}\right](\omega)}_{:=n(\omega)} = \int \|\omega\|^{2s} m(\omega - y) n(y) \, d\omega$$

$$= \int \|\omega - y + y\|^{2s} m(\omega - y) n(y) \, d\omega \le c_1 \int \|\omega - y\|^{2s} m(\omega - y) n(y) \, dy + c_2 \int \|y\|^{2s} m(\omega - y) n(y) \, dy$$

$$= c_1 [\|\omega\|^{2s} m(\omega)] * n(\omega) + c_2 m(\omega) * [\|\omega\|^{2s} n(\omega)].$$

Hence,

$$\|(-\Delta)^s (p^\star - \hat{p})\|_{L^2(\mathbb{R}^{d+1})} \le \underbrace{c_1 \|[\|\omega\|^{2s} m(\omega)] * n(\omega)\|_{L^2(\mathbb{R}^d)}}_{:= \mathrm{T}_1} + \underbrace{c_2 \|m(\omega) * [\|\omega\|^{2s} n(\omega)]\|_{L^2(\mathbb{R}^{d+1})}}_{:= \mathrm{T}_2}$$

Now, we can apply Young's convolution inequality on $\mathrm{T}_1$ and $\mathrm{T}_2$. Hence,

$$\mathrm{T}_1 \le c_1 \|\|\omega\|^{2s} m(\omega)\|_{L^2(\mathbb{R}^{d+1})} \|n(\omega)\|_{L^1(\mathbb{R}^{d+1})}$$

$$= c_1 \|\|\omega\|^{2s} m(\omega)\|_{L^2(\mathbb{R}^d)} \|\mathcal{F}[f_i + f_{i,v}]\|_{L^1(\mathbb{R}^{d+1})}$$

$$\le c_1 \|\|\omega\|^{2s} m(\omega)\|_{L^2(\mathbb{R}^{d+1})} \left[ \|\mathcal{F}[f_i]\|_{L^1(\mathbb{R}^{d+1})} + \|\mathcal{F}[f_{i,v}]\|_{L^1(\mathbb{R}^{d+1})} \right]$$

$$= c_1 \|\|\omega\|^{2s} m(\omega)\|_{L^2(\mathbb{R}^{d+1})} \left[ \|\mathcal{F}[f_i]\|_{L^1(\mathbb{R}^d)} + \|\mathcal{F}[f_i].\mathcal{F}[g_v]\|_{L^1(\mathbb{R}^{d+1})} \right]$$

$$\le c_1 \|\|\omega\|^{2s} m(\omega)\|_{L^2(\mathbb{R}^{d+1})} \left[ \|\mathcal{F}[f_i]\|_{L^1(\mathbb{R}^{d+1})} + \|\mathcal{F}[f_i]\|_{L^1(\mathbb{R}^{d+1})} \|\mathcal{F}[g_v]\|_{L^\infty(\mathbb{R}^{d+1})} \right].$$

In the last equation, we applied holder's inequality. From uniform continuity and the Riemann–Lebesgue lemma, we have

$$\|\mathcal{F}[g_v]\|_{L^\infty(\mathbb{R}^{d+1})} \le \|g_v\|_{L^1(\mathbb{R}^{d+1})} = \int g_v(vx) \, dx = \int t^{-d} |g(x/t)| dx = \int |g(x)| \, dx = \|g\|_{L^1(\mathbb{R}^{d+1})}.$$

Hence,

$$\mathrm{T}_1 \le c_1 \|\|\omega\|^{2s} m(\omega)\|_{L^2(\mathbb{R}^{d+1})} \left[ \|\mathcal{F}[f_i]\|_{L^1(\mathbb{R}^{d+1})} + \|\mathcal{F}[f_i]\|_{L^\infty(\mathbb{R}^{d+1})} \|g\|_{L^1(\mathbb{R}^{d+1})} \right]$$

Similarly,

$$\mathrm{T}_2 \le c_2 \|m(\omega)\|_{L^2(\mathbb{R}^d)} \|\|\omega\|^{2s} n(\omega)\|_{L^1(\mathbb{R}^{d+1})}$$

$$\le c_2 \|m(\omega)\|_{L^2(\mathbb{R}^{d+1})} \left[ \|\|\omega\|^{2s} \mathcal{F}(f_i)(\omega)\|_{L^1(\mathbb{R}^{d+1})} + \|\|\omega\|^{2s} \mathcal{F}(f_{i,v})(\omega)\|_{L^1(\mathbb{R}^{d+1})} \right]$$

$$= c_2 \|m(\omega)\|_{L^2(\mathbb{R}^d)} \left[ \|\|\omega\|^{2s} \mathcal{F}(f_i)(\omega)\|_{L^1(\mathbb{R}^{d+1})} + \|\|\omega\|^{2s} \mathcal{F}(f_i)(\omega) \mathcal{F}(g_v)(\omega)\|_{L^1(\mathbb{R}^{d+1})} \right].$$

In the above equation, we can apply holder's inequality to get,

$$\mathrm{T}_2 \le c_2 \|m(\omega)\|_{L^2(\mathbb{R}^{d+1})} \left[ \|\|\omega\|^{2s} \mathcal{F}(f_i)(\omega)\|_{L^1(\mathbb{R}^{d+1})} + \|\|\omega\|^{2s} \mathcal{F}(f_i)(\omega)\|_{L^1(\mathbb{R}^{d+1})} \|\mathcal{F}(g_v)(\omega)\|_{L^\infty(\mathbb{R}^{d+1})} \right]$$

$$\le c_2 \|m(\omega)\|_{L^2(\mathbb{R}^{d+1})} \left[ \|\|\omega\|^{2s} \mathcal{F}(f_i)(\omega)\|_{L^1(\mathbb{R}^{d+1})} + \|\|\omega\|^{2s} \mathcal{F}(f_i)(\omega)\|_{L^1(\mathbb{R}^{d+1})} \|\mathcal{F}(g_v)(\omega)\|_{L^\infty(\mathbb{R}^{d+1})} \right].$$

Hence,

$$\mathrm{T}_2 \le c_2 \|m(\omega)\|_{L^2(\mathbb{R}^{d+1})} \left[ \|\|\omega\|^{2s} \mathcal{F}(f_i)(\omega)\|_{L^1(\mathbb{R}^{d+1})} + \|\|\omega\|^{2s} \mathcal{F}[f_i](\omega)\|_{L^1(\mathbb{R}^{d+1})} \|g\|_{L^1(\mathbb{R}^{d+1})} \right]$$

The term $\|m(\omega)\|_{L^2(\mathbb{R}^{d+1})}$ and $\|\|\omega\|^{2s} m(\omega)\|_{L^2(\mathbb{R}^{d+1})}$ can be bounded using similar technique as before. We will use the result from Lemma 1.

$$\|\|\omega\|^{2s} m(\omega)\|^2_{L^2(\mathbb{R}^{d+1})} \le \int \|\omega\|^{4s} |\mathcal{F}[f_i](\omega)|^2 |1 - \mathcal{F}[g](v\omega)|^2 \, d\omega$$

$$\le \int_{v\|\omega\| \ge 1} \|\omega\|^{4s} |\mathcal{F}[f_i](\omega)|^2 | \, d\omega$$

$$= \int_{v\|\omega\| \ge 1} \|\omega\|^{4s} (1 + \|\omega\|^2)^{-\beta} (1 + \|\omega\|^2)^\beta |\mathcal{F}[f_i](\omega)|^2 | \, d\omega$$

$$\le 2^{2\beta} \sup_{v\|\omega\| \ge 1} \frac{\|\omega\|^{4s}}{(1 + \|\omega\|^2)^\beta} \|f_i\|^2_{W_2^\beta(\mathbb{R}^{d+1})}$$

$$\le 2^{2\beta} \sup_{v\|\omega\| \ge 1} \frac{1}{(1 + \|\omega\|^2)^{\beta - 2s}} \|f_i\|^2_{W_2^\beta(\mathbb{R}^{d+1})}$$

$$\le 2^{2\beta} \frac{v^{2(\beta - 2s)}}{(1 + v^2)^{\beta - 2s}} \|f_i\|^2_{W_2^\beta(\mathbb{R}^{d+1})}$$

$$= 2^{2s} \frac{(2v)^{2(\beta-2s)}}{(1+v^2)^{\beta-2s}} \|f_i\|^2_{W_2^\beta(\mathbb{R}^{d+1})} \le 2^{2s}(2v)^{2(\beta-2s)} \|f_i\|^2_{W_2^\beta(\mathbb{R}^{d+1})}.$$

Combining everything together, we have

$$\|(-\Delta)^s(p^\star - \hat{p})\|_{L^2(\mathbb{R}^{d+1})} \le c_1(2v)^{(\beta-2s)} \sum_{i=1}^{q} C_{T_1}^{(i)} \|f_i\|^2_{W_2^\beta(\mathbb{R}^{d+1})} + c_2(2v)^{\beta} \sum_{i=1}^{q} C_{T_2}^{(i)} \|f_i\|^2_{W_2^\beta(\mathbb{R}^{d+1})},$$

where $C_{T_1}^{(i)} = \|\mathcal{F}[f_i]\|_{L^1(\mathbb{R}^{d+1})} + \|\mathcal{F}[f_i]\|_{L^\infty(\mathbb{R}^{d+1})}\|g\|_{L^1(\mathbb{R}^{d+1})}$ and $C_{T_2}^{(i)} = \big\| \|\omega\|^{2s}\mathcal{F}(f_i)(\omega) \big\|_{L^1(\mathbb{R}^{d+1})} + \big\| \|\omega\|^{2s}\mathcal{F}[f_i](\omega) \big\|_{L^1(\mathbb{R}^{d+1})}\|g\|_{L^1(\mathbb{R}^{d+1})}$. $\square$

## C.2   Proof of Theorem 4

From the previous lemma, we have,

$$\|(-\Delta)^s(p^\star - \hat{p})\|_{L^2(\mathbb{R}^{d+1})} \le c_1(2v)^{(\beta-2s)} \sum_{i=1}^{q} C_{T_1}^{(i)} \|f_i\|^2_{W_2^\beta(\mathbb{R}^{d+1})} + c_2(2v)^{\beta} \sum_{i=1}^{q} C_{T_2}^{(i)} \|f_i\|^2_{W_2^\beta(\mathbb{R}^{d+1})},$$

for some positive $c_1$ and $c_2$ where $C_{T_1}^{(i)} = \|\mathcal{F}[f_i]\|_{L^1(\mathbb{R}^{d+1})} + \|\mathcal{F}[f_i]\|_{L^\infty(\mathbb{R}^{d+1})}\|g\|_{L^1(\mathbb{R}^{d+1})}$ and $C_{T_2}^{(i)} = \big\| \|\omega\|^{2s}\mathcal{F}(f_i)(\omega) \big\|_{L^1(\mathbb{R}^{d+1})} + \big\| \|\omega\|^{2s}\mathcal{F}[f_i](\omega) \big\|_{L^1(\mathbb{R}^{d+1})}\|g\|_{L^1(\mathbb{R}^{d+1})}$. Now, if we choose, for some constant $c$

$$v = \left( \frac{\varepsilon}{c2^{\beta-2s} \sum_{i=1}^{q}(C_{T_1}^{(i)} + C_{T_2}^{(i)})\|f_i\|^2_{W_2^\beta(\mathbb{R}^{d+1})}} \right)^{\frac{1}{\beta-2s}},$$

then

$$\|(-\Delta)^s(p^\star - \hat{p})\|_{L^2(\mathbb{R}^{d+1})} \le \varepsilon.$$

We can bound the other two terms as using lemma 2. We have,

$$\left\| \frac{\partial p^\star(x,t)}{\partial t} - \frac{\partial \hat{p}(x,t)}{\partial t} \right\|_{L^2(\mathbb{R}^{d+1})} \le 8\pi(2v)^{(\beta-1)} \sum_{i=1}^{q} \|f_i\|_{W_2^\beta(\mathbb{R}^{d+1})}\|f_i\|_{L^\infty(\mathbb{R}^{d+1})}$$

$$+ 2(2v)^{\beta} \sum_{i=1}^{q} \|f_i\|_{W_2^\beta(\mathbb{R}^{d+1})} \left\| \frac{\partial f_i}{\partial t} \right\|_{L^\infty(\mathbb{R}^{d+1})} \|g\|_{L_1(\mathbb{R}^{d+1})}.$$

If we choose

$$v = \min\left( \left( \frac{\varepsilon}{2^{\beta-1}16\pi C_1} \right)^{\frac{1}{\beta-1}}, \left( \frac{\varepsilon}{2^{\beta+1}C_2} \right)^{\frac{1}{\beta}} \right)$$

where $C_1 = \sum_{i=1}^{q} \|f_i\|_{W_2^\beta(\mathbb{R}^{d+1})}\|f_i\|_{L^\infty(\mathbb{R}^{d+1})}$ and $C_2 = \sum_{i=1}^{q} \|f_i\|_{W_2^\beta(\mathbb{R}^d)} \left\| \frac{\partial f_i}{\partial t} \right\|_{L^\infty(\mathbb{R}^{d+1})} \|g\|_{L_1(\mathbb{R}^{d+1})}$. Hence,

$$\left\| \frac{\partial p^\star(x,t)}{\partial t} - \frac{\partial \hat{p}(x,t)}{\partial t} \right\|_{L^2(\mathbb{R}^{d+1})} \le \varepsilon.$$

Similarly, from lemma 2

$$\left\| \frac{\partial p^\star(x,t)}{\partial x_k} - \frac{\partial \hat{p}(x,t)}{\partial x_k} \right\|_{L^2(\mathbb{R}^{d+1})} \le 8\pi(2v)^{(\beta-1)} \sum_{i=1}^{q} \|f_i\|_{W_2^\beta(\mathbb{R}^{d+1})}\|f_i\|_{L^\infty(\mathbb{R}^{d+1})}$$

$$+ 2(2v)^{\beta} \sum_{i=1}^{q} \|f_i\|_{W_2^\beta(\mathbb{R}^{d+1})} \left\| \frac{\partial f_i}{\partial x_k} \right\|_{L^\infty(\mathbb{R}^{d+1})} \|g\|_{L_1(\mathbb{R}^{d+1})}.$$

From [RC21, Theorem D.4 (Eq. D.27) ], we have,

$$\|p^\star(x,t) - \hat{p}(x,t)\|_{L^2(\mathbb{R}^{d+1})} \leq (2v)^\beta (1 + \|g\|_{L^1(\mathbb{R}^{d+1})}) \sum_{i=1}^q \|f_i\|_{W_2^\beta(\mathbb{R}^{d+1})} \|f_i\|_{L^\infty(\mathbb{R}^{d+1})}.$$

Hence, if we choose,

$$v = \min\left( \left( \frac{\varepsilon}{2^{\beta-1}16\pi C_1} \right)^{\frac{1}{\beta-1}}, \left( \frac{\varepsilon}{2^{\beta+1}C_2} \right)^{\frac{1}{\beta}}, \left( \frac{\varepsilon}{2^{\beta+1}C_3} \right)^{\frac{1}{\beta}} \right)$$

where $C_1 = \sum_{i=1}^q \|f_i\|_{W_2^\beta(\mathbb{R}^{d+1})} \|f_i\|_{L^\infty(\mathbb{R}^{d+1})}$, $C_2 = \sum_{i=1}^q \|f_i\|_{W_2^\beta(\mathbb{R}^{d+1})} \left\| \frac{\partial f_i}{\partial t} \right\|_{L^\infty(\mathbb{R}^{d+1})} \|g\|_{L^1(\mathbb{R}^{d+1})}$ and $C_3 = (1 + \|g\|_{L^1(\mathbb{R}^{d+1})}) \sum_{i=1}^q \|f_i\|_{W_2^\beta(\mathbb{R}^{d+1})} \|f_i\|_{L^\infty(\mathbb{R}^{d+1})}$, then, we have

$$\left\| \sum_{i=1}^d \frac{\partial}{\partial x_i} (\mu_i(x,t)p^\star(x,t) - \mu_i(x,t)\hat{p}(x,t)) \right\|_{L^2(\mathbb{R}^{d+1})} \leq d(R_\mu + R_{\mu_p})\varepsilon.$$

Hence, by combining everything together, we have

$$\left\| \frac{\partial \hat{p}(x,t)}{\partial t} + \sum_{i=1}^d \frac{\partial}{\partial x_i}(\mu_i(x,t)\hat{p}(x,t)) - (-\Delta)^s(\hat{p}(x,t)) \right\|_{L^2(\mathbb{R}^{d+1})} \leq \varepsilon(2 + d(R_\mu + R_{\mu_p})).$$

We can bound the trace of the matrix as follows. We have,

$$M_\varepsilon = \sum_{i=1}^q f_{i,v} f_{i,v}^\top.$$

From [RC21, Equation D.21], we have

$$Tr(M_\varepsilon) = \sum_{i=1}^q \|f_{i,v}\|_{\mathcal{H}} = c_\eta 2^{2\beta}(1 + (\frac{v}{3})^{2\beta} e^{\frac{89}{\eta_0 v^2}}) \sum_{i=1}^q \|f_i\|_{W_2^\beta(\mathbb{R}^{d+1})}^2$$

Since, we choose,

$$v = \left( \frac{\varepsilon}{96\pi^2 2^{\beta-2}C} \right)^{\frac{1}{\beta-2s}},$$

where $C$ depends on $f_i$s, $\mathcal{F}[f_i]$ and it's derivative.

$$Tr(M_\varepsilon) = c_\eta 2^{2\beta}(1 + (\frac{v}{3})^{2\beta} e^{\frac{89}{\eta_0 v^2}}) \sum_{i=1}^q \|f_i\|_{W_2^\beta(\mathbb{R}^{d+1})}^2$$

$$= c_\eta 2^{2\beta} \left( 1 + 3^{-2\beta} \left( \frac{\varepsilon}{96\pi^2 2^{\beta-2}C} \right)^{\frac{2\beta}{\beta-2s}} e^{\frac{89}{\eta_0 v^2}} \right) \sum_{i=1}^q \|f_i\|_{W_2^\beta(\mathbb{R}^{d+1})}^2$$

$$= c_\eta 2^{2\beta} \sum_{i=1}^q \|f_i\|_{W_2^\beta(\mathbb{R}^{d+1})}^2 \left( 1 + 3^{-2\beta} \left( \frac{\varepsilon}{\tilde{C}} \right)^{\frac{2\beta}{\beta-2s}} e^{\frac{89}{\eta_0 \left( \frac{\varepsilon}{\tilde{C}} \right)^{\frac{2}{\beta-2s}}}} \right)$$

$$= \hat{C}|\eta|^{1/2} \left( 1 + \varepsilon^{\frac{2\beta}{\beta-2s}} e^{\frac{\tilde{C}'}{\eta_0 \varepsilon^{\frac{2}{\beta-2s}}}} \right) \leq \hat{C}|\eta|^{1/2} \left( 1 + \varepsilon^{\frac{2\beta}{\beta-2s}} \exp\left( \frac{\tilde{C}'}{\eta_0} \varepsilon^{-\frac{2}{\beta-2s}} \right) \right)$$

where $\tilde{C} = 96\pi^2 2^{\beta-2}C$, $\tilde{C}' = 89\tilde{C}^{\frac{2}{\beta-2}}$, and $\hat{C} = \pi^{-d/2} 2^{2\beta} \sum_{i=1}^q \|f_i\|_{W_2^\beta(\mathbb{R}^{d+1})}^2 \max\left( 1, \frac{3^{-2\beta}}{\tilde{C}^{\frac{2\beta}{\beta-2s}}} \right)$.

## C.3  Approximation Properties of PSD Model for Fractional Laplacian (Proof of Theorem 4)

*Proof.* From the previous result in theorem 4, we know that there exists an $\mathsf{M}_\varepsilon \in \mathbb{S}_+(\mathcal{H}_X \otimes \mathcal{H}_T)$ with $\operatorname{rank}(\mathsf{M}_\varepsilon) \le q$, such that for the representation $\hat{p}(x,t) = \phi_{X,T}^\top \mathsf{M}_\varepsilon \phi_{X,T}$, following holds under assumption 2 on the coefficients of the fractional FPE,

$$\left\| \frac{\partial \hat{p}(x,t)}{\partial t} + \sum_{i=1}^{d} \frac{\partial}{\partial x_i}(\mu_i(x,t)\hat{p}(x,t)) + (-\Delta)^s \hat{p}(x,t) \right\|_{L^2(\tilde{x})} = O(\varepsilon), \tag{56}$$

for an appropriate choice of parameters. We also know that

$$\frac{\partial p^\star(x,t)}{\partial t} + \sum_{i=1}^{d} \frac{\partial}{\partial x_i}(\mu_i(x,t)p^\star(x,t)) + (-\Delta)^s p^\star(x,t) = 0. \tag{57}$$

We know that, for given $A_m$, $\tilde{p}(x,t) = \psi(x,t)^\top \tilde{P}\mathsf{M}_\varepsilon \tilde{P}\psi(x,t)^\top$ where $\tilde{P}$ is the projection operator defined earlier. We will utilize the Gagliardo–Nirenberg inequality to control the error in compression of fractional laplacian compressed model with respect to the original one. For any fixed $t$,

$$(-\Delta)^s[\hat{p}(x,t) - \tilde{p}(x,t)] = (-\Delta)^s[\psi(x,t)^\top(\tilde{P}\mathsf{M}_\varepsilon\tilde{P} - \mathsf{M}_\varepsilon)\psi(x,t)]$$

By applying Gagliardo–Nirenberg inequality for fractional laplacian [MP18], we have

$$\|(-\Delta)^s\beta_t(x)\|_{L^2(\mathbb{R}^d)} \le \underbrace{\|\beta_t(x)\|_{L^2(\mathbb{R}^{d+1})}^{1-\vartheta}}_{:=T_1} \cdot \underbrace{\|\Delta\beta_t(x)\|_{L^2(\mathbb{R}^{d+1})}^{\vartheta}}_{:=T_2},$$

where $\vartheta = s \le 1$ and $\Delta$ is normal laplacian operator. Let us consider the term $T_1$ and $T_2$ separately. In our case, $\beta_t(x) = \psi(x,t)^\top(\tilde{P}\mathsf{M}_\varepsilon\tilde{P} - \mathsf{M}_\varepsilon)\psi(x,t)$. In the proof of theorem 8, we have proved the bound on $\Delta\beta_t(x)$ and $\beta_t(x)$. From equation (43), we have,

$$|\psi(x,t)^\top(\tilde{P}\mathsf{M}_\varepsilon\tilde{P} - \mathsf{M}_\varepsilon)\psi(x,t)| \le 2\|(I-\tilde{P})\psi(x,t)\|_{\mathcal{H}_\eta}\|\mathsf{M}_\varepsilon\|^{1/2}\|\mathsf{M}_\varepsilon^{1/2}\psi(x,t)\|_{\mathcal{H}_\eta}$$
$$+ \|(I-\tilde{P})\psi(x,t)\|_{\mathcal{H}_\eta}^2\|\mathsf{M}_\varepsilon\|$$
$$= 2c_M^{1/2}f(x;\mathsf{M}_\varepsilon,\psi)^{1/2}\tilde{u}_1(x) + c_M\tilde{u}_1(x)^2,$$

where $c_M = \|\mathsf{M}_\varepsilon\|$ and we denoted by $\tilde{u}_1(x)$ the quantity $\tilde{u}_1(x) = \|(I-\tilde{P})\psi(x,t)\|_{\mathcal{H}_\eta}$ and we noted that $\|\mathsf{M}_\varepsilon^{1/2}\psi(x,t)\|_{\mathcal{H}_\eta}^2 = \psi(x,t)^\top\mathsf{M}_\varepsilon\psi(x,t) = f(x;\mathsf{M}_\varepsilon,\psi(x,t))$. Similarly, we have shown that

$$\frac{\partial^2 p(x,t)}{\partial x_i x_j} - \frac{\partial^2 \tilde{p}(x,t)}{\partial x_i x_j} = 2\psi_{ji}''(x,t)^\top(\mathsf{M}_\varepsilon - \tilde{P}\mathsf{M}_\varepsilon\tilde{P})\psi(x,t) + 2\psi_i'(x,t)^\top(\mathsf{M}_\varepsilon - \tilde{P}\mathsf{M}_\varepsilon\tilde{P})\psi_j'(x,t). \tag{58}$$

From equations (46) and (47)

$$\left|\psi_{ji}''(x,t)^\top(\mathsf{M}_\varepsilon - \tilde{P}\mathsf{M}_\varepsilon\tilde{P})\psi(x,t)\right| \le c_M\tilde{u}_1(x)\tilde{u}_{x_jx_i}(x) + c_M^{1/2}f(x;\mathsf{M}_\varepsilon,\psi)^{1/2}\tilde{u}_{x_jx_i}(x)$$
$$+ c_M^{1/2}\tilde{f}_{x_jx_i}(x;\mathsf{M}_\varepsilon,\psi)^{1/2}\tilde{u}_1(x), \tag{59}$$

where $\tilde{f}_{x_jx_i}(x;\mathsf{M}_\varepsilon,\psi) = \frac{\partial^2\psi(x,t)}{\partial x_jx_i}^\top \mathsf{M}_\varepsilon \frac{\partial^2\psi(x,t)}{\partial x_jx_i}$ and $\tilde{u}_{x_jx_i}(x) = \left\|(I-\tilde{P})\frac{\partial^2\psi(x,t)}{\partial x_jx_i}\right\|_{\mathcal{H}_\eta}$.
And,

$$\psi_i'(x,t)^\top(\mathsf{M}_\varepsilon - \tilde{P}\mathsf{M}_\varepsilon\tilde{P})\psi_j'(x,t) \le c_M\tilde{u}_{x_j}(x)\tilde{u}_{x_i}(x) + c_M^{1/2}f_{x_i}(x;\mathsf{M}_\varepsilon,\psi)^{1/2}\tilde{u}_{x_j}(x)$$
$$+ c_M^{1/2}\tilde{f}_{x_j}(x;\mathsf{M}_\varepsilon,\psi)^{1/2}\tilde{u}_{x_i}(x). \tag{60}$$

Follwoing similar steps of calculations as that is Theorem 8 and Theroem 2, we get

$$\left\| \frac{\partial(\hat{p}(x,t) - \tilde{p}(x,t))}{\partial t} + \sum_{i=i}^{d} \frac{\partial(\mu_i(x,t)(\hat{p}(x,t) - \tilde{p}(x,t)))}{\partial x_i} + (-\Delta)^s(p^\star(x,t)(\hat{p}(x,t) - \tilde{p}(x,t))) \right\|_{L^2(\tilde{x})}$$

$$= P_1 \|\mathsf{M}_\varepsilon\| C^2 q_\eta^2 e^{\frac{-2c\sigma}{h} \log \frac{c\sigma}{h}} R^{(d+1)/2} + P_2 C q_\eta e^{-\frac{c\sigma}{h} \log \frac{c\sigma}{h}} \|\mathsf{M}_\varepsilon\| R^{(d+1)/2}$$
$$+ P_3 \|\mathsf{M}_\varepsilon\|^{1/2} C q_\eta e^{-\frac{c\sigma}{h} \log \frac{c\sigma}{h}} \|\sqrt{p(x,t)}\|_{L^2(\tilde{x})}, \tag{61}$$

for some constant $P_1, P_2$ and $P_3$. Now, let us compute $\|\mathsf{M}_\varepsilon\|$. We have

$$\|\mathsf{M}_\varepsilon\| \leq Tr(\mathsf{M}_\varepsilon) = \hat{C} \tau^{(d+1)/2} \left( 1 + \varepsilon^{\frac{2\beta}{\beta-2s}} e^{\frac{\tilde{C}'}{\tau} \varepsilon^{-\frac{2}{\beta-2s}}} \right).$$

Let us choose, $\tau = \frac{\tilde{C}' \varepsilon^{-\frac{2}{\beta-2s}}}{\frac{2\beta}{\beta-2s} \log(\frac{1+R}{\varepsilon})}$. Hence,

$$\|\mathsf{M}_\varepsilon\| \leq C_3 (1+R)^{\frac{2\beta}{\beta-2s}} \varepsilon^{-\frac{d+1}{\beta-2s}},$$

For some constant $C_3$. Hence, Then, note that $\|p^{1/2}\|_{L^2(\mathcal{X})} = \|p\|_{L^1(\mathcal{X})}^{1/2}$. Hence, using the argument in [RC21], we have

$$\|p^{1/2}\|_{L^2(\tilde{x})} \leq 2R.$$

Term $R$ comes from the fact that for each time instance $p(x,t)$ is a density and $t \in (0 < R)$. Now,

$$\left\| \frac{\partial(\hat{p}(x,t) - \tilde{p}(x,t))}{\partial t} + \sum_{i=i}^d \frac{\partial(\mu_i(x,t)(\hat{p}(x,t) - \tilde{p}(x,t)))}{\partial x_i} + (-\Delta)^s(p^\star(x,t)(\hat{p}(x,t) - \tilde{p}(x,t))) \right\|_{L^2(\tilde{x})}$$
$$= P_1 C_3 (1+R)^{\frac{2\beta}{\beta-2s}} \varepsilon^{-\frac{d+1}{\beta-2s}} C^2 q_\eta^2 e^{\frac{-2c\sigma}{h} \log \frac{c\sigma}{h}} R^{(d+1)/2} + P_2 C q_\eta e^{-\frac{c\sigma}{h} \log \frac{c\sigma}{h}} C_3 (1+R)^{\frac{2\beta}{\beta-2s}} \varepsilon^{-\frac{d+1}{\beta-2s}} R^{(d+1)/2}$$
$$+ P_3 C_3^{1/2} R (1+R)^{\frac{\beta}{\beta-2s}} \varepsilon^{-\frac{d+1}{2(\beta-2s)}} C q_\eta e^{-\frac{c\sigma}{h} \log \frac{c\sigma}{h}} \|\sqrt{p(x,t)}\|_{L^2(\tilde{x})}.$$

By choosing $h = c\sigma/s$ with $s = \max(C', (1 + \frac{d+1}{2(\beta-2s)}) \log \frac{1}{\varepsilon} + (1 + \frac{d}{2}) \log(1+R) + \log(\hat{C}) + e)$, for some big enough $\hat{C}$. Since $s \geq e$, then $\log s \geq 1$, so

$$C e^{-\frac{c\sigma}{h} \log \frac{c\sigma}{h}} = C e^{-s \log s} \leq C e^{-s} \leq C'(1+R)^{-d/2} \varepsilon^{1 + \frac{d+1}{2(\beta-2s)}}.$$

Hence, combining evrything, we have from triangles inequality,

$$\left\| \frac{\partial(p^\star(x,t) - \tilde{p}(x,t))}{\partial t} + \sum_{i=i}^d \frac{\partial(\mu_i(x,t)(p^\star(x,t) - \tilde{p}(x,t)))}{\partial x_i} + (-\Delta)^s(p^\star(x,t) - \tilde{p}(x,t)) \right\|_{L^2(\tilde{x})}$$
$$= O(\varepsilon)$$

To conclude we recall the fact that $\tilde{x}_1, \dots, \tilde{x}_m$ is a $h$-covering of $T$, guarantees that the number of centers $m$ in the covering satisfies

$$m \leq (1 + \frac{2R\sqrt{d+1}}{h})^{d+1}.$$

Then, since $h \geq c\sigma/(C_4 \log \frac{C_5 \log(1+R)}{\varepsilon})$ with $C_4 = 1 + d/\min(2(\beta-2s), 2)$ and $C_5 = (\hat{C})^{1/C_4}$, and since $\sigma = \min(R, 1/\sqrt{\tau})$, then $R/\sigma = \max(1, R\sqrt{\tau}) \leq 1 + \sqrt{\tilde{C}'} \varepsilon^{-1/(\beta-2)} (\log \frac{1+R}{\varepsilon})^{-1/2}$, so after final calculation, we have

$$m^{\frac{1}{d+1}} \leq 1 + 2R\sqrt{d+1}/h \leq C_8 + C_9 \log \frac{1+R}{\varepsilon} + C_{10} \varepsilon^{-\frac{1}{\beta-2s}} \left( \log \frac{1+R}{\varepsilon} \right)^{1/2},$$

for some constants $C_8, C_9$ and $C_{10}$ independent of $\varepsilon$. $\qquad \square$

