# OpenReview forum: "Efficient Sampling of Stochastic Differential Equations with Positive Semi-Definite Models"
_NeurIPS.cc/2023/Conference — NeurIPS 2023 poster_

### Official Review · Reviewer_jfmV · 2023-06-24

**Soundness:** 3 good
**Presentation:** 3 good
**Contribution:** 3 good
**Rating:** 6
**Confidence:** 2

**Summary:**

This article investigates the approximation of the Fokker-Planck equation and the fractional Fokker-Planck equation using positive semi-definite (PSD) models. The authors demonstrate that for a given accuracy $\epsilon$, there exists a PSD model with a dimension not exceeding $\epsilon^{-(d+1)/(\beta - 2s)}(\log 1/\epsilon)^{d+1}$ that approximates the corresponding partial differential equation (PDE). By utilizing this result in conjunction with the PSD model-based sampling method proposed in [1], the authors are able to provide samples of the stochastic differential equation (SDE) corresponding to the Fokker-Planck equation or fractional Fokker-Planck equation at any given time $t$.

[1]Marteau-Ferey U, Bach F, Rudi A. Sampling from arbitrary functions via psd models[C]//International Conference on Artificial Intelligence and Statistics. PMLR, 2022: 2823-2861.

**Strengths:**

1. This article demonstrates clear and logical writing, making it easy to understand the problem formulation considered by the authors as well as the fundamental concepts and properties of the PSD model adopted.
2. As a newly proposed model, this paper presents a novel application scenario for the PSD model, which has not been previously explored in related works. This contributes to its significant novelty.
3. From the perspective of sampling based on SDEs, the authors provide a new design approach for sampling by combining existing works, offering a fresh perspective on sampling techniques.

**Weaknesses:**

1. Although the author mentions it in the paper, the influence of initial conditions on the solution of the PDE can sometimes be significant. It would enhance the completeness of the theoretical analysis in this article if the author could provide an analysis of whether the PSD model can satisfy the initial conditions or, in cases where exact satisfaction is not possible, characterize the approximation of the initial conditions' impact on the final solution's approximation.
2. Personally, I have limited knowledge of non-log-concave sampling and numerical modeling of PDEs. However, it would be beneficial if the author could compare the effectiveness of the PSD model-based sampling method with other traditional methods (such as Langevin Monte Carlo) in terms of sampling performance. Additionally, comparing the approximation methods of the PSD model for PDE solutions with other existing approaches, either from the perspective of complexity analysis or purely experimental results, would provide valuable insights.

**Questions:**

1. In line 180, should it be $E e^{j\langle u, L_{t-s}^\alpha\rangle} = \exp\{-(t-s)||u||_2^\alpha\}$?
2. In line 216, should it be $Y \in \mathbb{R}^{n\times d^\prime}$?

**Limitations:**

Despite the theoretical guarantees provided by the author, I believe the biggest limitation of this article lies in the lack of clear demonstration of the advantages of the PSD model. It would be beneficial if the author could explain how the PSD model stands out compared to other similar models in terms of approximating FPE and subsequent sampling tasks. This would significantly enhance the persuasiveness of the article.

---

> ### Author Rebuttal · Authors · 2023-08-08
>
> We thank the reviewer for positively assessing our paper. Here below we address the concerns raised by the reviewer.
>
> **Approximation Error with Initial Condition**:
> We bound the $L_2$ error of the difference between the true value and approximated value of the (fractional) Fokker-Planck equation. However, to achieve this, we need to get the approximation error of all three terms individually in the equation and that also includes approximating $p(x,t)$. Please refer to Eqns 39, 40, 41, and 42, and the block of equations after 47 in the appendix where we bound each term individually. This continues in the rest of the proofs. We get the same guarantees for $p(x,t)$ that we get for our PDE which is similar to (Rudi and Ciliberto, 2021, Theorem 6). Hence, the initial distribution will not change any approximation result in the paper. It will only act as a constraint in the optimization problem to obtain the parameter of the PSD model. Also, please see our response to reviewer T6Bn regarding the algorithm.
>
> **Weakness, Comparison, and Limitations**:   Previous papers about Langevin Monte Carlo and projected Langevin monte carlo algorithms relied on assumptions such as dissipativity, log-Sobolev, or Poincare inequality, which are not always satisfied and we do not require any of such assumptions. Furthermore, we are not
> aware of an analysis that covers such a broad class of solutions that we cover in this paper. We only assume that the density function is $\beta$ time differentiable. This assumption is much weaker.   The existing result for fractional Fokker-Planck is even more scarce. We are not aware of an approach that provides approximation guarantees of the solution of the fractional Fokker-Planck equation that we can sample from. Since the assumptions about the density function are quite different in sampling literature and in our paper, it is hard to do one on one comparison of the results. We have provided worst-case guarantees under the much weaker assumption ($\beta$ times differentiable) and it is not fair to compare them to methods that make stronger assumptions (dissipativity, log-Sobolev, Poincare) to prove convergence results.
>
> Our sample complexity results for approximating the solution of the (fractional) Fokker-Planck Equation in theorems 2 and 5 are better than approximation guarantees in [LCL+21]. On top of that, our method comes with one more advantage over existing machine learning models to approximate the solution of a (fractional) Fokker-Planck equation. Generally, the approximate solution of a PDE obtained by ML models [LCL+21, RPK19, Y+18] is not bound to stay positive and hence can not act as a candidate to approximate the solution of a (fractional) Fokker-Planck equation whose solution is a probability density. In the previous work [RC21], it was shown that the PSD model can be used for the effective representation of probability density. Hence, PSD model-based representation acts as a good candidate to approximate the solution of a (fractional) Fokker-Planck equation. We theoretically show the validity of the PSD model in approximating the solution of the (fractional) Fokker-Planck equation and provide a sample complexity guarantee that was not known before. This guarantee immediately can be used to provide a sample quality guarantee in Corollary 1. Sampling result in such generality was not known before.
>
> We will take care of the minor comments in the revision. In line 180, $(t-s)^\alpha$ is correct. We will fix the dimension of Y in line 216.
>
> Also, please look at our response to the reviewer oceV regarding contributions and experiments. Please look at our response to reviewer biEz for general remarks.
>
>  We hope that we have answered all your concerns and that the scores will be raised further.

---

> > ### Author Response · Authors · 2023-08-11
> > **Start of the author reviewer discussion**
> >
> > We hope that our response answers the questions the reviewer asked. Please let us know the further questions or confusion about this work so that we can address it. If we have answered all the questions from the reviewer, then in light of all the novel contributions we have made in this paper, we would like to request the reviewer to reevaluate the score assigned to this paper.

---

> > > ### Author Response · Authors · 2023-08-14
> > > **Reminder: Start of the author reviewer discussion**
> > >
> > > Please let us know if you have further questions or confusion about this work that we can address. If we have answered all the questions from the reviewer, then in light of all the novel contributions we have made in this paper, we would like to request the reviewer to reevaluate the score assigned to this paper.

---

### Official Review · Reviewer_T6Bn · 2023-07-04

**Soundness:** 3 good
**Presentation:** 3 good
**Contribution:** 3 good
**Rating:** 7
**Confidence:** 3

**Summary:**

The authors introduced a new method of sampling solutions of the Fokker--Planck equation (FPE) using a positive semidefinite (PSD) model. While the algorithm used for the PSD model is not new, the authors created a framework to sample from the FPE, which is quite original based on my knowledge.

While I believe the results are very neat, I would like to clarify several questions with the authors before providing a confident score. For now I will recommend borderline accept, and will raise the score once my questions are adequately addressed.

**Strengths:**

The approach to drawing samples for the Fokker--Planck is very unique and original. I believe alternative methods that hold promise should be welcomed and more carefully studied, regardless of how practical it is at the moment.

**Weaknesses:**

There are a few clarifications I would like to have regarding the implementation of the algorithm.

**Questions:**

1. To start, while I understand the algorithm of [MFBR22] is not part of the contribution for this work, can the authors provide a high level explanation of how the algorithm works, and comment on its computational complexity? I believe all readers can benefit from understanding the core algorithm, and plus I believe readers should be able to implement the algorithm presented in this paper without having to reference other works.

2. Can authors clarify the role that $q$ plays in this problem? In particular, is $q$ given or chosen by the user, and how does the value of $q$ affect the computational complexity of this algorithm?

3. Under the discussion of "Efficient sampling method" the authors mentioned a step of finding the best model $\hat{A}_m$ by minimizing an error function. Suppose I am an applied scientist who wants to sample from a given FPE, how would I compute this model $\hat{A}_m$ from the PDE? On a high level, I would like to understand how to implement this sampler end to end, and I don't think I fully understand this part.

Once again, I would be happy to raise my score once my questions are adequately answered.

---

> ### Author Rebuttal · Authors · 2023-08-08
>
> We thank the reviewer for assessing our work positively. Here below we address the major concerns raised by the reviewer.
>
> **Sampling Algorithm**: We will add the details of the sampling algorithm in the revision. Here below, we are describing the methodology of [MFBR22] for sampling algorithm for PSD models on a finite hyper-rectangle. The algorithm is an iterative sampling procedure. Given a bounded hyper-rectangle $Q$, $p_Q$ denotes the PSD models-based density in the hyper-rectangle Q. Details are in [MFBR22]. Given the function $f$, the algorithm will take three inputs $(Q,N, \varepsilon)$: the hyper-rectangle $Q$ (with
> sides parallel to the axes) from which we would like to sample, the number of i.i.d. samples $N$
> which we would like to obtain, and a parameter $\varepsilon$  which defines the quality of the approximation. at first,  $Q$ is cut in half in its longest direction forming two sub-rectangles $Q_1$ and $Q_2$. If $X_Q$ were a random variable following the law of $p_Q$, then $X_Q \in Q_i$ with $p_i$ (can be computed in closed form) and $X_Q | \{X_Q \in Q_i\}$ follows the law of $p_{Q_i}$ Therefore when looking for a sample from $p_Q$,   one of the two smaller sub-rectangles $Q_i$ are chosen randomly with probability $p_i$ in which we look for the sample and then call the algorithm recursively to get a sample from $p_{Q_i}$. By applying appropriate stopping criteria that depend on $\varepsilon$, we get a random sample in $Q$. More details about the algorithm are provided in [MFBR22]. The computational and statistical guarantees are provided in Theorems 1 and 2 of [MFBR22].
>
> **Effect of q**: From Lemma 2 in the appendix, it is clear that the approximation error grows linearly with $q$. Hence, the sample complexity will increase linearly with $q$.  Intuitively, as $q$ grows, the complexity of the target density function increases and hence, it would require more samples to approximate the target density using the PSD model. However, we are not allowed to choose $q$ as it is a property of the target density. For example, if the probability is bounded away from 0 then q=1. (For further details, please look at: ''*Second order conditions to decompose smooth functions as sums of squares*", Ulysse Marteau-Ferey, Francis Bach, and Alessandro Rudi)
>
> **Algorithm**: The algorithm to sample consists of two parts. In the first part, we get the approximate solution of the corresponding (fractional)-FPE using the PSD model and in the later part, sampling can be done using the method described in [MFBR22]. There are various ways to obtain the PSD model-based approximation for the solution of the (fractional)-FPE. Here below we describe a couple of ways. \\
>
> (1).  If the drift $\mu$ satisfies the regularity condition as in [BRS16, Theorem 1.1], then the drift function $\mu$ can be approximated by a function $\hat{\mu}$ in Gaussian RKHS and the approximation guarantees are due
>  to interpolation result in Lemma 4. Once, we have $\hat{\mu}$, we can solve the following constraint SDP to obtain the ${A}$.
>
> $\min_{\tilde{p}(x,t)} {\left\\| \sum_{i =1}^d \sum_{j =1}^d D_{ij}\frac{\partial^2}{\partial x_i \partial x_j} \tilde{p} (x,t) -  \frac{\partial  \tilde{p}(x,t)}{\partial t}  - \sum_{i =1}^d \frac{\partial }{\partial x_i}(\hat{\mu}_i(x,t)  \tilde{p}(x,t) )   \right\\|}$  such that $\tilde{p}(x,0) = p_0(x)$.  $p_0(x)$ is the initial density and the norm is $L^{2} (\tilde{\mathcal{X}})$ norm. The above optimization problem is clearly a constraint semi-definite program because of the representation of $\tilde{p}(x,t)$  as a PSD model. Given $m$ base points pair $(\tilde{x}_i,\tilde{t}_i) \in \tilde{\mathcal{X}}$ for $i \in \{1,2, \cdots , m\}$ of space and time (that is selected uniformly over the ball of radius $R$), we have, $\tilde{p}(x,t)$ given in equation 6 of the paper. Hence, the above optimization problem optimizes for a PSD matrix $A$. Now, the only remaining question to answer is if the norm $L^{2} (\tilde{\mathcal{X}})$ in the above optimization problem can be computed in the closed form or not.
>
> Since, the representation of $\tilde{p}(x,t)$ is in the span of the product of two functions in Gaussian RKHS and $\hat{\mu}$ lies in Gaussian RKHS by approximation, hence $\\|\cdot\\|_{L^{2} (\tilde{\mathcal{X}})}$ can be computed in closed form. Result in Corollary 1 of the paper is presented under this scheme.
>
> (2). As a second procedure, we can follow the approach described in section 3.2 in [RC21] and optimize an empirical version of the problem that is
>
> $\min_{A \geq 0} \frac{1}{n} \sum_{k=1}^n {\left\\| \sum_{i =1}^d \sum_{j =1}^d D_{ij}\frac{\partial^2}{\partial x_i \partial x_j} \tilde{p} (x_k,t_k) -  \frac{\partial  \tilde{p}(x_k,t_k)}{\partial t}  - \sum_{i =1}^d \frac{\partial }{\partial x_i}(\hat{\mu}_i(x_k,t_k)  \tilde{p}(x_k,t_k) )   \right\\|}^2$  such that $\tilde{p}(x,0) = p_0(x)$ and $\tilde{p}(x,t)$ is given in equation 6 of the paper.
>
> In the above optimization problem, $(x_k,t_k)$ for $k \in \{1,\cdots , n\}$ are iid samples from the hyper-rectangle of side $R$ and  $(\tilde{x}_i,\tilde{t}_i) \in \tilde{\mathcal{X}}$ for $i \in \{1,2, \cdots , m \}$ are $m$ base points pair of space and time (that is selected uniformly over the ball of radius $R$). As long as $(x_k,t_k)$ for $k \in \{1,\cdots , n\}$ remain iid, the sampling procedure of $(x_k,t_k)$ would not change our theoretical result. Hence, we can also choose uniformly at random. A small amount of regularization can be added to control the norm of $A$. However, this procedure would require us to study the error arising due to empirical approximation in the problem (Theorem 7, RC21). Since the focus of this work is to study the approximation properties of the PSD model in approximating the solution of the (fractional) FPE and sampling from it, we do not discuss the empirical approximation method in the paper.
>
> We hope that we have answered all your concerns and that the scores will be raised further.

---

> > ### Author Response · Authors · 2023-08-11
> > **Start of the author reviewer discussion**
> >
> > We hope that our response answers the questions the reviewer asked. Please let us know the further questions or confusion about this work so that we can address it. If we have answered all the questions from the reviewer, then in light of all the novel contributions we have made in this paper, we would like to request the reviewer to reevaluate the score assigned to this paper.

---

> > > ### Author Response · Authors · 2023-08-14
> > > **Reminder: Start of the author reviewer discussion**
> > >
> > > Please let us know if you have further questions or confusion about this work that we can address. If we have answered all the questions from the reviewer, then in light of all the novel contributions we have made in this paper, we would like to request the reviewer to reevaluate the score assigned to this paper.

---

> > > > ### Comment · Reviewer_T6Bn · 2023-08-16
> > > > **Response**
> > > >
> > > > I apologize for my delayed response, I have been traveling and wrestling with a heavy review load this year. I'm sorry if this created additional stress for the authors.
> > > >
> > > > Indeed, I believe my concerns have been addressed, and I will raise my score to accept. I find this work quite interesting, and I hope the authors continue to develop this topic further.

---

> > > > > ### Author Response · Authors · 2023-08-16
> > > > >
> > > > > Thank you very much for acknowledging our response.

---

### Official Review · Reviewer_oceV · 2023-07-06

**Soundness:** 4 excellent
**Presentation:** 2 fair
**Contribution:** 4 excellent
**Rating:** 7
**Confidence:** 4

**Summary:**

This paper considers numerical methods for the Fokker-Plank equation and its "fractional Laplacian" version. These partial differential equations describe the evolution of densities of certain random processes: in the case of standard FP, the process is a regular diffusion (an SDE driven by Brownian motion), whereas the fractional version corresponds to a process driven by $\alpha$-stable noise. Accordingly, the authors focus on initial conditions given by probability densities.

The idea behind the paper is to adapt the fairly recent methodology of positive semidefinite models (PSD) to this PDE problem. PSD models estimate nonnegative functions via feature maps $\phi:X\to H$, where $X$ is the domain of the function and $H$ is a Hilbert space. Given $\phi$, the nonnegative function of interest is estimated as $\langle \phi, M\phi\rangle$, where the PSD linear operator $M$ is chosen from the data. The whole method is based on the same idea as the well-known kernel trick: there is no need to explicitly deal with the Hilbert space $H$; only the kernel matrix over data points will matter.

In a nutshell, the paper gives a method for approximately solving the two aforementioned PDEs via PSD models. The salient features of this approach are as follows.

1) The PSD estimator can be obtained via a semidefinite program (as shown in previous work).

2) The method will output probability densities, and one can sample from them via methods from previous work.

3) The complexity of the problem grows like $\approx \epsilon^{-d/\beta}$, where $\epsilon$ is the target accuracy, $d$ is the dimension and $\beta$ measures the degree of smoothness of the solution. The exponent is not quite right, but the point is that sufficient smoothness will avoid the curse of dimensionality.

One may note that this method has rigorous error guarantees. This sets it apart from other PDE solvers based on machine learning ideas, such as Physics Inspired Neural Nets (PINNs).


**Strengths:**

The paper gives strong convergence guarantees for a new PDE solving method. Although several ingredients come from previous work, they are combined with new ideas into an impressive package. As mentioned above, this is a ML-based PDE solver with rigorous guarantees. Importantly, these guarantees do not require coercivity or hypercontractivity properties that are usually required from Langevin samples for solutions.

**Weaknesses:**

* No experiments are performed to compare the method with other PDE solvers (though this is understandable in a heavily mathematical paper).
* The writing is a bit sloppy in places: I found quite a few typos and have several minor comments on writing (I could probably find some more typos if I kept looking). See the list below.

NB: were it not for the sloppiness, I would have given this paper a higher score.

---

_List of typos and comments on writing_

(Line numbers refer to the full paper in the SM.)

Lines 146/7: "It was" and "It has" should be "They were" and "They have".

Line 180: why define $j=\sqrt{-1}$ if you could have just written $\sqrt{-1}$ in the exponent? (You never use this notation again, do you?)

Line 185: Shouldn't the condition for infinite $p$-th moment be $p\geq \alpha$?

Line 201 and elsewhere: I find it a bit odd that the transpose notation is used for vectors in Hilbert space. If you are going to use it, at least explain here what is going on.


Line 251: why is this sensence needed here?

Line 205: $\eta$ was treated as a vector before, and will be treated as a vector later in the text. Here, however, it seems to be a scalar.

Line 223: "The cost of the algorithm is..." -- per sample point, right?

Line 254: "Hence, $p^\star(x,t)$"

Line 255 should end with ":".

Line 270: the definition of $\psi$ should be in a numbered display for easier refererence.

Display (8): if $p^*$ is the true solution, why do we have it in here? (Its contribtion vanishes.) Indeed, the definition of the loss later does not involve $p^*$; see eg. (9). (On the other hand, $p^\star$ shows up again in the display right after line 989.)

Line 294 and elsewhere: when you are mentioned a numbered theorem or assumption, start the corresponding word with a capital letter.

Line 295: $\mathcal{H}_X\otimes\mathcal{H}_T$

Line 318: "by Gaussian kernel approximation" -- should this be here?

Line 365: "Similarly to the previous section, two steps are required to obtain"

Line 327 and elsewhere: I think $\mathbb{E}_t$ was not defined anywhere.

Line 352: "If the kernel satisfies the Bochner..."

Line 355 and elsewhere: this is a matter of taste, but couldn't you replace "utilize" with "use".

Line 376: "The major difficulty"

Line 391 (taste): "does require" could be "require".

Line 395: I think "approximated" should be "approximating".

Proposition 4 is also in Evans's book, which was cited previously (I am saying this from memory, so please check. $W_2$ is replaced by $H$ in that book.)

Many spots of the SM: the inequality $\|f\star g\|_p\leq \|f\|_p\|g\|_1$ should be mentioned, as it is used in several steps.

Lemma 2: it seems that $\tilde{p}$ is replaced by $f$ in the statement of the Lemma and at several steps of the proof. (Eg. the display right after 665.) This also seems to take place in the next Lemma.

Line 680: Since you are repeating a definition from the preceding proof, please say so.

Line 682: "We use the result of Lemma 1"

Line 746: Should "The following result holds" be here?

Display (34): this "fill distance" is the smallest value of $h$ such that $\tilde{X}$ is an $h$-net of $[-R,R]^d$.

Display below line 777: how should I interpret the norm in the RHS?

Lines 873/4: why are you spaeking of conditional probabilities here?

Line 891: what is $f(\delta)$?

Line 910: "redefine" should be "define again" (they are not the same).

Line 928: capitalize "Holder".

**Questions:**

* Have you attempted experiments against other ML approaches, or other methods from the literature?
* Methods have been proposed recently to speed up SDPs, including randomized sketches  (https://math.paperswithcode.com/paper/scalable-semidefinite-programming). Could these methods be applicable in the present setting?



**Limitations:**

They have not addressed limitations explicitly, but I do not think that that would have been necessary.

---

> ### Author Rebuttal · Authors · 2023-08-08
>
> Thanks a lot for assessing our work positively. Here below we address the major concerns raised by the reviewer.
>
> **Typos and Presentation**: Thank you very much for pointing us to typos. We will fix the typos and improve the presentation. We will also include suggested citations in our paper.
>
> Line 251: If a probability density function has a limit at infinity, that limit is always zero. We mention this because the PSD model-based representation of the density vanishes at infinity. We will clarify.
>
> Confusion regarding transpose in RKHS and $\eta$ will be clarified.
>
> Line 777: For our case, this is just the supremum of $f$ in the set $\mathcal{X}$.
>
> Line 873: We will fix that. That would be just "probability".
>
> Line 891: $f(\delta)$ is $k(\delta)$.
>
> Other minor confusion regarding $\tilde{p}$ and $f$ would be fixed.
>
>
>
> **Contributions and Experiments**: There is a significant lack of literature on solving the (fractional) Fokker-Planck equation and
> sampling from the solution. Our framework not only addresses this problem but also allows for the approximation of the fractional Laplacian acting on density, which is a distinct research problem. We believe that our work represents a valuable contribution to the field and has the potential to inspire further research in this area.
> The theoretical results presented in this paper are non-trivial and come with worst-case guarantees, which makes them stand out from methods that rely on stronger assumptions. Our findings are particularly significant in situations where traditional assumptions, such as log-Sobolev and Poincare fail. Despite the absence of real-world experiments, we hope that the reviewer could evaluate our theoretical contributions from the perspective of not requiring strict assumptions. Thank you very much for pointing out a reference for faster SDP via sketching. This method can certainly be utilized in solving the SDP program arising in the PSD model faster.

---

> > ### Author Response · Authors · 2023-08-11
> > **Start of the author reviewer discussion**
> >
> > We hope that our response answers the questions the reviewer asked. Please let us know the further questions or confusion about this work so that we can address it.

---

> > > ### Comment · Reviewer_oceV · 2023-08-11
> > > **Thank you**
> > >
> > > Thank you for the rebuttal. I do want to insist that it would have been nice to see experiments, though I understand that math is more important here.

---

> > > > ### Author Response · Authors · 2023-08-14
> > > > **RE: Thank you.**
> > > >
> > > > Thanks a lot for the acknowledgment.

---

### Official Review · Reviewer_biEz · 2023-07-07

**Soundness:** 2 fair
**Presentation:** 1 poor
**Contribution:** 2 fair
**Rating:** 3
**Confidence:** 2

**Summary:**

This paper studies the problem of efficiently sampling from a SDE given a drift function and diffusion tensor. It proposes a solution based on computing a PSD model that satisfies the Fokker-Planck solution associated with the SDE, and then sampling from the resulting PSD model.

**Strengths:**

The paper seems to have a lot of technical content, and it is clear that the authors have put in a lot of effort.

**Weaknesses:**

This paper is extremely difficult to read. The theorem statements (and the paper overall) need to be drastically simplified to be understandable. The problem doesn't seem to be well motivated, and it's not clear from reading the paper how the solution compares to others in the literature. For instance, the [LCL+21] paper is mentioned in related work, but it's not at all clear how the solution proposed of learning a PSD model compares to their work. There are no experiments to support this idea of fitting a PSD model. Overall, I think this paper needs to be rewritten to be more easily understandable.

**Questions:**

- Can you provide experimental evidence for the theoretical claims?
- Can you provide simplified theorem statements?
- Can you provide a more clear comparison with related work?

**Limitations:**

Yes

---

> ### Author Rebuttal · Authors · 2023-08-08
>
> We thank the reviewer for spending time reviewing our work. We agree that our paper is on the technical side; however, we suspect that the reviewer might have found the paper difficult to read if the subject is not close enough to their field of expertise. In such a case, we believe that the strong rejection would be an unfair suggestion, as the paper would be clear and understandable and the contributions would be significant for its intended audience at NeurIPS.
>
> We would now address the concerns raised by the reviewer here. We also request the reviewer to read our responses to other reviews to get further clarification. We hope that the reviewer could reconsider their decision based on our explanations.
>
> **Theorem Statements**: Theorem 2 and theorem 5 are about the approximation error guarantees of approximating the solution of the Fokker-Planck and fractional Fokker-Planck equation with the PSD model. More specifically, the statement says that we get $O(\varepsilon)$ approximation error given enough samples $m=\tilde{O}(\varepsilon^{-(d+1)/\beta-2s})$ where $\beta$ and $s$ are described in the paper. Corollary 1 provides a guarantee about the sample quality of samples obtained from the PSD model and the error guarantee is provided in Wasserstein-1 distance. Theorem 1 and Theorem 4 are intermediate approximation results for the infinite-dimensional PSD model. These results are useful in obtaining the final approximation results in theorems 2 and 5. Finally, the result of theorem 3 is about approximating the fractional Laplacian operator acting on a probability density represented by a PSD model. Theorem 3 clearly states that while using PSD model-based representation for probability density, one can approximate the fractional Laplacian operator (non-local operator) acting on a probability density for a wide choice of kernel functions. Approximating non-local operators is an independent research problem where we show the effectiveness of PSD-based representation (theorem 3).
>
> **Motivation**: As we described in the introduction of our paper, the main motivation behind this work is to propose an approach that can provably sample from a stochastic differential equation. We take an altogether different approach to address this problem that was not explored in the literature before to the best of our knowledge. We propose a two-step procedure. First, we approximate the solution of a (fractional) Fokker-Planck equation (using the PSD model) that corresponds to a stochastic differential equation.  In the next step, a sample is sampled from the PSD model. Under our proposed approach, we managed to prove the sampling and approximation results under much weaker conditions on the solution density functions ($\beta$ times differentiable) than what exists in the literature.
>
> **Advantage and Comparison**: To the best of our knowledge, we are not aware of any algorithm which provably can sample from the solution of a Fokker-Planck in a bounded domain under as much of a weak condition on the solution($\beta$ times differentiable) as ours. The convergence bound for existing sampling algorithms like Langevin requires dissipativity, log-Sobolev constants, or similar quantities. For example, very recent works ([Lam21] ; [ZL22] ) are able to provide convergence only under dissipativity-like assumptions.  We make no such assumption and still show that there exists an algorithm that can be used to sample from complex solutions of (fractional) Fokker-Planck equation in a ball given the solution is regular enough. If the solution is $d$ times differentiable or more (that is often the case), the dependence on the dimension vanishes up to logarithmic factors.
>
> It is clear from the results in theorem 2 and 5 that the sample complexity result in our work is better than the one reported in [LCL+ 21]. Also, the solution function obtained using   *Physics-inspired neural networks* (PINN) [RPK19], *Deep Ritz method* [Y+18], and other existing machine learning methods are not bound to stay positive in the domain and are often not integrable. Since the solution function is a probability density in our case and PSD based model satisfies the property of a density function (section 3.2 and [RC21]), hence comparison with results for methods like PINN and Deep Ritz Method [LCL+ 21] is not fair as those methods are not applicable in our case and can not be used in sampling from the solution.  We have tried our best to include existing related results from PDE and sampling in the related work section of our paper.
>
> **Experiment and Contribution**: There is a significant lack of literature on solving the (fractional) Fokker-Planck equation and
> sampling from the solution. Our framework not only addresses this problem but also allows for the approximation of the fractional Laplacian acting on density, which is a distinct research problem. We believe that our work represents a valuable contribution to the field and has the potential to inspire further research in this area. The theoretical results presented in this paper are non-trivial and come with worst-case guarantees, which makes them stand out from methods that rely on stronger assumptions. Our findings are particularly significant in situations where traditional assumptions, such as log-Sobolev and Poincare fail. Despite the absence of real-world experiments, we hope that the reviewer could re-evaluate our theoretical contributions from the perspective of not requiring strict assumptions and the novel contributions we make in multiple research problems.

---

> > ### Author Response · Authors · 2023-08-11
> > **Start of the author reviewer discussion**
> >
> > We hope that our response to the reviews (all the reviews) provided more clarification on the contributions we made to this work. Please let us know the further questions or confusion about this work so that we can address it. Please also look at other reviews and our responses to the questions asked by other reviewers. In light of all the novel contributions we have made to this paper, we would like to request the reviewer to reevaluate the score assigned to this paper.

---

> > > ### Author Response · Authors · 2023-08-14
> > > **Reminder: Start of the author reviewer discussion**
> > >
> > > Please let us know if you have further questions or confusion about this work that we can address. If we have answered all the questions from the reviewer, then in light of all the novel contributions we have made in this paper, we would like to request the reviewer to reevaluate the score assigned to this paper.

---

> > > > ### Comment · Reviewer_biEz · 2023-08-16
> > > >
> > > > I reread the paper after reading your comments. Currently I am not at all convinced by the story provided in this paper. The PSD model doesn't seem to be well-motivated, beyond statements like "PSD models are an effective way to model non-negative functions and enjoy the nice properties of linear models and more generally RKHS" which are provided without justification. The theorems and effectiveness of the solution are difficult to interpret.
> > > >
> > > > I would be much more convinced if you could:
> > > >
> > > > 1. Provide experiments that demonstrate distributions where this algorithm performs much better than simpler algorithms like Langevin Monte Carlo
> > > > 2. Provide theoretical examples where Langevin Monte Carlo provably fails to sample correctly, while your proposed algorithm succeeds. In particular, even though you claim: "Remark. It is evident from the aforementioned result that our method excels in sampling from the Stochastic Differential Equation (SDE) while ensuring the particle density adheres to Assumption 1 within a bounded domain. In contrast, traditional sampling methods such as Langevin struggle to sample from this extensive class due to the absence of dissipativity, log-sobolev, and Poincaré-like conditions.", this seems like it could possibly just be a failing of the existing analyses of Langevin, rather than a failing of the actual algorithm. It would be much clearer if you could provide a specific example, along with a proof that your algorithm succeeds on this specific example, but Langevin fails.
> > > > 3. Provided more easily interpretable theorem statements with direct comparisons to simpler algorithms (langevin). Currently the theorem statements are very difficult to interpret.
> > > >
> > > > For the reasons above, I maintain my score.

---

> > > > > ### Author Response · Authors · 2023-08-16
> > > > >
> > > > > Dear Reviewer,
> > > > > Thanks a lot for your response. We believe there has been a misunderstanding that we would like to clarify.
> > > > >
> > > > > 1. PSD Models for Probability Representation: We dedicated an entire section 3.2 to the effectiveness of using PSD Models for Probability Representation. It is clear from the formulation in Equation 5 that f is a non-negative function. Proposition 1 in the main paper says that marginalization is possible in closed form for the PSD model with a Gaussian kernel. Please also look at Proposition 3. We are not aware of another representation that comes with these guarantees.  This has also been discussed in detail in [RC21] and [MFBR20]. Please also look at the paper [RMFB20].
> > > > >
> > > > > 2. Theorem statements:  We clearly explained in our previous rebuttal that what are the implications of each of the theorem statements. Our proposed approach works in two parts. In the first approach, we approximate the (fractional)-Fokker Planck equation and we provide the guarantee for that in our theorems. If you are familiar with fokker planck and fractional Fokker Planck equations (section 3.1 in the paper) then the theorem statements are really saying that PSD-based approximation can approximate the solution of the fokker planck equation well. The theorem statements are currently in their simplest form stating the approximation error guarantee. The sampling result is given in Corollary 1.
> > > > >
> > > > > 3. Theoretically, we are not aware of any prior work that doesn't utilize either log-Sobolev or Poincare-type inequality to provide convergence for Langevin monte carlo. Even the latest results in this area (COLT 2023) rely on some kind of Poincare or log-Sobolev inequality to be satisfied.
> > > > >
> > > > >   (i) Towards a Complete Analysis of Langevin Monte Carlo: Beyond Poincar\'e Inequality: Mousavi-Hosseini et al.
> > > > >
> > > > >  (ii) Utilising the CLT Structure in Stochastic Gradient-based Sampling: Improved Analysis and Faster Algorithms:  Das et al.
> > > > >
> > > > > There is no one-on-one comparison between Poincare or log-Sobolev type inequality and our condition (that is beta times differentiable). Proposition 3 in our paper describes the extent to which the PSD model can approximate. Even if it is the artifact of analysis for Langevin that sampling guarantee does not hold when Poincare or log-Sobolev inequality fails to hold, our result theoretically improves upon existing sampling results. In that sense, our work is a valuable contribution. Log-Sobolev/Poincare inequality does not hold for the class of density discussed in Proposition 3.
> > > > >
> > > > > 4. We have already provided clarification regarding the experiments that we would reiterate. There is a significant lack of literature on solving the (fractional) Fokker-Planck equation and sampling from the solution. Our framework not only addresses this problem but also allows for the approximation of the fractional Laplacian acting on density, which is a distinct research problem. We believe that our work represents a valuable contribution to the field and has the potential to inspire further research in this area. The theoretical results presented in this paper are non-trivial and come with worst-case guarantees, which makes them stand out from methods that rely on stronger assumptions. Our findings are particularly significant in situations where traditional assumptions, such as log-Sobolev and Poincare fail.
> > > > >
> > > > > Despite the absence of real-world experiments, we hope that the reviewer could re-evaluate our theoretical contributions from the perspective of not requiring strict assumptions and the novel contributions we make in multiple research problems.
> > > > >
> > > > > Regards,
> > > > > Authors.

---

### Decision · Program_Chairs · 2023-09-21

**Decision:**

Accept (poster)

**Comment:**

Most reviewers think the paper is interesting.  Please revise the paper and show more benefits of your method over Langevin algorithms.